# Deviation in development of dorsal association tracts during preadolescence links to concurrent and future cognitive performance and transdiagnostic psychopathology

Danni Wang [1], Christopher J. Hammond[2], Betty Jo Salmeron [1], Xiang Xiao[1], Laura Murray [1], Hong Gu [1], Tianye Zhai[1], Annika Quam [1], Justine Hill [1], Hieu Nguyen[1], Hanbing Lu[1], Amy Janes[1], Thomas J. Ross [1] & Yihong Yang[1] ✉

Many psychiatric disorders begin during adolescence, coinciding with the rapid development of brain white matter (WM). However, it remains unclear whether deviations from normal WM development during this period contribute to psychopathology. In this study, we developed normative models of brain age based on specific WM tracts using three large-scale developmental datasets (~10,000 subjects). We found that tract-specific deviations in WM development of association and limbic/subcortical systems were linked to concurrent and future cognition and psychopathology. The spatial pattern of the association system aligned closely with high-order brain networks and mitochondrial maps. Importantly, delayed brain-age especially in dorsal association tracts predicted psychiatric disorders across diagnoses and disorder onset over a 2-year follow-up. By identifying tract-specific WM development during preadolescence as a predictor of cognitive capacity and psychiatric risks, this study provides a framework for tracking individualized brain development and understanding the neurobiological underpinnings of cognition and transdiagnostic psychopathology.

Psychiatric disorders are among the most common causes of morbidity and mortality[1,2], yet much is still unknown regarding their underlying developmental pathophysiologies[3]. One emerging hypothesis is that a large amount of variance in the risk for psychopathology comes from common "transdiagnostic" instead of disorder-specific factors. This is supported by findings showing that different psychiatric disorders share risk genes and show overlapping alterations in brain structure, connectivity and function[4–6]. Further, approximately 75% of all psychiatric disorders start before

the age of 21, with 35% starting before age of 14[7], suggesting that deviations in neurodevelopment may contribute to psychopathology risk in a transdiagnostic manner.

Personalized characterization of white matter (WM) tract development may provide significant insight into links between disrupted neurodevelopment and psychopathology as there is significant temporal overlap between WM changes during preadolescence and the onset of many psychiatric disorders[8–10]. It is plausible that WM alterations may contribute to mental health issues given that WM tract

[1]Neuroimaging Research Branch, National Institute on Drug Abuse, National Institutes of Health, Baltimore, Maryland, USA. [2]Department of Psychiatry and Behavioral Sciences, Johns Hopkins University, Baltimore, MD, USA. ✉e-mail: yihongyang@intra.nida.nih.gov

integrity in association and limbic tracts is linked with cognitive and socioemotional processes in pediatric populations[11]. Developmental variance in the structural integrity of these critical pathways may have widespread influences on cognitive and emotional functioning, which may in turn influence risk for psychopathology[12,13]. Further, the energetic demands of WM is higher during the preadolescent and early adolescent window, driven by maturation-related increases in physiological processes such as myelin synthesis and axon–oligodendrocyte[9]. Mitochondria, which are central to energy metabolism, have been shown in histological studies to exhibit substantial alterations in individuals with psychiatric disorders[14,15], implicating mitochondrial dysfunction as a potential contributor to impaired WM integrity. Given that WM enables efficient inter-regional communication and supports the transmission of neurotransmitter signals, its developmental disruption could have cascading effects on brain function. However, links between WM development, mitochondrial functioning, and psychopathology are poorly understood.

One way to assess normative growth and define pathological deviations in development is through the use of normative modeling which has been successfully applied in clinical contexts including growth charts in pediatrics[16] and standardized achievement and intelligence (IQ) tests in psychology[17]. Recently, this framework has been extended to measures of brain structure and function to quantify person-specific deviations in developmental trajectories relative to typically developing controls[18–21]. In the context of assessing individual brain development/aging level, machine learning prediction modeling using magnetic resonance imaging (MRI) data has allowed for the estimation of the biological age of the brain of healthy individuals[22]. Deviations between this neuroimaging-based 'brain age' and chronological age, termed the brain age gap (BAG), may reflect disruptions in normal developmental/aging trajectories and provide mechanistic information into cognitive development and the pathophysiology of psychiatric disorders[22–24]. To date, this method has predominantly focused on studies using morphological measures of gray matter (GM), with aggregate results from these studies suggesting that psychopathology is associated with accelerated brain aging in adults and youth[22–24]. However, few BAG studies have used diffusion MRI (dMRI) measures of WM, and these studies are mostly in adults. Results from these WM-BAG studies also suggest that adult patients with single psychiatric diagnoses (e.g., schizophrenia and bipolar disorder) may have more advanced WM brain aging relative to non-psychiatric controls[25,26]. While these prior studies highlight the relevance of the BAG in psychopathology, the role of WM maturation, particularly the tract-specific development, in adolescents is still unclear. Our study seeks to fill this knowledge gap.

In the present study, we sought to characterize WM BAG patterns derived from tract-specific microstructure in preadolescents with varying levels of psychopathology and examine associations between tract-based BAGs and a broad set of behavioral assessments spanning cognitive and psychopathology domains in youth. First, we built and cross-validated WM tract-based brain age models using diffusion MRI (dMRI) data from the Lifespan Human Connectome Project Development (HCP-D) dataset ($N = 611$). We then performed out-of-sample validation, using models derived from HCP-D data to predict brain ages in two independent datasets including Adolescent Brain Cognitive Development (ABCD) study ($N_{Baseline} = 8,688$) and Healthy Brain Network (HBN) pediatric mental health study ($N = 978$). Next, we used baseline data from the ABCD study and applied a multivariate sparse canonical correlation analysis (sCCA) to identify latent brain-behavior associations between WM-derived BAGs and cognition and psychopathology. Then, we assessed whether the tract-wise patterns of developmental levels linking cognition and psychopathology domains can be explained by tract-wise mitochondrial profiles from postmortem data[27]. Lastly, to ascertain the clinical utility of WM BAGs in follow-up cognitive performance and risk for psychiatric illness, we examined tract-based BAG associations with i) the cognitive performance measured 2 or 3 years after baseline, and ii) the cumulative number of psychiatric diagnoses assessed concurrently and 2 years after baseline, as well as longitudinal transitions between healthy and psychiatrically diagnosed states over the 2-year study period.

## Results

### Whole-brain and individual WM tract profiles predicted chronological age

First, we extracted 54 WM tracts from dMRI in three datasets (HCP-D, HBN, and ABCD) and categorized the tracts into 7 systems (Fig. 1A). The GM regions connected by each tract are indicated in Supplementary Table 1. Tract-based features were constructed using diffusion fractional anisotropy (FA) profiles along tracts (Fig. 1B). Because dMRI metrics are highly site-dependent (Supplementary Fig. 1), which can hinder cross-site model training and application, we applied data harmonization (see Supplementary Fig. 1 for harmonization effects) prior to all downstream analyses. We then built brain age models using these tract-based features of individual tracts (25 combined bilateral association/projection tracts and 4 callosal tracts) or whole brain (concatenated all tracts) from typically developing adolescents in HCP-D dataset ($N = 611$; 330 females, age range = 5.58-21.92 years) (Fig. 1C). The models were trained by a Gaussian Process Regression (GPR) algorithm[28], a nonparametric Bayesian machine learning approach capable of capturing the nonlinear developmental trajectories of WM microstructure. In addition, GPR has been widely adopted in brain-age research[29,30], and previous comparative studies have demonstrated that GPR performs competitively or better than alternative algorithm[30,31]. Performance of the brain age model from whole-brain tracts is shown in Supplementary Fig. 2 and Supplementary Table 2. The 5-fold cross-validation performance of the GPR model showed a Pearson $R$ for the relationship between chronological age and predicted age of 0.855 and a root mean squared error (RMSE) of 2.264 years. To test the prediction performance of the brain age model, we selected 978 participants from the independent HBN dataset (366 females, age range = 5.58–21.90 years) within the same age range as the HCP-D study. As shown in Supplementary Table 3 and Supplementary Fig. 2, whole-brain brain-age model significantly estimated age in HBN dataset with an $R$ of 0.622 and a RMSE of 3.337 years. For the prediction of brain ages in another independent dataset, the ABCD study ($N_{Baseline} = 8,688$, 4,205 females, $N_{2-y-follow-up} = 5,883$, 2,741 females; $N_{4-y-follow-up} = 2,351$, 1,118 females, age-range = 8.91-15.70 years), of which the age-range was much narrower and the prediction performance was with an $R$ of 0.543 and an RMSE of 2.041 years (Supplementary Fig. 2).

To evaluate the effects of age-range and intra-subject variability on the predictive performance of the brain-age models, we performed two additional analyses. First, we repeatedly resampled the same number of HBN participants ($N = 100$) across 100 iterations while systematically varying the age-range. Second, we resampled the same number of ABCD participants ($N = 500$ per timepoint) with 100 repetitions, and compared samples composed of non-overlapping subjects versus samples including overlapping subjects across the three time points. As shown in Supplementary Fig. 3, predictive performance ($R$ values) improved as the age-range becomes wider. In the ABCD resampling experiment, the distributions of $R$ values were comparable between the results from overlapped ($0.592 \pm 0.010$; Supplementary Fig. 4) and non-overlapped ($0.580 \pm 0.016$; Supplementary Fig. 4) samples. The overlapped samples exhibited only 1.3% greater variance explained, indicating that intra-subject overlap had limited influence on overall model performance.

For brain age models built from FA profiles of individual tracts, all 29 tract-based brain age models significantly predicted the age in the 5-fold cross-validation dataset (HCP-D) after false discovery rate (FDR) correction for multiple comparisons. The cross-validation

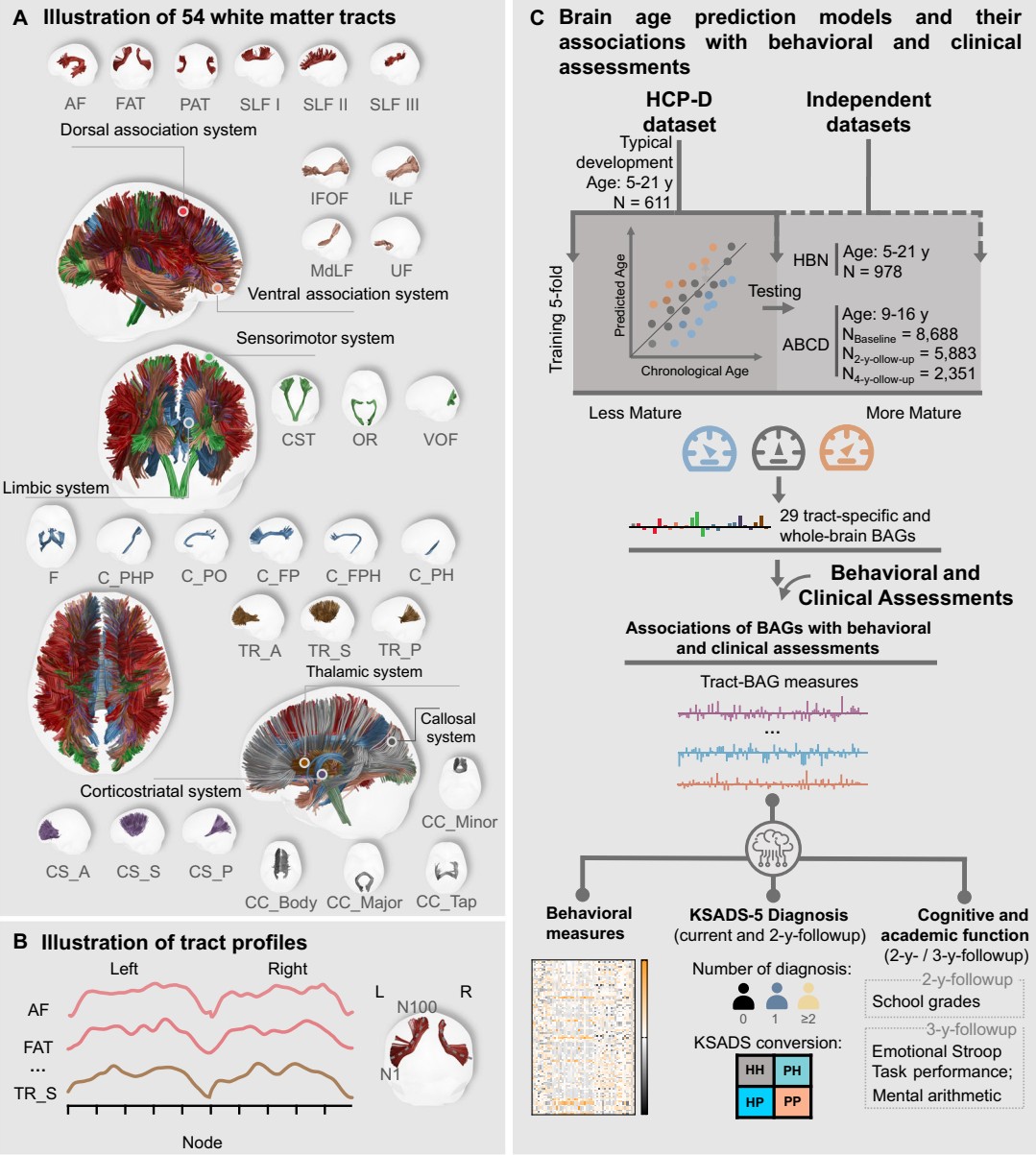

**Fig. 1 | Overview of study design. A** White matter tracts used for brain age modeling: 54 white matter tracts grouped into 7 distinct systems, for which normative models of brain age were constructed using tract-specific features. **B** Tract profiles of fractional anisotropy (FA). 100 segments (nodes) were evenly sampled along each unilateral tract, generating 200 FA values per bilateral tract. **C** Predictive models of brain age were generated using FA tract-profiles of the specific tracts from the participants in Human Connectome Project in Development (HCP-D) dataset with 5-fold cross-validation. One whole-brain and 29 specific-tract normative developmental models were established separately. Using models trained on HCP-D dataset, brain age gaps (BAGs) based on whole-brain and individual tracts were computed for each participant in the independent Healthy Brain Network pediatric mental health (HBN) and Adolescent Brain Cognitive Development (ABCD) datasets. Associations of the BAGs with behavioral and clinical assessments were investigated. Abbreviations: HH healthy-persistent; PH disorder-remitted; HP disorder-new-onset; PP disorder-persistent; KSADS-5 Kiddie Schedule for Affective Disorders and Schizophrenia for DSM-5; AF Arcuate Fasciculus; FAT Frontal Aslant Tract; PAT Parietal Aslant Tract; SLF Superior Longitudinal Fasciculus; IFOF Inferior Fronto Occipital Fasciculus; ILF Inferior Longitudinal Fasciculus; UF Uncinate Fasciculus; MdLF Middle Longitudinal Fasciculus; C_FPH Cingulum Frontal Parahippocampal; C_FP Cingulum Frontal Parietal; C_PH Cingulum Parahippocampal; C_PHP Cingulum Parahippocampal Parietal; C_PO Cingulum Parolfactory; F Fornix; CST Corticospinal Tract; OR Optic Radiation; VOF Vertical Occipital Fasciculus; TR_A Thalamic Radiation Anterior; TR_P Thalamic Radiation Posterior; TR_S Thalamic Radiation Superior; CS_A Corticostriatal Tract Anterior; CS_P Corticostriatal Tract Posterior; CS_S Corticostriatal Tract Superior; CC Corpus Callosum; CC_Body CC Body; CC_Major CC Forceps Major; CC_Minor CC Forceps Minor; CC_Tap CC Tapetum. Icons in Fig. 1C were made from https://www.svgrepo.com/. The person (by CyCraft, https://www.svgrepo.com/svg/393155/person-filled) and artificial-brain-computer (by howcolour, https://www.svgrepo.com/svg/416386/artificial-brain-computer) icons were adapted in color and are used under the Creative Commons license (CC BY 4.0, https://creativecommons.org/licenses/by/4.0/deed.en).

performance in HCP-D dataset was with $R$s ranging from 0.411 to 0.781 (RMSEs ranging from 2.633 to 3.687). See Supplementary Table 2 for the cross-validation performance of HCP-D dataset. For the prediction of brain age in testing datasets (HBN and ABCD), all tracts also exhibited significant prediction for age (Supplementary Table 3-4).

For each participant, the WM tract predicted age is referred to as brain age, and the difference between the brain age and chronological age is defined as tract-based BAG. The tract-based BAG quantifies the extent to which an individual's tract-specific biological age relative to other same-aged peers based on their tract-specific FA profiles. A

negative BAG indicates that an individual's predicted brain age is less than their chronological age (i.e., individual deviation under model-derived developmental trajectory), while a positive BAG indicates that an individual's predicted brain age is higher than their chronological age (i.e., individual deviation over model-derived developmental trajectory). To correct the potential brain age bias caused by regression dilution, we further trained brain age models with bias correction, a linear transformation of chronological age[32], on the training sets and applied them to the testing data sets. The predictive performance of the corrected tract-based brain ages is reported in Supplementary Table 2-4 and Supplementary Fig. 2. We used age-bias corrected BAGs for all primary statistical analyses, whereas uncorrected BAGs were used for validating brain age model performance. While there is ongoing debate regarding the optimal method for bias correction in brain age modeling[33,34], our key findings remained robust regardless of whether age-bias correction was applied (see sensitivity analyses), as the BAGs were only adjusted by a linear model of chronological age derived from an independent dataset, and chronological age was additionally included as a covariate in all subsequent analyses.

Puberty and its accompanying hormonal changes play a critical role in shaping brain development. To validate that BAGs capture biologically meaningful developmental processes, we examined their associations with both pubertal stage assessments and salivary hormone levels (DHEA, testosterone, and estradiol). Across the first two timepoints, higher whole-brain tract BAGs (with age-bias correction) were significantly associated with more advanced pubertal stage ($t_{(10,120)} = 4.685$, $p_{FDR} = 5.68 \times 10^{-6}$, Supplementary Fig. 5), after controlling for age, sex, and visit. In addition, whole-brain tract BAGs showed significant positive associations with higher levels of DHEA ($t_{(10,630)} = 3.558$, $p_{FDR} = 5.00 \times 10^{-4}$), testosterone ($t_{(9,872)} = 4.705$, $p_{FDR} = 5.68 \times 10^{-6}$), and estradiol ($t_{(4,583)} = 2.238$, $p_{FDR} = 0.025$). These hormone analyses accounted for covariates including age, sex, visit, caffeine intake and physical activity in the past 12 hours, collection time since midnight, saliva collection duration, and time from collection to freezer storage.

## Tract-based BAGs associated with cognition and psychopathology

To comprehensively characterize tract-based BAGs and their associations with general cognition and psychopathology, we examined associations of tract-based BAGs with a wide range of cognitive functions and psychopathological behaviors using multivariate sCCA, with age and sex as covariates. In the sCCA, we included 30 tract-based BAGs (29 tract-specific and a whole-brain tract BAGs with age-bias correction) and 51 "behavioral assessments" (20 neurocognitive and 31 psychopathology-related measures, see Methods) from the baseline ABCD dataset, which were utilized in our previous functional study[35]. The analysis showed that the first two canonical modes were statistically significant compared to null distributions generated by permutation tests (sCCA mode 1: $R = 0.19$, $p_{FDR} < 0.001$; sCCA mode 2: $R = 0.13$, $p_{FDR} < 0.001$). The scree plot of the covariance explained by all canonical modes is shown in Supplementary Fig. 6.

The loading patterns of the tract-based BAGs on the two significant sCCA modes were highly heterogeneous (Fig. 2A). BAGs obtained from most tracts in the dorsal and ventral association systems exhibited significant positive loadings for Mode 1. In contrast, BAGs from tracts mainly from the sensorimotor, limbic, cortico-striatum, cortico-thalamus and callosal systems had significant positive loadings for Mode 2. Therefore, in the following discussion, the 1st sCCA mode is referred to as Association BAG mode and the 2nd mode is referred to as Subcortical/limbic BAG mode. Cognitive measures had generally heavy and positive loadings on the two sCCA modes (Fig. 2B). To evaluate the specific behavioral measures associated with each sCCA mode, we compared the distributions of loadings generated

from 1000 bootstrap tests (see Methods). The measures with an effect size more than 0.5 were considered specific to the corresponding mode. As shown in Fig. 2B and Supplementary Table 5, the Association BAG mode significantly loaded on cognitive measures including total cognition composite and fluid cognition composite scores. Several psychopathological assessments, including Child Behavioral Checklist (CBCL) Attention Problems, Attention Deficit / Hyperactivity Disorder, and Total Problems exhibited significant and specific negative loadings on the Association BAG mode. For the Subcortical/limbic BAG mode, significant and specific cognitive scores included Little Man Task efficiency, Pattern Comparison processing speed and Stop Signal Reaction (SST) time, while Parent General Behavior Inventory (PGBI) Total Score of Mania showed significant negative loading.

Furthermore, we performed a post-hoc statistical analysis to explore the univariate relationships between the 30 tract-based BAGs (with age-bias correction) and 51 behavioral measures. The heatmap of the t values is in Supplementary Figs. 7-8, showing significant associations of nearly all the cognitive measures with tract-based BAGs. The Manhattan plot in Fig. 2C visualizes the associations between all tract-based BAGs and behavioral measures. The top 10 tract-based BAGs showing the strongest correlations with either cognitive or psychopathology domains are highlighted. Across both domains, dorsal association tracts consistently ranked higher than tracts from other systems. Notably, the behavioral measures showing the strongest associations with dorsal association tracts included total intelligence, fluid intelligence, CBCL Total Problems, Attention Problems, and ADHD scores.

## Functional decoding based on spatial association between brain regions connected by WM tracts and task-related brain activation

WM tracts connect distinct brain functional units that correspond to relevant task-related brain activations. To further characterize functions of GM regions connected by the WM tracts with loadings in the two identified sCCA modes, we utilized brain activation maps in the Neurosynth meta-analysis based on 100 topics[36]. For the functional activation map of each topic, we calculated the brain activation level within the GM regions connected by each association/projection WM tract. The topics (Supplementary Table 6) related to high-order cognitive ability, emotion and psychiatric disorders were selected to perform spatial correlation with the loading patterns of the two modes. A spatial permutation test (1000 times) was applied to test the significance of spatial correlation. As shown in Fig. 3A and Fig. 3B, similar spatial correlations were observed between the loading pattern of the Association BAG mode and distributions of cognitive-control- (e.g., "Cognitive-control & task performance") and language-related (e.g., "Reading & Words") brain activation connected by WM tracts. Furthermore, the loading patten of the Subcortical/limbic BAG mode was correlated with reward or emotion related topics such as "Depression," "Fear Conditioning & PTSD," and "Reward" (Fig. 3A-B and Supplementary Table 7).

To qualitatively compare differences in functional networks connected by the tracts significantly loaded on the two sCCA modes, we projected their connected brain regions onto the cortical surface and overlapped with 7-network parcellation by Yeo et. al. [37]. We observed that high-order networks (e.g., VAN, FPN and DMN) had the largest proportions of the brain regions connected by the tracts in the Association BAG mode. Moreover, the tracts in the Subcortical/limbic BAG mode connected mostly to the brain regions in sensorimotor and limbic networks (Fig. 3C).

## Mitochondrial correlates of two cognition-psychopathology profiles

To explore potential biological mechanisms underlying the two cognition-psychopathology profiles, we incorporated the only

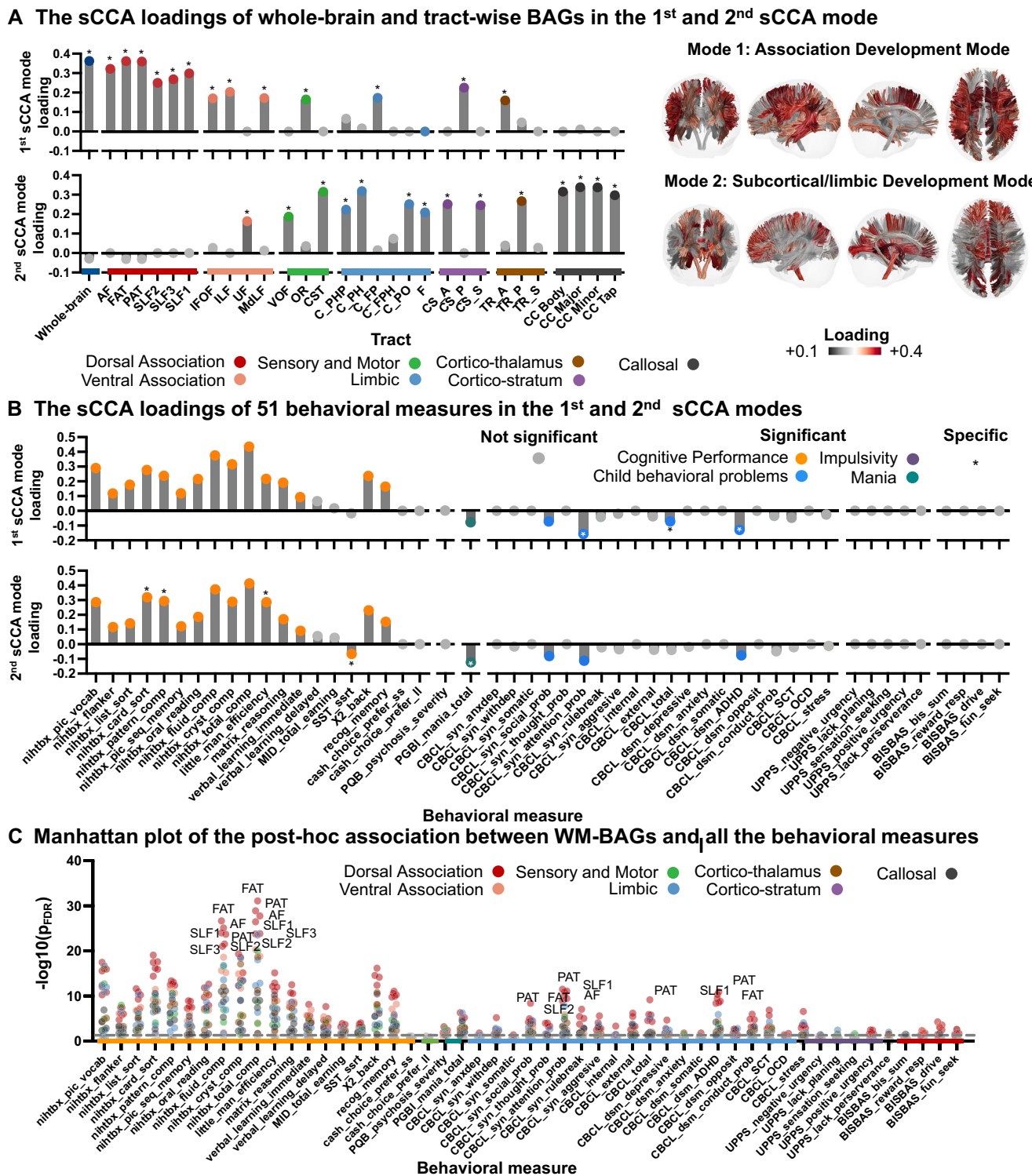

**A** **The sCCA loadings of whole-brain and tract-wise BAGs in the 1st and 2nd sCCA mode**

Mode 1: Association Development Mode

Mode 2: Subcortical/limbic Development Mode

Loading +0.1 ▬ +0.4

Dorsal Association ● (red) Sensory and Motor ● Cortico-thalamus ● (green) Callosal ● (grey)
Ventral Association ● (salmon) Limbic ● (blue) Cortico-stratum ● (purple)

**B** **The sCCA loadings of 51 behavioral measures in the 1st and 2nd sCCA modes**

Not significant ●   Significant   Specific *
Cognitive Performance ● Impulsivity ●
Child behavioral problems ● Mania ●

Behavioral measure

**C** **Manhattan plot of the post-hoc association between WM-BAGs and all the behavioral measures**

Dorsal Association ● Sensory and Motor ● Cortico-thalamus ● Callosal ●
Ventral Association ● Limbic ● Cortico-stratum ●

Behavioral measure

available set of six high-resolution ex-vivo mitochondrial maps derived from a postmortem adult brain[27], including 3 enzymatic activity maps (CI: NADH–ubiquinone oxidoreductase; CII: succinate dehydrogenase; CIV: cytochrome oxidase) reflecting energy-transformation, 1 mitochondrial content map (MitoD), and 2 derived maps representing tissue respiratory capacity ($TRC$; $TRC = \frac{\sqrt{CI} + \sqrt{CII} + \sqrt{CIV}}{3}$) and mitochondrial respiratory capacity (MRC; MRC = TRC/MitoD)[27] (see "Methods"). MRC refers to the ability of mitochondria to generate energy through oxidative phosphorylation, reflecting the energetic demands and metabolic support of specific tracts[27]. Tract-wise spatial

mitochondrial distributions were quantified by overlaying the mitochondrial maps onto the *HCP-1065 tract atlas*[38], from which mean mitochondrial values were extracted for each tract (Fig. 4A). Statistical significance of each mitochondrial profile was assessed using a spatial spin-permutation null model (1000 iterations). As shown in Fig. 4B, WM tracts with higher loadings in the Association BAG mode were significantly associated with increased enzymatic activity (CI: rho = 0.50, CII: rho = 0.48, CIV: rho = 0.51, $p_{FDR} < 0.05$), elevated mitochondrial content (rho = 0.49, $p_{FDR} < 0.05$) and higher MRC (rho = 0.52, $p_{FDR} < 0.05$).

**Fig. 2 | Multivariate associations between tract-based brain age gaps (BAGs; with age-bias correction) and behavioral assessments using sparse canonical correlation analysis (sCCA). A** Loadings of the brain features on the two sCCA modes. Color bar indicates sCCA loading. **B** Loadings of behavioral assessments on the two sCCA modes. Statistical significance was assessed using 1000 bootstrap resampling iterations, with loadings considered significant when the 95% confidence intervals excluded zero. Significant loadings are colored according to the behavioral/tract systems to which they belonged, with those specific to each mode highlighted with asterisks. **C** The Manhattan plot of the post-hoc associations between tract-based BAGs and all the behavioral measures. The dots above the gray dotted line indicate that the tract-based BAGs were significantly correlated with behavioral assessments ($p_{FDR} < 0.05$). Tract-based BAGs with significant associations are labeled with colors. The top 10 tract-based BAGs that were significantly associated with cognition or psychopathology domains are highlighted with the tract names, respectively. The two-tailed $p$ values with Benjamini-Hochberg false-discovery-rate (FDR) correction were provided in Source Data File. Abbreviations: AF Arcuate Fasciculus; FAT Frontal Aslant Tract; PAT Parietal Aslant Tract; SLF Superior Longitudinal Fasciculus; IFOF Inferior Fronto Occipital Fasciculus; ILF

Inferior Longitudinal Fasciculus; UF Uncinate Fasciculus; MdLF Middle Longitudinal Fasciculus; C_FPH Cingulum Frontal Parahippocampal; C_FP Cingulum Frontal Parietal; C_PH Cingulum Parahippocampal; C_PHP Cingulum Parahippocampal Parietal; C_PO Cingulum Parolfactory; F Fornix; CST Corticospinal Tract; OR Optic Radiation; VOF Vertical Occipital Fasciculus; TR_A Thalamic Radiation Anterior; TR_P Thalamic Radiation Posterior; TR_S Thalamic Radiation Superior; CS_A Corticostriatal Tract Anterior; CS_P Corticostriatal Tract Posterior; CS_S Corticostriatal Tract Superior; CC Corpus Callosum; CC_Body CC Body; CC_Major CC Forceps Major; CC_Minor CC Forceps Minor; CC_Tap CC Tapetum; nihtbx: NIH Toolbox; MID Monetary Incentive Delay; SST_ssrt Stop Signal Reaction Time of Stop Signal Task; PQB Prodromal Questionnaire Brief Version; PGBI Parent General Behavior Inventory; CBCL Children Behavior Check List; ADHD Attention-deficit/hyperactivity Disorder; SCT Sluggish Cognitive Tempo; OCD Obsessive-compulsive Disorder; UPPS Urgency, Perseverance, Premeditation and Sensation-seeking; BISBAS: Behavioral Inhibition & Behavioral Activation Scales. See Supplementary Table 19 and 20 for more details of the behavioral measures. Source data are provided as a Source Data file.

## Baseline tract-based BAGs are prospectively linked to cognitive performance two to three years later

Having established the associations between deviation in development of WM tracts and both cognitive function and psychopathology, we next investigated whether developmental deviations in WM tracts could reflect meaningful variance in cognitive performance at follow-ups. To avoid circularity, we selected general cognitive measures[39] that were not included in the prior sCCA analyses. Specifically, we included math ability (assessed via Stanford Mental Arithmetic Response Time Evaluation [SMARTE]), school grade and the overall performance of emotional Stroop task (measured by overall accuracy [STRP_ACC]), from a 2- or 3-year-follow-up (Fig. 5A). For the relationships with these three follow-up cognitive measures, the Association BAG mode demonstrated greater effect sizes (School Grade: $F_{(1,7868)} = 59.3$, $\Delta R^2_{adjusted} = 0.729$, $p_{FDR} = 1.48 \times 10^{-14}$; SMARTE: $F_{(1,5623)} = 102.9$, $\Delta R^2_{adjusted} = 1.698$, $p_{FDR} = 5.76 \times 10^{-24}$; STRP_ACC: $F_{(1,5967)} = 33.6$, $\Delta R^2_{adjusted} = 0.538$, $p_{FDR} = 6.84 \times 10^{-9}$) compared to those of Subcortical/limbic BAG mode (School Grade: $F_{(1,7868)} = 24.6$, $\Delta R^2_{adjusted} = 0.297$, $p_{FDR} = 7.12 \times 10^{-7}$; SMARTE: $F_{(1,5623)} = 69.6$, $\Delta R^2_{adjusted} = 1.15$, $p_{FDR} = 9.19 \times 10^{-17}$; STRP_ACC: $F_{(1,5967)} = 14.3$, $\Delta R^2_{adjusted} = 0.219$, $p_{FDR} = 1.60 \times 10^{-4}$). Post-hoc tract-wise analyses further revealed significant associations between tract-based BAGs association and the three cognitive measures, as shown in Fig. 5B and Supplementary Table 8-10. Among the three cognitive performances, more positive tract-based BAGs were associated with better cognitive performance assessed 2 to 3 years later.

To evaluate whether tract-specific BAGs capture more and distinct variance compared with tract-wise FA values, we conducted the supplementary statistical analyses using baseline tract-wise FA values to associate with follow-up cognitive performance. As shown in Supplementary Fig. 13, baseline FAs of the Cingulum Parahippocampal explained the greatest variance in 2-year follow-up school grades ($F_{(1,7868)} = 49.15$, $\Delta R^2_{adjusted} = 0.602$, $p_{FDR} = 2.57 \times 10^{-12}$) and 3-year follow-up SMARTE performance ($F_{(1,5623)} = 88.31$, $\Delta R^2_{adjusted} = 1.46$, $p_{FDR} = 7.94 \times 10^{-21}$), whereas baseline FAs of the FAT accounted for the highest variance in 3-year follow-up STRP_Acc ($F_{(1,5967)} = 21.47$, $\Delta R^2_{adjusted} = 0.338$, $p_{FDR} = 3.66 \times 10^{-6}$) among the 30 tracts. Across all three follow-up cognitive measures, a greater number of tracts showed higher variance explained by tract-specific BAGs than by FA values (Supplementary Fig. 13A). In addition, we examined the similarity of effect patterns between tract-specific BAGs and FA values using Spearman correlation. As illustrated in Supplementary Fig. 13B, the FA-based effect patterns did not significantly resemble those derived from the tract-specific BAGs, suggesting that tract-specific BAGs provide complementary and differential variance linking to follow-up cognitive performance.

## More negative tract-based BAGs at baseline concurrently associated with greater number of psychiatric (KSADS-5) diagnoses

We assessed whether tract-based BAGs were associated with psychiatric diagnoses using general linear models (GLMs). 8594 participants with valid dMRI and psychiatric assessments from the ABCD dataset at baseline were used for the main analysis. For each participant, categorical psychiatric diagnoses were measured using the parent-reported Kiddie Schedule for Affective Disorders and Schizophrenia for DSM-5 (KSADS-5). sCCA variate scores derived from the Association BAG mode at baseline showed a significant association with the cumulative numbers of KSADS-5 diagnoses assessed concurrently ($F_{(2,8591)} = 15.06$, $p_{FDR} = 9.44 \times 10^{-6}$, Fig. 6A-left), such that individuals, with less tract-based age relative to chronological age had higher number of psychiatric disorders. sCCA variate scores derived from the Subcortical/limbic BAG mode showed a smaller effect for the association ($F_{(2,8591)} = 6.22$, $p_{FDR} = 0.008$, Fig. 6A-left).

In addition, we also conducted the post-hoc association analyses on the relationships of BAGs (with age-bias correction) obtained from individual tracts and whole-brain tracts, respectively, with the clinical diagnoses. BAG of the whole-brain tracts at baseline was significantly associated with the cumulative number of KSADS-5 diagnoses (0, 1, or >=2) assessed at the same time, after FDR correction (KSADS-5 effect: $F_{(2,8591)} = 7.05$, $p_{FDR} = 0.004$, Fig. 6B-upper). Moreover, BAGs of all tracts within dorsal association system at baseline showed significant associations with the cumulative number of diagnoses assessed concurrently (e.g., FAT: $F_{(2,8591)} = 9.78$, $p_{FDR} = 0.001$ and PAT: $F_{(2,8591)} = 13.89$, $p_{FDR} = 1.52 \times 10^{-5}$, Fig. 6B-upper). See Supplementary Table 11 for results of other tracts and post-hoc analyses. By contrast, none of the tract-wise FA values at baseline were significantly associated with the cumulative number of concurrent diagnoses when analyzed using the same statistical models (Supplementary Table 12).

## Deviation in WM developmental trajectory assessed via BAGs at baseline prospectively predicted cumulative number of psychiatric (KSADS-5) diagnoses at 2-year follow-up

Next, the GLM analysis revealed that the scores derived from both sCCA variates at baseline were significantly associated with the cumulative number of diagnoses at 2-year follow-up (Association BAG mode: $F_{(2,7904)} = 11.90$, $p_{FDR} = 2.21 \times 10^{-4}$; Subcortical/limbic BAG mode: $F_{(2,7904)} = 7.20$, $p_{FDR} = 0.004$, Fig. 6A-middle). Further, post-hoc subgroup-wise analyses contrasting the difference between each pair of the subgroups showed that the scores of both sCCA variates were significantly lower in the subgroups with two or more diagnoses compared to the subgroup with no diagnosis.

Post-hoc tract-wise comparisons demonstrated that age-bias-corrected BAG of the whole-brain tracts at baseline were significantly

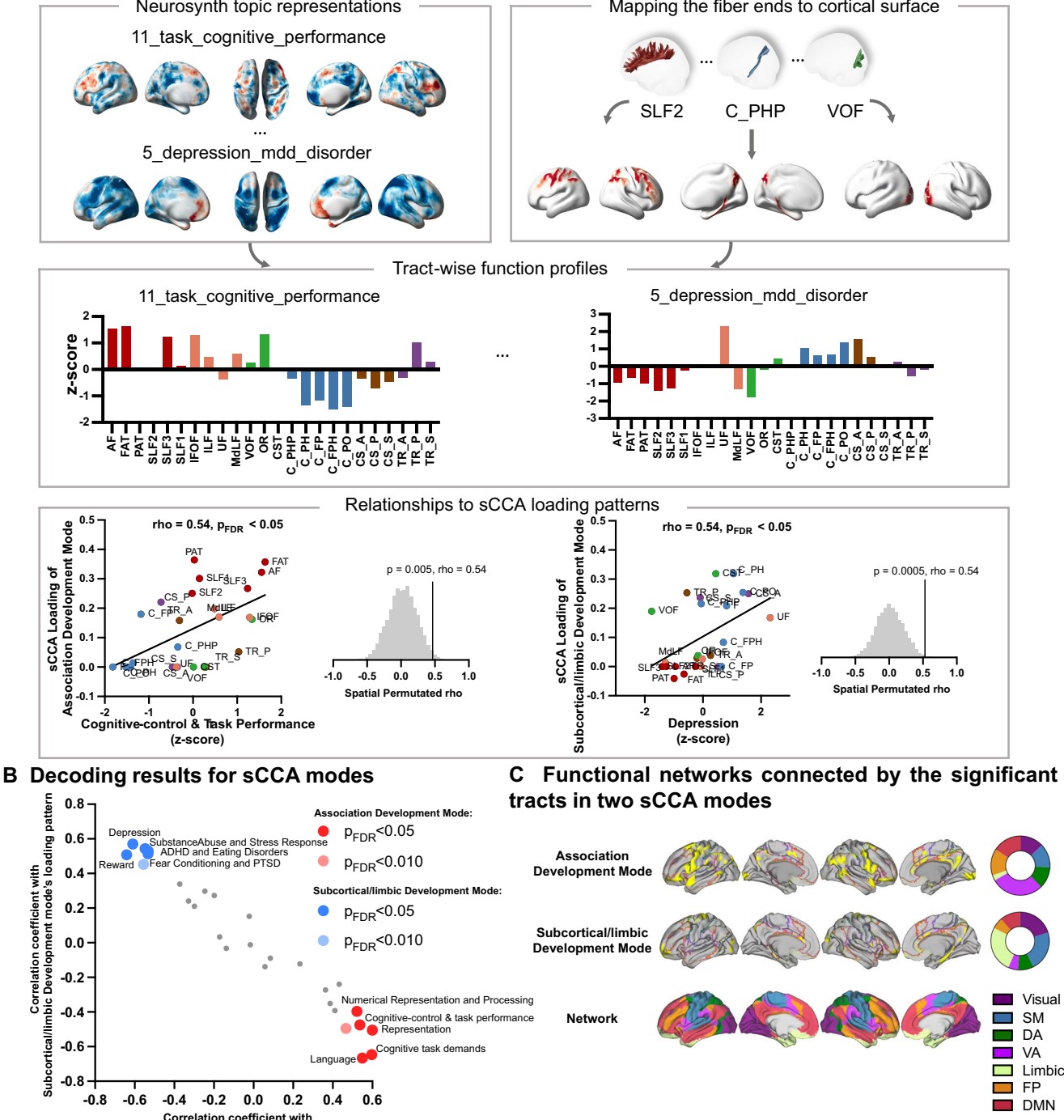

**A  Decoding the functions of Association and Subcortical/limbic Development modes**

Neurosynth topic representations

11_task_cognitive_performance

...

5_depression_mdd_disorder

Mapping the fiber ends to cortical surface

SLF2    C_PHP    VOF

Tract-wise function profiles

11_task_cognitive_performance

5_depression_mdd_disorder

Relationships to sCCA loading patterns

rho = 0.54, p_FDR < 0.05

p = 0.005, rho = 0.54

rho = 0.54, p_FDR < 0.05

p = 0.0005, rho = 0.54

**B  Decoding results for sCCA modes**

Association Development Mode:
- $p_{FDR} < 0.05$
- $p_{FDR} < 0.010$

Subcortical/limbic Development Mode:
- $p_{FDR} < 0.05$
- $p_{FDR} < 0.010$

**C  Functional networks connected by the significant tracts in two sCCA modes**

Association Development Mode

Subcortical/limbic Development Mode

Network

- Visual
- SM
- DA
- VA
- Limbic
- FP
- DMN

associated with the cumulative number of diagnoses at 2-year follow-up ($F_{(2,7904)} = 7.03$, $p_{FDR} = 0.005$, Fig. 6B-middle), with the significantly more negative BAG in the subgroup with 2 or more diagnoses compared to the subgroup with no diagnosis. Age-bias-corrected BAGs of 6 association tracts and 5 limbic/cortico-subcortical tracts at baseline showed significant associations with the cumulative number of diagnoses at 2-year follow-up (e.g., FAT: $F_{(2,7904)} = 8.64$, $p_{FDR} = 0.002$; CS_P: $F_{(2,7904)} = 7.91$, $p_{FDR} = 0.003$; Fig. 6B-middle and Supplementary Table 11). In addition, supplementary comparative analyses using tract-wise FA values revealed that lower harmonized FA in the Corticospinal Tract (CST: $F_{(2,7904)} = 12.98$, $p_{FDR} = 7.05 \times 10^{-5}$) and Cingulum Para-hippocampal Parietal (C_PHP: $F_{(2,7904)} = 5.14$, $p_{FDR} = 0.044$) were

significantly associated with a greater number of diagnoses. Results for other tracts are provided in Supplementary Table 12. The tract-wise group effects based on FA exhibited a different directional pattern of variance in relation to the number of concurrent diagnoses compared with those observed for tract-wise BAGs (FA vs. BAG: rho = 0.366, $p_{FDR} = 0.1380$, Supplementary Fig. 14).

**Deviation in WM developmental trajectory assessed via BAGs at baseline prospectively associated with psychiatric diagnosis status transitions from baseline to 2-year follow-up**

Brain variate scores of both sCCA modes at baseline significantly associated with the status transitions of - diagnoses from baseline to

**Fig. 3 | Results of tract-based functional decoding analysis. A** Workflow for evaluating meta-analytic functional decoding of two sparse canonical correlation analysis (sCCA) loading patterns of the tract-based brain age gaps (BAGs). Selected Neurosynth[36] topic maps represented on cortical surface were visualized in the left top panel. For each topic, a tract-wise functional profile was calculated by averaging z-scored brain functional activation probability within the gray matter (GM) regions connected by each white matter (WM) tract. In the right panel, representative scatterplots of spatial correlations between tract-wise functional profiles and sCCA loading patterns are shown. **B** Functional decoding results for two sCCA loading patterns. Spearman correlation coefficients with two sCCA loading patterns are plotted on x and y axes, respectively. The significance of the results was used by one-sided spin-permutation test, with Benjamini-Hochberg false-discovery-rate (FDR) correction applied. The significantly correlated topics are highlighted and labeled. **C** Proportions of the functional networks connected by the WM tracts significant in the two sCCA modes. Pie plots demonstrate the percentages of tract-connected cortical voxels located in the seven resting-state cortical networks

(defined by Yeo et al.[37] for the two sCCA modes. Abbreviations: SMN sensorimotor network; DAN dorsal attention network; VAN ventral attention network; FP frontoparietal network; DMN default mode network; FDR false discovery rate; PTSD Posttraumatic Stress Disorder; ADHD Attention-deficit/hyperactivity Disorder; AF Arcuate Fasciculus; FAT Frontal Aslant Tract; PAT Parietal Aslant Tract; SLF Superior Longitudinal Fasciculus; IFOF Inferior Fronto Occipital Fasciculus; ILF Inferior Longitudinal Fasciculus; UF Uncinate Fasciculus; MdLF Middle Longitudinal Fasciculus; C_FPH Cingulum Frontal Parahippocampal; C_FP Cingulum Frontal Parietal; C_PH Cingulum Parahippocampal; C_PHP Cingulum Parahippocampal Parietal; C_PO Cingulum Parolfactory; F Fornix; CST Corticospinal Tract; OR Optic Radiation; VOF Vertical Occipital Fasciculus; TR_A Thalamic Radiation Anterior; TR_P Thalamic Radiation Posterior; TR_S Thalamic Radiation Superior; CS_A Corticostriatal Tract Anterior; CS_P Corticostriatal Tract Posterior; CS_S Corticostriatal Tract Superior; CC Corpus Callosum; CC_Body CC Body; CC_Major CC Forceps Major; CC_Minor CC Forceps Minor; CC_Tap CC Tapetum. Source data are provided as a Source Data file.

2-year follow-up (Association BAG mode: $F_{(3, 7826)} = 10.52$, $p_{FDR} = 2.11 \times 10^{-5}$; Subcortical/limbic BAG mode: $F_{(3, 7826)} = 4.75$, $p_{FDR} = 0.008$, Fig. 6A-right). Post-hoc subgroup-wise comparisons showed that the brain variate scores of both sCCA modes were significantly lower in Disorder-persistent (Association BAG mode: $z = -4.410; p < 0.001$; Subcortical/limbic BAG mode: $z = -3.685; p = 0.001$) compared to Healthy-persistent subgroup, while the brain variate scores of the Association BAG mode were significantly lower in Disorder-new-onset and Disorder-remitted subgroups compared to Healthy-persistent subgroup.

Furthermore, supplementary post-hoc tract-wise comparisons revealed that the age-bias-corrected BAG of the whole-brain tracts at baseline showed a significant association with diagnostic status transitions ($F_{(3, 7826)} = 6.06$, $p_{FDR} = 0.002$, Fig. 6B-bottom). For BAGs (with age-bias correction) of individual tracts, all the tracts in the dorsal association system (e.g., AF: $F_{(3, 7826)} = 7.04$, $p_{FDR} = 0.001$; FAT: $F_{(3, 7826)} = 7.27$, $p_{FDR} = 0.001$) (Fig. 6B-bottom and Supplementary Fig. 6) were significantly associated with diagnostic status transitions. See Supplementary Table 11 for other tracts. Comparative analyses showed that the tract-wise FAs in tracts such as the CST and the C_PHP were significantly associated with diagnostic status transitions. However, no significant relationship was observed between FA- and BAG-based group effects related to the diagnostic status transitions (rho = 0.205, $p_{FDR} = 0.278$, Supplementary Fig. 14).

### Consistency of the BAG variates and their associations with cognitive performance and psychiatric diagnoses

To further evaluate the consistency of the two sCCA BAG variates, we repeated the sCCA analyses using behavioral measures from either the cognition or psychopathology domain. In both domain-specific analyses, the sCCA variates of Association and Subcortical/limbic BAG modes still showed significance after permutation testing. The loading patterns of the re-derived sCCA variates showed significant consistency with the Association and Subcortical/limbic BAG modes using full behavioral set, respectively, on both behavioral and BAG sides. See Supplementary Figs. 10–12 for more details.

Next, we examined the associations between psychopathology-derived sCCA BAG variates and follow-up cognitive performance. Consistent with the full-behavior-derived results, psychopathology-derived Association BAG variate (School Grade: $F_{(1, 7868)} = 58.86$, $\Delta R^2_{adjusted} = 0.72$, $p = 1.90 \times 10^{-14}$; SMARTE: $F_{(1, 5623)} = 93.50$, $\Delta R^2_{adjusted} = 1.55$, $p = 6.00 \times 10^{-22}$; STRP_ACC: $F_{(1, 5967)} = 32.13$, $\Delta R^2_{adjusted} = 0.51$, $p = 1.51 \times 10^{-8}$;) exhibited stronger associations with follow-up cognitive performance than the Subcortical/limbic BAG variate (School Grade: $F_{(1, 7868)} = 19.62$, $\Delta R^2_{adjusted} = 0.23$, $p = 9.57 \times 10^{-6}$; SMARTE: $F_{(1, 5623)} = 65.47$, $\Delta R^2_{adjusted} = 1.08$, $p = 7.16 \times 10^{-16}$; STRP_ACC: $F_{(1, 5967)} = 11.63$, $\Delta R^2_{adjusted} = 0.18$, $p = 0.001$).

Furthermore, we also investigated the relationships between cognition-derived sCCA BAG variates and the cumulative number of KSADS-5 diagnoses at baseline, at 2-y-follow-up, and the transition of the transdiagnostic status between baseline and 2-y-follow-up. GLM analyses demonstrated the statistical patterns were consistent with the behavior-derived results: more negative Association and Subcortical/limbic BAG variate scores were associated with more comorbid psychiatric disorders both at baseline (Association mode: $F_{(2, 8591)} = 14.96$, $p = 3.24 \times 10^{-7}$; Subcortical/limbic mode: $F_{(2, 8591)} = 6.54$, $p = 0.0015$) and at 2-y-follow-up (Association mode: $F_{(2, 7904)} = 11.94$, $p = 6.67 \times 10^{-6}$; Subcortical/limbic mode: $F_{(2, 7904)} = 7.74$, $p = 0.0005$), and the longitudinal transition of KSADS-5 diagnosis status (Association mode: $F_{(3, 7826)} = 10.44$, $p = 7.49 \times 10^{-7}$; Subcortical/limbic mode: $F_{(3, 7826)} = 5.07$, $p = 0.0017$).

### Sensitivity analysis for the effects of puberty, socioeconomic status (SES) and age-bias correction

We conducted several sensitivity analyses to evaluate the robustness of our findings and determine if they remained significant after controlling for potential confounding factors. First, to test whether age-bias correction influenced the magnitude and significance of associations observed between WM BAGs and psychiatric diagnoses, we repeated the analyses using two more alternative strategies: (i) uncorrected BAGs with chronological age included as a covariate, and (ii) age-bias-corrected BAGs without age as a covariate. The results (Supplementary Table 13-14) remained consistent, indicating that the observed associations were not dependent on the bias correction procedure.

Second, we sought to account for potential effects of pubertal development on our outcomes, by re-running our main analyses including Pubertal Development Scale (PDS) scores as an additional covariate. This analysis yielded unchanged results, suggesting that puberty did not confound the observed associations (Supplementary Table 15).

Lastly, to account for effects related to SES on our outcomes, we reran our main analyses including two measures of SES (family income and parental education) as covariates. The results of this analysis (Supplementary Table 16) were unchanged from our main results, indicating that the associations observed between tract-based BAG and psychiatric diagnoses could not be better accounted for by SES-related factors.

### Discussion

In this study, we investigated links of WM tract development during preadolescence with cognitive performance and transdiagnostic dimensions of psychological functioning. First, we established and validated normative models of development for specific and whole-

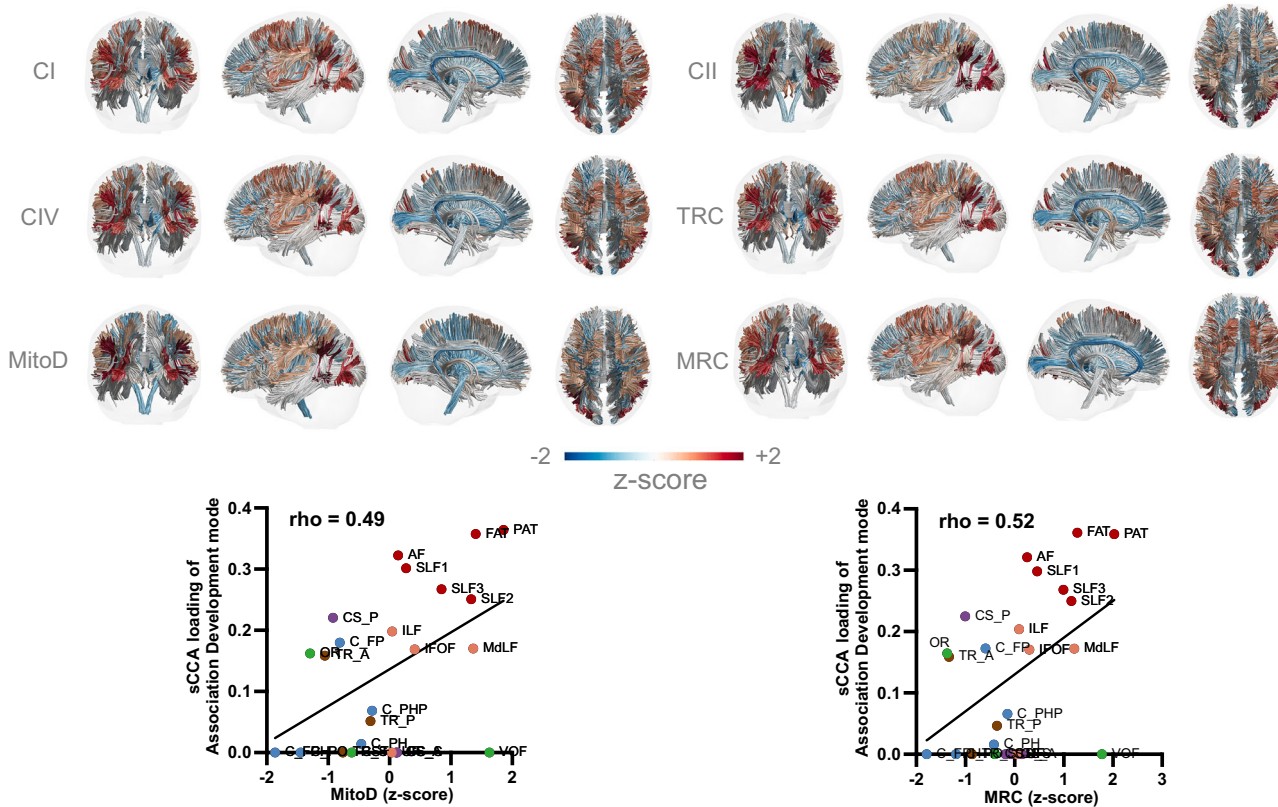

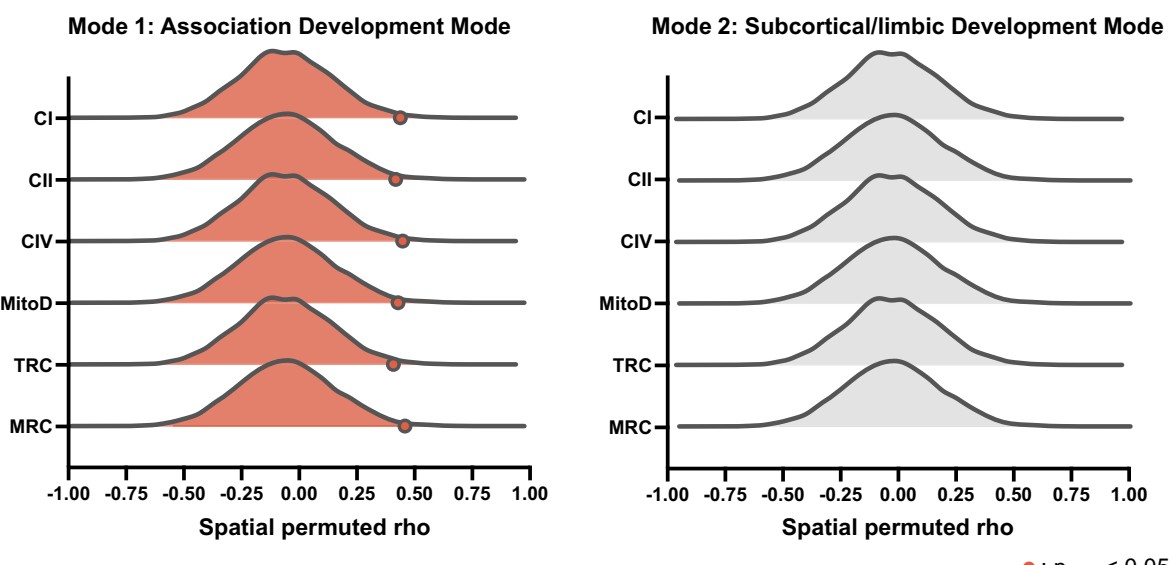

**Fig. 4 | Results of tract-based mitochondrial decoding analysis. A** Visualization of tract-wise mitochondrial profiles and spatial correlations with sparse canonical correlation analysis (sCCA) loadings for the Association BAG mode. **B** Association between tract-wise mitochondrial profiles and sCCA loadings of two modes. The distributions of rho values generated by spin-permutation tests (10,000 iterations) are plotted. The significance of the results was used by one-sided spin-permutation test, with Benjamini-Hochberg false-discovery-rate (FDR) correction applied. The mitochondrial map showing significant association with sCCA loading pattern is highlighted. Abbreviations: CI complex I; CII complex II; CIV complex IV; TRC tissue respiratory capacity; MitoD mitochondrial density; MRC mitochondrial respiratory capacity; AF Arcuate Fasciculus; FAT Frontal Aslant Tract; PAT Parietal Aslant Tract; SLF Superior Longitudinal Fasciculus; IFOF Inferior Fronto Occipital Fasciculus; ILF Inferior Longitudinal Fasciculus; UF Uncinate Fasciculus; MdLF Middle Long-itudinal Fasciculus; C_FPH Cingulum Frontal Parahippocampal; C_FP Cingulum Frontal Parietal; C_PH Cingulum Parahippocampal; C_PHP Cingulum Para-hippocampal Parietal; C_PO Cingulum Parolfactory; F Fornix; CST Corticospinal Tract; OR Optic Radiation; VOF Vertical Occipital Fasciculus; TR_A Thalamic Radiation Anterior; TR_P Thalamic Radiation Posterior; TR_S Thalamic Radiation Superior; CS_A Corticostriatal Tract Anterior; CS_P Corticostriatal Tract Posterior; CS_S Corticostriatal Tract Superior; CC Corpus Callosum; CC_Body CC Body; CC_Major CC Forceps Major; CC_Minor CC Forceps Minor; CC_Tap CC Tapetum. Source data are provided as a Source Data file.

## A  Data collection timeline

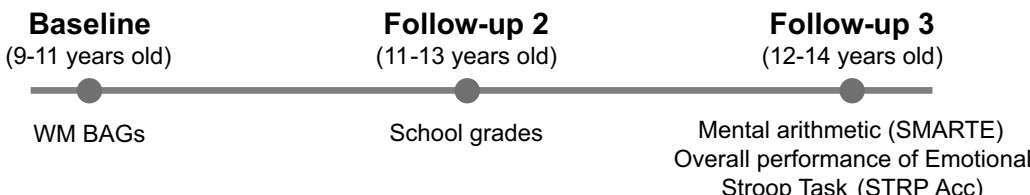

| **Baseline** | **Follow-up 2** | **Follow-up 3** |
| (9-11 years old) | (11-13 years old) | (12-14 years old) |
| WM BAGs | School grades | Mental arithmetic (SMARTE) Overall performance of Emotional Stroop Task (STRP Acc) |

## B  WM tracts / sCCA variates significantly associated with follow-up cognitive measures

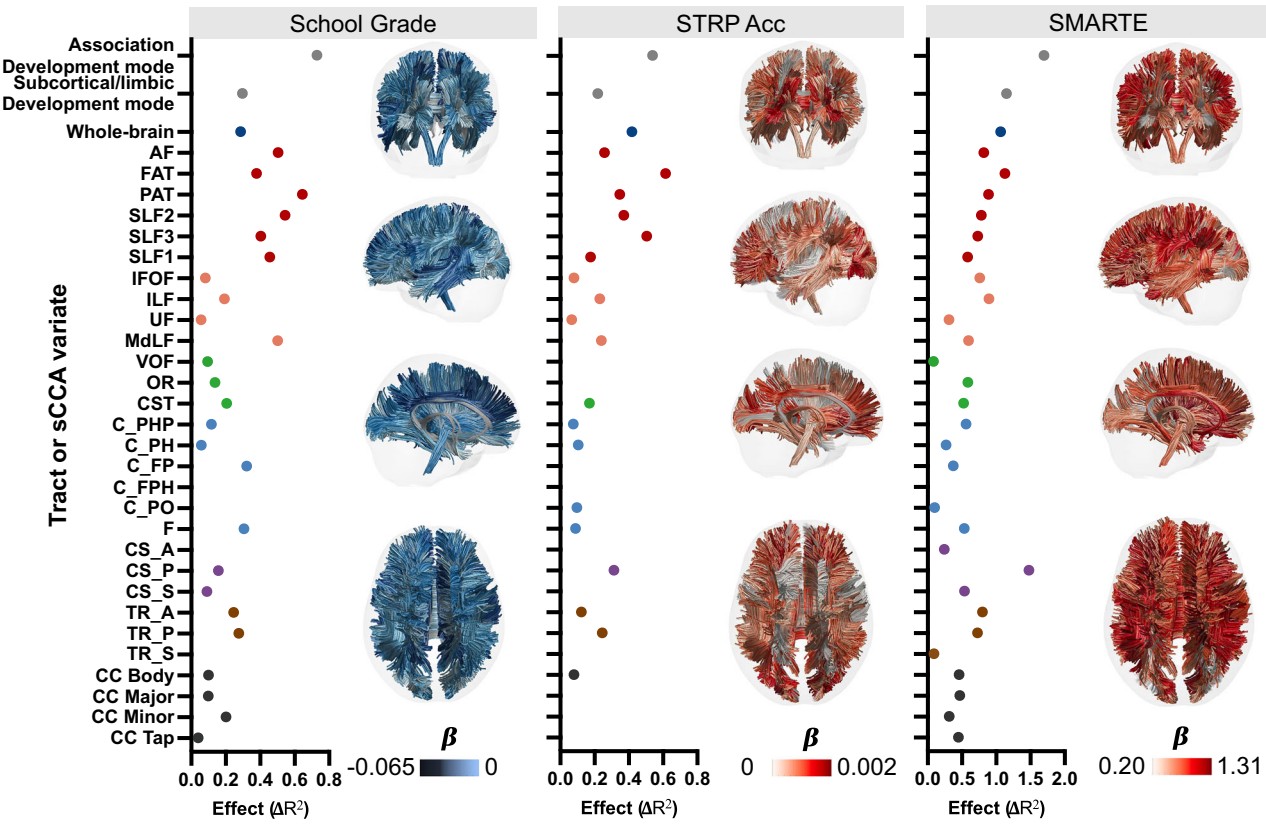

**Fig. 5 | Relationships between baseline tract-BAG measures and cognitive performance at follow-ups. A** Data collection timeline for cognitive measures at follow-ups. School grade was measured at 2-year follow-up, while Emotional Stroop Task performance and math ability were assessed at 3-year follow-up. **B** Relationships between tract-based measures (with age-bias correction) and follow-up cognitive measures. Only the tract-based measures showing significant associations with cognitive measures are plotted. Effect sizes ($\Delta R^2$) were calculated based on the change in overall proportion of variance by adding the predictor into the covariates-only model. The color of each tract on the right side of the sub-panel corresponds to the GAM coefficient of tract-based BAGs with significance. Abbreviations: Stanford Mental Arithmetic Response Time Evaluation; STRP_Acc emotional Stroop task accuracy; GAM generalized additive model; AF Arcuate

Fasciculus; FAT Frontal Aslant Tract; PAT Parietal Aslant Tract; SLF Superior Longitudinal Fasciculus; IFOF Inferior Fronto Occipital Fasciculus; ILF Inferior Longitudinal Fasciculus; UF Uncinate Fasciculus; MdLF Middle Longitudinal Fasciculus; C_FPH Cingulum Frontal Parahippocampal; C_FP Cingulum Frontal Parietal; C_PH Cingulum Parahippocampal; C_PHP Cingulum Parahippocampal Parietal; C_PO Cingulum Parolfactory; F Fornix; CST, Corticospinal Tract; OR Optic Radiation; VOF Vertical Occipital Fasciculus; TR_A Thalamic Radiation Anterior; TR_P Thalamic Radiation Posterior; TR_S Thalamic Radiation Superior; CS_A Corticostriatal Tract Anterior; CS_P Corticostriatal Tract Posterior; CS_S Corticostriatal Tract Superior; CC Corpus Callosum; CC_Body CC Body; CC_Major CC Forceps Major; CC_Minor CC Forceps Minor; CC_Tap CC Tapetum. Source data are provided as a Source Data file.

brain WM tracts using three large, independent dMRI datasets. Critically, our results show that chronological age during development can be significantly predicted using tract-based imaging features as well as whole-brain features. Next, we identified two distinct tract-BAG-based latent variates, weighted on the developmental levels of association tracts and subcortical/limbic tracts respectively, using a multivariate correlation analysis, that were associated with a wide range of cognitive and psychopathological measures. In both tract-based BAG modes, more positive BAGs were linked with greater cognitive function. In contrast, advanced BAG of association tracts was linked to lower psychopathology symptoms as measured by the CBCL, while the

development of subcortical/limbic tracts was positively related to lower manic symptoms. Then, we showed that positive BAG of association tracts was more predictive of follow-up cognitive performance than other individual tracts. Finally, we found that negative BAG of the dorsal association tracts was concurrently and prospectively associated with the number of psychiatric diagnoses at baseline and 2-year follow-up, and predicted diagnostic status changes over that 2-year period. Collectively, the overlap in findings among these distinct analyses provides strong evidence that general cognitive performance and transdiagnostic psychiatric disorders are associated with deviation in development of the association tracts during preadolescence.

**A    Associations between baseline BAGs of the sCCA variates and transdiagnostic status**

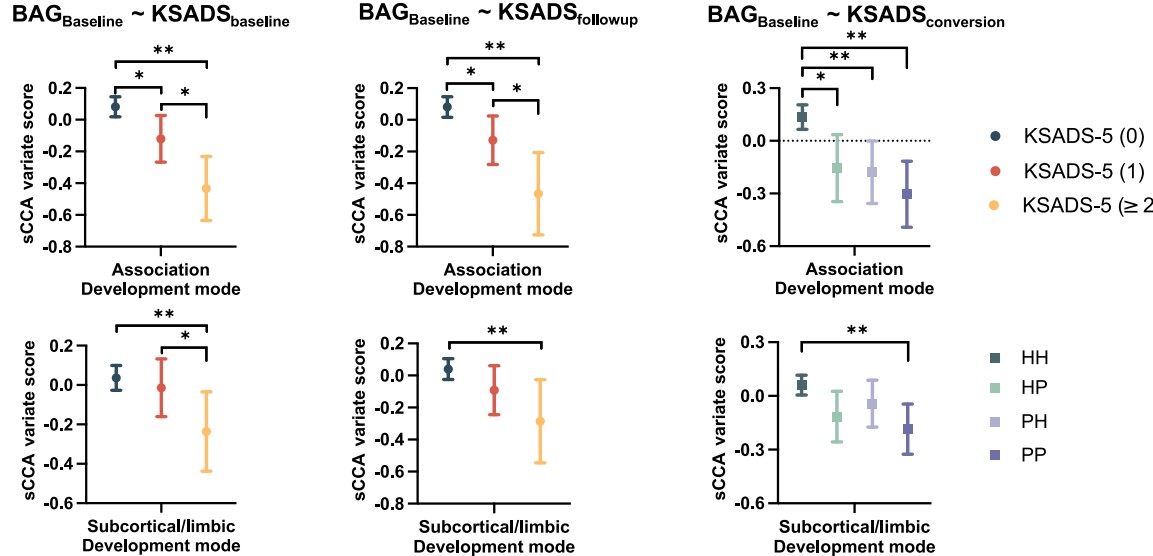

**B    Associations between baseline tract-based BAGs and transdiagnostic status**

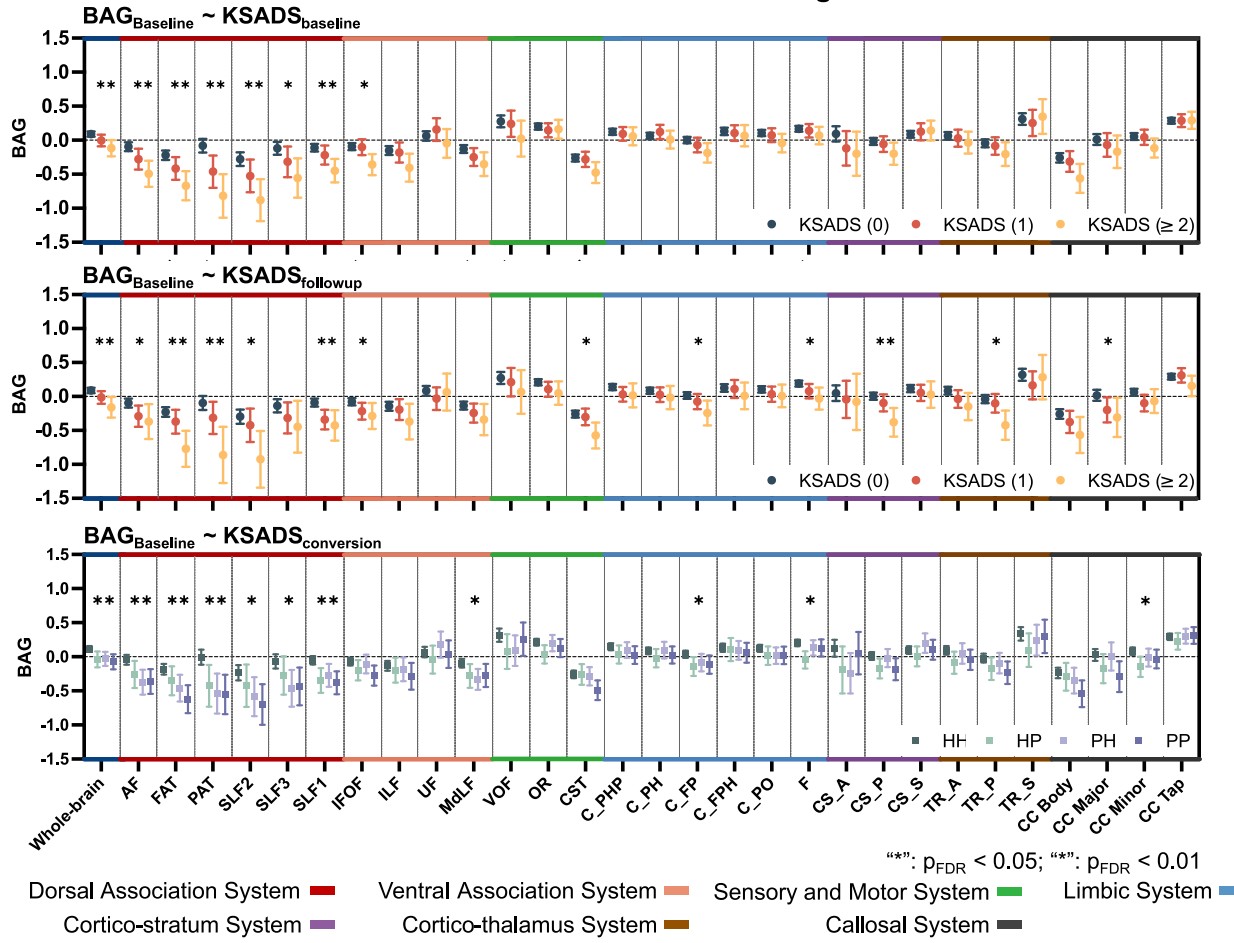

## Deviations in WM tract development assessed via BAG during preadolescence

Development trajectories of WM tracts have been extensively studied, indicating significant WM growth throughout childhood and adolescence[40,41]. Our current study demonstrated that chronological age was significantly predicted using tract-based dMRI features combined with machine learning algorithms. Our whole-brain model achieved an $R$ of 0.855 and RMSE of 2.264 years without age-bias correction in cross-validation. These results are comparable to other brain-age prediction studies in adolescents[30,42–44]. Further, our study's use of three independent datasets is a major strength reflecting the generalizability of the findings. We conducted a unique characterization on individual development of 29 WM tracts extracted from dMRI data with high reliability[38]. Using diffusion features of individual WM tracts, chronological age was significantly predicted in independent testing datasets. Interestingly, individuals with similar BAGs predicted

**Fig. 6 | Relationships of baseline tract-BAG measures with the cumulative number of KSADS-5 diagnoses at baseline, at 2-y-follow-up, and the transition of the transdiagnostic status between baseline and 2-y-follow-up.** Group-wise comparisons were conducted by generalized linear model (GLM) analyses. After false discovery rate (FDR) correction, a two-tailed $p$ value less than 0.05 was considered significant. **A** Groupwise comparison of the baseline sparse canonical correlation analysis (sCCA) variate scores with the cumulative number of KSADS-5 diagnoses at baseline ($n = 8594$), 2-y-follow-up ($n = 7907$) and 2-y-follow-up transdiagnostic status transitions ($n = 7830$). **B** Associations between baseline tract-based BAGs (with age-bias correction) and the cumulative number of KSADS-5 diagnoses at baseline, 2-y-follow-up and 2-y-follow-up transdiagnostic status transitions. The top two panels show the tract-BAG sCCA variate scores of the groups with different cumulative number of KSADS-5 diagnoses at baseline and 2-y-follow-up. The bottom panel displays the tract-BAG sCCA variate scores of four groups including Healthy-persistent (HH), Disorder-remitted (PH), Disorder-new-onset (HP) and Disorder-persistent (PP) group. Each dot represents the group-level mean BAG-related score, and error bars indicate 95% confidence intervals. "*" $p_{FDR} < 0.05$; "**" $p_{FDR} < 0.01$. Abbreviations: BAG brain-age gap; KSADS-5 Kiddie Schedule for Affective Disorders and Schizophrenia for DSM-5 (KSADS-5); AF Arcuate Fasciculus FAT, Frontal Aslant Tract PAT, Parietal Aslant Tract SLF, Superior Longitudinal Fasciculus; IFOF Inferior Fronto Occipital Fasciculus; ILF Inferior Longitudinal Fasciculus; UF Uncinate Fasciculus; MdLF Middle Longitudinal Fasciculus; C_FPH Cingulum Frontal Parahippocampal; C_FP Cingulum Frontal Parietal; C_PH Cingulum Parahippocampal; C_PHP Cingulum Parahippocampal Parietal; C_PO Cingulum Parolfactory; F Fornix; CST Corticospinal Tract; OR Optic Radiation; VOF Vertical Occipital Fasciculus; TR_A Thalamic Radiation Anterior; TR_P Thalamic Radiation Posterior; TR_S Thalamic Radiation Superior; CS_A Corticostriatal Tract Anterior; CS_P Corticostriatal Tract Posterior; CS_S Corticostriatal Tract Superior; CC Corpus Callosum; CC_Body CC Body; CC_Major CC Forceps Major; CC_Minor CC Forceps Minor; CC_Tap CC Tapetum. Source data are provided as a Source Data file.

by the whole-brain WM tracts showed varying tract-specific BAG profiles. Conceptually, BAG represents the statistical deviation from the age-normative trajectory. However, the calculation of BAG could be impacted by both physiological factors (e.g., neurodevelopmental deviation) and technical noise (e.g., MRI data quality). Therefore, validating that BAG captures neurodevelopmentally meaningful variance is essential. In our study, tract-based BAG showed significant positive associations with pubertal development, including both pubertal stage and hormone levels. This is in line with previous findings showing that puberty is associated with increased FA in major tracts of adolescents[45], and that higher FAs in WM regions such as the CC and cingulum are linked to elevated gonadal hormone levels in adolescent females[46]. Together, these results provide initial evidence that tract-based BAGs track with known metrics of developmental benchmarks and are consistent with prior BAG findings derived from GM morphology[47]. Thus, our tract-based BAG approach is capable of providing unique insights into typical or atypical WM development at the individual level.

## Tract-based BAGs define dimensions shared by cognition and psychopathology

Our sCCA analyses identified two brain-behavior modes (Association BAG and Subcortical/limbic BAG modes), linking advanced BAG of WM tracts with better cognitive function and less risk of psychopathology (Fig. 2). More specifically, greater development of WM association tracts was significantly linked with lower total psychological and attentional problems. Post-hoc univariate correlation analysis also showed that individuals with advanced BAG of WM tracts generally had better cognitive performance and reduced psychiatric vulnerability (Supplementary Figs. 7-8). Over the last decade, growing evidence has linked the development of WM microstructure to high-order cognitive abilities such as working memory, inhibition and cognitive shifting[48]. A study using the ABCD dataset also showed that a latent factor derived from executive functions was positively linked to higher FA in nearly all the involved WM tracts[49], which fits with the wealth of findings linking WM deficits and various psychiatric disorders[19,50]. Importantly, our findings extend previous dMRI brain age research by demonstrating that tract-specific BAGs capture shared dimensions of cognitive and psychopathological variation during adolescence. While prior dMRI studies have reported associations between whole-brain WM BAGs and psychiatric disorders in adults, the directionality of these findings was opposite to our results. For instance, studies in adult schizophrenia and bipolar disorder patients have shown more positive whole-brain WM BAGs[25,26], suggesting accelerated aging of WM microstructure. However, our results revealed that more positive tract-based BAGs were associated with better cognitive performance and reduced psychopathological risks. This contradiction may reflect fundamental differences between adolescent and adult neuropathology from the

developmental view. Consistent with this neurodevelopmental interpretation, Roy et al. found that more positive whole-brain WM BAGs were associated with better educational opportunity in youth[44], and Ullman et al. demonstrated that higher whole-brain WM brain-age related to better working memory and numerical ability in 6-year-old children[51]. These age-dependent patterns in younger agers support that WM deficits were proposed to occur prior to illness onsets, suggesting that psychiatric disorders are neurodevelopmental in nature[52]. Furthermore, our tract-specific approach further reveals that the crucial roles of dorsal association tracts during adolescence, revealing tract-specific vulnerabilities that may be neglected in global metrics.

This work also proposes a framework that integrates meta-analytic brain activation and mitochondrial/chemoarchitectural maps to elucidate neurobiological mechanisms of specific WM tracts. By analyzing the brain task-fMRI activations in regions connected by these tracts, we identified that the BAGs of Association BAG mode are associated with cognitive-control, language and attention functions, while the Subcortical/limbic BAG mode correlates with emotion processing and psychiatric symptoms. Our results showing cognitive and psychopathology associations with WM BAGs specific to association and subcortical/limbic tracts reinforces previous brain-behavior findings and may offer preliminary insights into how WM development contributes to distinct brain functional activation patterns derived from meta-analyses. Previous studies have shown that functional connectivity of the association system (e.g., frontoparietal and salience-ventral attention networks) contributed more to higher-order cognition[35,53]. In addition, abnormal (hypo- or hyper-) functional activation was evident particularly in the association system (e.g., salience-ventral attention networks) in transdiagnostic psychiatric disorders[6,54]. Consistent with this model, our results demonstrated that greater development of association tracts captured the largest variations in the behavioral set and related more to higher-order cognition and lower total psychopathology. Through mitochondrial decoding analyses, our tract-based analyses provide exploratory evidence that WM tracts most strongly implicated in cognition and psychopathology also show higher mitochondrial enrichment (Fig. 4). Specifically, we found that the WM tracts with high loadings in the Association BAG Mode had higher mitochondrial content and energy metabolism than other WM tracts. Brain regions exhibiting higher MRC are known to have emerged later in evolutionary timeline[27], suggesting increased vulnerability during neurodevelopment from an evolutionary psychiatry perspective[55]. WM tracts connecting association cortices undergo a more protracted development, reaching adult-level maturation at a later stage than those connecting sensory-motor cortices[11]. During early adolescence, WM tracts, particularly the association connections, have greater myelination, which requires enhanced mitochondrial activity and metabolism[9]. Consequently, WM tracts connecting association cortices may be particularly susceptible to mitochondrial

dysfunction during this developmental window. Impaired mitochondrial function, as a result of exogenous or endogenous factors, may delay the maturation of these tracts, potentially compromising adolescents' adaptive capacity in the face of cognitive and emotional challenges. Our findings align with post-mortem studies that have shown reduced WM mitochondrial density in patients with psychiatric disorders compared to non-psychiatric controls[14], providing the preliminary support for the "mitochondrial dysfunction" hypothesis in psychiatric disorders[15].

Our findings revealed apparently contrasting associations between BAGs derived from WM and those derived from whole-brain GM morphological features with respect to psychiatric symptom severity. Previous research has indicated that larger, positive BAGs based on GM features are often linked to more severe psychopathology[24]. The divergent associations of WM- and GM-derived BAGs with psychopathology suggest that brain age reflects distinct and multi-dimensional maturational processes. GM maturation, reflected by changes in cortical volume, surface area and thickness, typically followed by an inverted U-shaped developmental trajectory by early synaptogenesis peaks in early childhood, followed by synaptic pruning and cortical thinning extending through adolescence[21]. In contrast, WM maturation reflects a more protracted course characterized by continued increases in myelination and fiber organization into early adulthood, supporting increasingly efficient long-range communication[9]. Thus, a more "advanced" GM age during adolescence may indicate accelerated pruning and reduced plasticity[24], consistent with prior work linking higher GM BAG to vulnerability to psychiatric symptoms. Conversely, a more "advanced" WM age likely reflects greater myelination and fiber organization. This interpretation aligns with developmental frameworks suggesting that deviations in neurodevelopmental pace may differ across tissue compartments and that low-risk environments support prolonged plasticity in GM but continued strengthening of WM pathways[56]. The opposing directions of GM and WM BAG associations therefore likely represent complementary neurodevelopmental clocks rather than conflicting findings. Future multimodal longitudinal studies are needed to trace how GM and WM maturation jointly shape risk versus resilience trajectories across adolescence.

### Advanced BAG of specific tracts as a potential predictive biomarker of later cognitive performance

Our findings also highlight the potential of advanced WM BAG in linking to cognitive performance two to three year later. Advanced WM BAG, particularly along the tracts implicated in the Association Development, was consistently associated with better cognitive performance. The superiority of this mode over Subcortical/limbic BAG mode in predicting later-life cognitive performance suggests that more matured or optimized development of high-order association tract systems may have more long-term cognitive advantage. Multivariate predictive modeling further supports this point, revealing that using all tract-based BAG measures, more specifically from dorsal association and cortico-striatal systems, provided the strongest predictive power for adolescents' cognitive performance at follow-ups. The WM tracts within Association BAG mode connected the high-order brain networks of DMN, FPN and VAN. Integration of the triple networks including VAN, FPN and DMN networks is critical for maintaining cognitive-control ability in both adolescents and adults[57]. Cognitive-control ability includes inhibitory control, cognitive flexibility and working memory, which plays a vital role in high-order cognitive tasks like math performance, language and learning[58]. Previous functional study showed that higher functional connectivity levels of high-order brain networks, including DMN, FPN and VAN, were associated with better concurrent scholastic performance in participants aged 7 to 9 years old[59]. DTI studies in adolescents and adults also consistently found increased FAs in corticocortical and corticostriatal tracts were related to cognitive-control ability[58,60,61]. Importantly, tract-based BAGs explained more variance than "raw" whole-tract FA values in many tracts, indicating that BAG captures developmentally specific variance beyond microstructural integrity alone. Incorporating age-normative information appear to provide complementary and differential sensitivity to individual differences relevant to cognitive development. Our findings verify the structural backbone of these high-order brain networks also be related to cognitive performance in adolescents and extend that the BAG-indexed maturation level of structural connectivity at baseline have the predictive potential for follow-up cognitive performance.

### Negative BAG of specific tracts: insights into transdiagnostic and conversion status

In this study, we investigated the associations between tract-based BAGs and transdiagnostic status concurrently and at 2-year follow-up, as well as the conversion in diagnostic status, which reflects the change of transdiagnostic status across the 2-year follow-up period. We found that transdiagnostic dimensions of psychopathology and clinical status transitions from healthy to psychiatrically diagnosed states were both associated with more negative BAG of dorsal association tracts. The dorsal association tracts mainly connect the cortical brain areas of inferior parietal, frontal and superior parietal regions[38,60], largely overlapping with dorsal attention, ventral attention, and frontoparietal functional brain networks. This is consistent with our previous study showing that functional connectivity of the attention and frontoparietal systems is positively associated with cognition and negatively associated with psychopathology in preadolescents[35]. Moreover, reduced GM volume in the salience-ventral attention system is observed across individuals with psychiatric conditions in a disease non-specific manner, suggesting that disruptions in these networks have transdiagnostic implications[5,62]. Conversely, in the current study, the WM tracts mainly connecting subcortical and limbic brain regions, which were related to reward-related measures, showed less predictive potential for the cumulative number of diagnoses concurrently and at 2-year follow-up. The development of limbic-, subcortical- and cortical-subcortical-based WM tracts (e.g., cingulum, fornix), involved in emotion regulation and decision-making, has shown a protracted developmental trajectory compared to projection and association tracts[11,61,63]. Their prolonged maturation period coincides with the onset of mood disorders[64] and may suggest its critical role for supporting cognition-emotion integration.

### Limitations and future work

This study has several limitations. First, in this study, we primarily used data from subjects aged 8 to 11 years, with clinical and cognitive measures from the ABCD dataset at baseline. Given that there are different developmental windows for specific tracts[11], it is plausible that links between psychopathology and developmental disruptions in other tracts would be evident if larger age ranges were assessed. Future studies should include further released follow-up data from the ABCD dataset and other developmental datasets with a broader age range to investigate how tract-specific developmental variability plays crucial roles in the neuropathology of transdiagnostic disorders at different life stages. Additionally, we trained the tract-based models using the HCP-D dataset, which provides high-quality dMRI of adolescents and covers a wide age range in adolescence, allowing us to avoid overfitting by training models independently from the ABCD and HBN datasets. However, the moderate sample size of HCP-D ($N = 611$) suggests that future work may benefit from incorporating larger and more diverse multisite datasets to further enhance model generalizability. Second, BAGs may be influenced by multiple factors beyond true neurodevelopmental variation. Previous studies have shown that both technical aspects (e.g., dMRI acquisition and preprocessing schemes[65]) and non-technical factors (e.g., pharmacological

manipulation[66]) can affect BAG estimations. Although we validated the relationships between tract-based BAG and developmental benchmarks (pubertal stages and hormone levels), tract-based BAGs should nonetheless be interpreted as relative indices of deviation from normative WM developmental trajectories. Further work leveraging large-scale longitudinal datasets, integrating a broader range of lifestyle and environmental factors, and more biologically informed tract-based imaging modalities may help disentangle these sources of variability and improve interpretability. Third, our findings indicated that accelerated WM development was associated with better cognitive performance and lower transdiagnostic psychiatric dimensions, a relationship that contrasts with findings linking accelerated GM with greater psychiatric symptoms. Future multimodal BAG research is essential to examine how differing developmental trajectories of various neuroimaging markers such as WM and GM contribute to cognitive outcomes and psychiatric risk profiles. Fourth, our tract-wise mitochondrial profiles are currently based on a single postmortem adult brain that is not age/developmentally matched with our pre-adolescent sample. While it's well known that mitochondrial content and function undergo changes across the lifespan[67,68], the utilized maps remain the only available postmortem dataset offering such high-resolution, voxel-wise mitochondrial data in WM[27]. To assess developmental trajectories of mitochondrial properties in WM, future work is needed to measure mitochondrial profiling in adolescent brains. In the present study, we primarily focused on the spatial patterning of tract-wise mitochondrial distribution, which is presumed to be relatively stable across age groups. Positron emission tomography studies have reported comparable spatial distributions of mitochondrial translocator protein (measured using [$^{11}$C]PK11195) in both adults and children[69,70].

In summary, our work showed that tract-specific WM features significantly predicted brain age for adolescents via a machine learning approach. We established tract-based BAGs, which enables deconvolution of the different developmental levels within an individual and measures brain development at tract-wise resolution. Using sCCA, we identified two distinct tract-BAG based latent brain variates, one loading on association tracts and the other on tracts connecting limbic or subcortical regions, which were associated with a wide range of cognitive and psychopathological measures. These findings provide early evidence that deviation in development of WM tracts, especially in dorsal association tracts, may underlly transdiagnostic risk for psychiatric disorders that emerge during preadolescence and early adolescence.

## Methods
### Participants
This study complies with all relevant ethical regulations. All datasets included in this study are publicly available and anonymized human brain sources, which have been approved by their institutional boards or ethics committees[71–73]. No new human subjects were recruited for this study. The written informed consent was obtained for all participants in the original studies. For participants under the age of 18, written consent was provided by their legal guardians and assent was obtained from the participants. As the study focus on tract-based developmental correlates without a prior hypothesis regarding sex differences, sex was included as a covariate in all primary analyses, and no sex- or gender-specific analyses were conducted.

**Human Connectome Project in Development (HCP-D) Study.** The HCP-D dataset, designed to characterize healthy brain development in children and adolescents, was utilized as training and cross-validation for building a white-matter tract-based brain age model. This developmental dataset featured a predominantly cross-sectional design and comprised 652 individuals (aged 5.58 to 21.92 years; 351 females) with typical development[71], of whom are without the following exclusion

criteria: 1) Serious neurological condition; (2) History of serious head injury; (3) Long term use of immunosuppressants or steroids; (4) Premature birth; (5) Claustrophobia; (6) Hospitalization >2 days for certain physical or psychiatric conditions or substance use; (7) Treatment >12 months for psychiatric conditions or (9) Pregnancy or other MRI contraindications. All MRI data were acquired on 3 T Siemens Prisma Scanners at four imaging sites. See more details about scanning parameters in Supplementary Table 17-18. The study was approved by a central Institutional Review Board at Washington University in St. Louis.

**Healthy Brain Network pediatric mental health (HBN) Study.** To validate the prediction performance in an independent dataset with the similar age range, we included 1687 subjects (600 females) from 5.58 to 21.90 years old from HBN study[72]. The MRI data was collected with 3 T Siemens Prisma scanners at City College of New York and the CitiGroup Cornell Brain Imaging Center, and with a 3 T Siemens Tim Trio scanner at Rutgers University Brain Imaging Center. See Supplementary Table 17-18 for more scanning details. The study was approved by the Chesapeake Institutional Review Board.

**Adolescent Brain Cognitive Development (ABCD) Study.** We used the brain age model built from the HCP-D data to predict brain age for participants in the ABCD study, which is a prospective, ongoing, multi-site assessment of development in U.S. children from age 9–10 years though adulthood[73]. The MRI and behavioral data (v5.1) at two time-points, baseline, 2-year- and 4-year-follow-ups, were collected from 11,875 children. The T1-weighted (T1w) anatomical MRI and dMRI were acquired with magnetization-prepared rapid acquisition gradient echo (MPRAGE) and simultaneous multi-slice/multiband echo-planar imaging (EPI) sequences, respectively, using a unified protocol designed for different types of 3 T MRI scanner platforms including Siemens Prisma and Prisma Fit, General Electronic Discovery 750, and Philips Medical System Ingenia and Achieva dStream. See Supplementary Table 17-18 for more scanning parameter details.

### MRI data preprocessing and quality control (QC)
**HCP-D.** This dataset underwent the HCP's minimal preprocessing workflow[74], including the steps of b0 intensity normalization, and corrections of EPI distortion, eddy-current, head-motion and gradient nonlinearity. The preprocessed files and QC metrics of 636 subjects were downloaded from the Fiber Data Hub (link: https://brain.labsolver.org/hcp_d.html). The data identified as "outlier" in the QC table were excluded ($N = 15$), which were with lower neighboring diffusion weighted imaging (DWI) correlation values (0.46 - 0.70).

**HBN Study.** The dMRI data was preprocessed by QSIPrep according to[75]. Preprocessed 3 T dMRI data of 1,769 subjects were downloaded from AWS S3 (link: s3://fcp-indi/data/Projects/HBN/BIDS_curated/derivatives/qsiprep/). Because the dMRI data was collected according to 15 different acquisition parameters, we only included the subjects with the most common dMRI acquisition scheme of 'SOTE_64-dir_most_common'. Quality control (QC) procedure was from previous studies[75,76]. The subjects whose dMRI quality score ('xgb_qc_score', or 'dl_qc_score' if 'xgb_qc_score' was not available) less than 0.5 were excluded. A total of 1,102 subjects passed the quality control procedure.

**ABCD Study.** Standard dMRI preprocessing was performed by the ABCD Data Analysis, Informatics and Resource Center[77], with the following steps of eddy-current correction, head-motion correction, adjusting diffusion gradients for head motion, robust tensor fitting, correcting B0 distortion and gradient distortion using opposite phase encoding pairs of b0 images, registering b0 images to T1w images using mutual information and cubic interpolation to resample at a

1.7 mm isotropic resolution. The QC procedure included both manual and automated checks, which were described elsewhere[77]. Specifically, we excluded the images that met one of the following criteria: (1) with artifacts preventing radiology reads or with incidental findings (mrif_score = 0, mrif_score = 3 or mrif_score = 4); (2) failed raw QC or total repetitions for all OK scans was less than 103 for dMRI data (iqc_dmri_ok_ser = 0 or iqc_dmri_ok_nreps <103); (3) with dMRI B0 unwarp unavailable, with registration of dMRI to T1w larger than 17 mm, or dMRI maximum dorsal and ventral cutoff scores larger than 47 and 54 respectively (apqc_dmri_bounwarp_flag = 1, apqc_dmri_reg-t1_rigid > 17, apqc_dmri_fov_cutoff_dorsal > 47 or apqc_dmri_fov_cu-toff_ventral > 54); (4) failed manual post-processing QC for dMRI data (dmri_dti_postqc_qc = 0) and (5) failed raw QC for T1w data (iqc_-t1_ok_ser = 0). See Supplementary Fig. 15 for more details.

## Assessments of cognition and psychopathology

The behavioral dataset, as used in our previous study[35], comprised multisource data from the ABCD study assessing cognitive functioning and psychopathology across multiple domains. The dataset included 20 neurocognitive and 31 psychopathology-related assessments. The cognitive assessments covered different aspects of cognition including fluid cognition, crystalized cognition, overall cognition, processing speed, working memory, inhibitory control, episodic memory, vocabulary, reading, visuospatial ability, attention and cognitive flexibility, derived from NIH Toolbox cognition battery, Matrix Reasoning Test from the Wechsler Intelligence Scale for Children-V (WISC-V), Rey Auditory Verbal Learning Test (RAVLT), Little Man Task, Recognition memory, Emotional n-back fMRI task, Stop-signal fMRI task and Monetary Incentive Delay task. See Supplementary Table 19 for more details of the above neuro-cognitive measurements. The psychopathology-related assessments included the scales of the Child Behavioral Checklist (CBCL)[78], the personality measures from the Modified Urgency, Perseverance, Pre-meditation and Sensation-seeking (UPPS-P) for Children from PhenX[79,80], the mania scale from the 73-item Parent General Behavior Inventory (P-GBI), psychosis risk symptom subscales from Youth-report Prodromal Questionnaire Brief Version (PQ-B)[81] and the subscales assessing two motivational systems from a modified Behavioral Inhibition & Behavioral Activation Scales. See Supplementary Table 20 for details of psychopathology-related subscales.

## Assessments of clinical diagnosis and status transitions

We used parent-reported Kiddie Schedule for Affective Disorders and Schizophrenia for DSM-5 (KSADS-5)[80] to measure the current categorical psychiatric diagnoses at baseline and 2-year follow-up. Psychiatric disorders (Depressive Disorder [MDD], Psychotic Disorder [PSD], Bipolar Disorder [BPD], Attention-deficit/hyperactivity Disorder [ADHD], Substance Use Disorder [SUD], Alcohol Use Disorder [AUD], Obsessive-compulsive Disorder [OCD], Social Anxiety Disorder [SAD], Generalized Anxiety Disorder [GAD], Separation Anxiety Disorder [SED], Eating Disorder [ED], Conduct Disorder [CD], Posttraumatic Stress Disorder [PTSD], Oppositional Defiant Disorder [ODD], Specific Phobia [SP], Panic Disorder [PAD], Agoraphobia [AGO], Disruptive mood dysregulation disorder [DMDD], Enuresis and Encopresis Disorder [EED], and Tic Disorder [TD]) were used for the analyses. For each specific disorder, the score would be labeled as "0" for "absence of diagnosis" and as "1" for "definitive diagnosis".

## Diffusion MRI preprocessing

The dMRI data were reconstructed using generalized q-sampling imaging[82] with a diffusion sampling length ratio of 1.25. A deterministic fiber tracking algorithm[83] was used with augmented tracking strategies[84] to improve reproducibility. We utilized default parameters for fiber tracking[38]. Topology-informed pruning[85] was applied to tractography with 2 iteration(s) to remove false connections. The above preprocessing was performed using DSI studio (https://dsi-studio.labsolver.org [38]).

## Tract profile quantification and data harmonization

A total of 54 tracts were automatically extracted using DSI studio. These tracts included 6 pairs of bilateral dorsal association tracts (left and right Arcuate Fasciculus [AF], Frontal Aslant Tract [FAT], Parietal Aslant Tract [PAT], Superior Longitudinal Fasciculus 1 [SLF1], Superior Longitudinal Fasciculus 2 [SLF2], Superior Longitudinal Fasciculus 3 [SLF3] tracts), 4 pairs of bilateral ventral association tracts (left and right Inferior Fronto Occipital Fasciculus [IFOF], Inferior Longitudinal Fasciculus [ILF], Uncinate Fasciculus [UF] and Middle Longitudinal Fasciculus [MdLF] tracts), 6 pairs of bilateral limbic tracts (left and right Cingulum Frontal Parahippocampal [C_FPH], Cingulum Frontal Parietal [C_FP], Cingulum Parahippocampal [C_PH], Cingulum Para-hippocampal Parietal [C_PHP], Cingulum Parolfactory [C_PO] and Fornix [F] tracts), 3 pairs of bilateral sensory-motor tracts (left and right Corticospinal Tract [CST], Optic Radiation [OR] and Vertical Occipital Fasciculus [VOF] tracts), 3 pairs of bilateral thalamic radiation tracts (left and right Thalamic Radiation Anterior [TR_A], Thalamic Radiation Posterior [TR_P] and Thalamic Radiation Superior [TR_S] tracts), 3 pairs bilateral corticostriatal tracts (left and right Corticos-triatal Tract Anterior [CS_A], Corticostriatal Tract Posterior [CS_P], and Corticostriatal Tract Superior [CS_S]) and 4 corpus callosum tracts (Corpus Callosum Body [CC_Body], Corpus Callosum Forceps Major [CC_Major], Corpus Callosum Forceps Minor [CC_Minor] and Corpus Callosum Tapetum [CC_Tap]). The 54 tracts are illustrated in Fig. 1A. The GM regions mainly connected by the tracts are indicated in Supplementary Table 1. For each identified tract, 100 equal-length segments (or nodes thereafter) were sampled along the tract from one end to the other (Fig. 1B). We then computed the FA value for each node in all the tracts using a kernel density estimator with the default bandwidth setting. The 100-node FA tract-profile was generated for each tract of each participant. 611 participants in HCP-D dataset (aged 5.58 to 21.92 years; 281 females and 330 males), 978 participants in HBN dataset (aged from 5.58 to 21.90 years, 366 females), and 8688 at baseline, 5883 at 2-year- and 2531 at 4-year-follow-up in ABCD dataset (age-range = 8.91-15.70 years) had successful automatic tract extraction for all 54 WM tracts.

In order to eliminate the influence of the differences across acquisition sites, we harmonized the FA tract-profile data using a ComBat algorithm in Matlab (https://github.com/Jfortin1/ComBatHarmonization/tree/master/Matlab)[86]. In our study, KSADS-5 diagnosis, age and sex were designed as the biological variables to reserve.

## Brain age prediction models and brain age gap

**Brain age prediction models.** An overview of the brain age prediction pipeline is outlined in the Fig. 1C. We used the HCP-D dataset for training and cross-validation of brain age prediction models because it included high-quality dMRI from a moderate cohort of healthy participants ($N = 611$) covering a wide age range (5.58–21.90 years). We then applied these models to predict brain age in the independent ABCD and HBN datasets. The 25 pairs of projection/association tracts (each pair of bilateral tracts were concatenated) and 4 corpus callosum tracts were used to build "tract-specific" models respectively, while all 54 tracts were concatenated to build a "whole-brain" model. Using a 5-fold cross-validation on 80% subjects from HCP-D dataset, a Gaussian Process Regression (GPR) model[28,30] (https://github.com/garedaba/brainAges) in Python was trained to predict individual age based on the whole-brain tract-profile or 29 tract-specific profiles respectively. The remaining 20% HCP-D subjects were used to validate the brain age model performance in a 5-fold cross-validation procedure.

The trained brain age prediction models from the HCP-D cohort were then used to predict brain ages of the subjects at baseline and 2-year-follow-up in the ABCD dataset and the subjects in the HBN dataset. Model accuracy was evaluated in the cross-validation dataset and the independent testing dataset using the $R$ (Pearson correlation between chronological age and predicted age) and the RMSE (root mean squared error). To assess whether tract-specific brain-age prediction exceeded chance, we generated an empirical null distribution by randomly permuting chronological ages across participants and re-establishing the prediction model on each permuted dataset. This procedure was repeated 1000 times, yielding a null distribution of R values under the assumption that there is no predictive utility of WM features on chronological age prediction. The $p$ value is determined as the proportion of permutation-based R values derived greater than or equal to the observed R. Values of $p < 0.05$ were considered as the predictive performance significantly exceeded chance. Multiple comparisons across tracts ($n = 30$) were implemented using the Benjamini-Hochberg false-discovery-rate (FDR) procedure.

**Brain age gap calculation and brain-age bias correction.** For each subject, brain age gap (BAG) was calculated as the difference between predicted brain age and chronological age. Age-bias correction was conducted by the method proposed by Beheshti et al.[32]. In the training dataset, we performed linear regression between chronological age and BAG, as follows:

$$BAG = \alpha Age_{Chronological} + \beta \tag{1}$$

The derived slope ($\alpha$) and intercept ($\beta$) were applied to the chronological ages of the individuals in the cross-validation and the independent testing datasets. We then subtracted the calculated values from the predicted BAGs to correct for age bias, as follows:

$$BAG_{Correced} = BAG_{Uncorrected} - (\alpha Age_{Chronological} + \beta) \tag{2}$$

To avoid potential prediction accuracy overestimates[34], we only reported the prediction performance by uncorrected brain ages. Tract-based BAGs without age-bias correction were used for testing the robustness of our results described in the primary statistical analysis section.

**Biological interpretation of BAGs through developmental bench markers**

We further included pubertal stage scales and salivary hormone levels as developmental bench markers. Pubertal maturation of the ABCD subjects was measured by the Pubertal Development Scale (PDS)[87]. We utilized the parent-reported PDS as it is likely more reliable than the self-reported version for the age range of our study participants and widely used in other ABCD studies[88]. Because the ABCD 4-year follow-up data are only partially released, analyses in this section focused on the baseline and 2-year follow-up data. Of 8688 participants at baseline and 5833 participants at 2-year follow-up, pubertal development data assessed by the Pubertal Development Scale (PDS) were available for 8263 baseline participants and 4532 follow-up participants. The PDS provides puberty category measures ranging from prepubertal to post-pubertal. For male participants, the puberty category score was calculated by summing items related to voice deepening, body hair growth, and facial hair growth. These scores were then categorized as follows[89]: prepubertal (= 3), early pubertal (4 or 5), mid pubertal ($\geq 6$ and $\leq 8$), late puberty ($\geq 9$ and $\leq 11$); postpubertal (= 12). For female participants, the puberty category score was derived by summing items on body hair growth and breast development, with categorization as follows: prepubertal (= 2), early pubertal (= 3 and no menarche), mid pubertal (> 3 and no menarche), late puberty ($\leq 7$ and menarche); postpubertal (= 8 and menarche). See Supplementary

Fig. 5 for distributions of sex-specific pubertal stage distributions across two timepoints. Linear Mixed Effects Models (LMMs) were conducted to explore the relationship between pubertal stage and whole-brain BAGs with age-bias correction:

$$BAG = \beta_0 + \beta_1 Hormone + \beta_2 Age + \beta_3 Sex + \beta_4 Timepoint + (1|Subject) \tag{3}$$

Salivary hormone concentrations of DHEA, testosterone, and estradiol were assayed by Salimetrics (https://salimetrics.com), with samples collected between 7:00 am and 7:00 pm. Hormone data were obtained from the ABCD variables of hormone_scr_dhea_mean (DHEA), hormone_scr_ert_mean (testosterone), and hormone_scr_hse_mean (estradiol) in table phy_y_sal_horm. Data cleaning and quality control followed the previous published studies[90,91]: i) participants with mismatched biological sex (reported at saliva collection versus biological sex) were excluded; ii) hormone levels outside biologically plausible reference ranges (DHEA: 5 ~ 1000 pg/ml; Testosterone: 5 ~ 500 pg/ml; Estradiol: 0 ~ 1500 pg/ml) were removed; iii) factors of age, sex, visit timepoint, whether have caffeine intake and physical activity in the past 12 hours, saliva collection time since midnight (minutes), saliva collection duration (minutes), and time from collection to freezer storage (minutes) were included as the confounding effects in all analyses. After QC, hormone assessments were available for 10,302 participants for DHEA ($N_{Baseline} = 8266$, $N_{2-y-follow-up} = 2036$), 10,684 participants for testosterone ($N_{Baseline} = 8323$, $N_{2-y-follow-up} = 2361$), and 6413 participants for estradiol ($N_{Baseline} = 3944$, $N_{2-y-follow-up} = 2469$). Statistical analyses were performed using LMMs:

$$\begin{aligned} BAG = \beta_0 &+ \beta_1 Hormone + \beta_2 Age + \beta_3 Sex + \beta_4 Timepoint \\ &+ \beta_5 CaffineIntake + \beta_6 PhysicalActivity \\ &+ \beta_7 CollectionTime + \beta_8 CollectionDuration \\ &+ \beta_9 FreezeTime + (1|Subject) \end{aligned} \tag{4}$$

**Statistical analysis for clinical applications**

**Associations between tract BAGs and behavioral assessments.** To explore the multivariate associations between the tract-based BAGs with age-bias correction and the multidomain behavioral assessments, we performed sparse canonical correlation analysis (sCCA) between Z-scored brain and behavioral measurements of the ABCD baseline datasets (https://github.com/cedricx/sCCA/tree/master/sCCA/code/final)[92]. Prior to sCCA analysis, we regressed age and sex out of both the brain and behavioral measures. sCCA identifies multiple latent variables that show maximal correlations between the brain and behavior datasets, with regularization to handle multicollinearity issues and achieve sparsity.

$$arg\max_{u,v} u^T X^T Y v, subject\ to\ ||u||_2^2 = 1, ||v||_2^2 = 1, ||u||_1 = C_1 and ||v||_2 = C_2 \tag{5}$$

Here, X and Y are the multi-dimensional tract-based BAG and behavioral datasets, respectively. In addition, u and v are the coefficients used to project the datasets to latent variables. The regularization parameters include the sparsity of $\boldsymbol{u}$ and $\boldsymbol{v}$, which were determined by the following grid-search and nested cross-validation procedure within the training dataset. Both parameters were incremented by 0.1 from 0 to 1, resulting 100 combinations of different $C_1$ and $C_2$ values. Subsequently, we randomly resampled two-thirds of the subjects for ten times. Through two-thirds of the sample, the coefficients of the sCCA model with $C_1$ and $C_2$ were estimated. The combination of the regularization parameters with the largest canonical correlation coefficient was used for the final model. Furthermore, we performed 1000 times of permutation tests to assess the statistical

significance of canonical correlation modes, with shuffling the behavioral set order during each iteration. To determine the significance of each behavioral measure within each mode, we used a bootstrap resampling procedure, repeated 1000 times. Behavioral and brain measures for each sCCA mode were deemed significant if their 95% confidence intervals did not include zero.

Additionally, to assess the specificity of the sCCA modes, we performed paired t-tests to compare the behavioral loading distributions derived from the bootstrap tests. A behavioral measure was considered specific to an sCCA mode if its effect size (Cohen's d value) was more than 0.5.

In addition, a post-hoc analysis investigating the relationships between tract BAGs and each cognitive or clinical score was conducted by generalized linear model (GLM) with the covariates of sex and age as follows:

$$BAG = \beta_0 + \beta_1 Behavior + \beta_2 Age + \beta_3 Sex \tag{6}$$

All tested associations (30 × 51 tests) were corrected for multiple comparisons by using the FDR method. Corrected p and t values are shown in Supplementary Figs. 7–8.

**Associations between tract-BAG measures and clinical diagnoses at baseline.** To assess whether inter-individual differences in the tract-BAG measures capture clinically relevant differences in psychiatric diagnoses, we characterized associations of these BAGs with the cumulative number of KSADS-5 diagnoses in the ABCD dataset. We categorized the participants into three subgroups based on their cumulative number of their KSADS-5 diagnoses at baseline. Participants in the first subgroup were with no KSADS-5 diagnosis ($N = 6466$), in the second subgroup with a single KSADS-5 diagnosis ($N = 1337$), and in the third subgroup with at least two KSADS-5 diagnosis ($N = 791$). Tract-BAG measures (including sCCA variate scores and individual tract BAGs) versus cumulative number of KSADS-5 diagnoses were assessed using general linear models (GLMs) controlled for age and sex, as follows:

$$BAG_{Baseline} = \beta_0 + \beta_1 KSADS_{Baseline} + \beta_2 Age + \beta_3 Sex \tag{7}$$

where categorical variable $KSADS_{Baseline}$ is the cumulative number of KSADS-5 diagnoses (0, 1, or ≥ 2) at baseline. We then performed Tukey tests for post-hoc comparisons. The GLM analysis was conducted using 'glm' in R.

We also conducted a validation analysis using GLMs to evaluate the associations between tract-BAG measures and transdiagnostic status in participants with available 2-year follow-up data. These participants, who had both brain and clinical measures available at the 2-year follow-up, were categorized into three groups: those without a KSADS-5 diagnosis ($N = 4131$), those with one KSADS-5 diagnosis ($N = 741$), and those with at least two KSADS-5 diagnoses ($N = 336$).

**Associations between tract-BAG measures at baseline and clinical diagnoses at 2-year follow-up.** To explore the predictive potential of tract-BAG measures for clinical diagnoses of psychiatric disorders at 2-year-followup, we used GLMs to assess associations between the tract-BAG measures at baseline and cumulative number of KSADS-5 diagnoses at 2-year-followup, as follows:

$$BAG_{Baseline} = \beta_0 + \beta_1 KSADS_{followup} + \beta_2 Age + \beta_3 Sex \tag{8}$$

where variable $KSADS_{followup}$ is the cumulative number of KSADS-5 diagnoses at 2-year follow-up. The participants were categorized into three subgroups according: those without KSADS-5 diagnosis ($N = 6288$), those with a single KSADS-5 diagnosis ($N = 1116$) and those with at least two KSADS-5 diagnoses ($N = 503$).

**Associations between tract-BAG measures at baseline and status transitions in KSADS-5 diagnoses from baseline to 2-year follow-up.** We investigated whether tract-BAG measures associated with diagnostic status transitions. Based on whether the participants had psychiatric diagnoses at baseline and/or 2-year follow-up, they were categorized into 4 subgroups: Healthy-persistent (0 psychiatric diagnosis at both baseline and 2-year follow-up, $N_{HH} = 5,236$), Disorder-remitted (≥1 psychiatric diagnosis at baseline and 0 psychiatric diagnosis at 2-year follow-up, $N_{PH} = 995$), Disorder-new-onset (0 psychiatric diagnosis at baseline and ≥1 psychiatric diagnosis at 2-year follow-up, $N_{HP} = 751$) and Disorder-persistent (≥1 psychiatric diagnosis at both baseline and 2-year follow-up, $N_{PP} = 848$). GLMs were conducted to assess the association between tract BAGs versus the diagnostic transition status of psychiatric diagnosis, as follows:

$$BAG_{Baseline} = \beta_0 + \beta_1 KSADS_{Transition} + \beta_2 Age + \beta_3 Sex \tag{9}$$

**Associations between tract-BAG measures at baseline and cognitive performance at 2-year or 3-year follow-up.** To evaluate longitudinal cognitive outcomes, we examined the associations between baseline tract-based BAG measures and follow-up cognitive performance not included in the prior sCCA analysis. Three general cognitive assessments from follow-up visits were used: Emotional Stroop Task (3-Year Follow-Up)[93]: This task assessed processing speed/attention, inhibitory control, and emotional information processing. We used the overall accuracy score as the measure. Self-Reported Academic Grade (SAG; 2-Year Follow-Up)[39]: School performance was evaluated using self-reported grades, ranging from 1 (A +) to 12 (failing), with lower scores indicating better academic performance. Stanford Mental Arithmetic Response Time Evaluation (SMARTE)[39]: This measure of math ability included dot enumeration, single-digit, and mixed-digit arithmetic problems. The total number of correct responses was used as the performance measure.

To assess the relationship between baseline tract-based BAGs and each follow-up cognitive measure, we employed generalized additive models (GAMs), which are widely used in large-scale datasets[94]. Age and sex were included as covariates in all models. The unique contribution of each tract-based BAG measure was quantified by computing the change in model variance (adjusted $\Delta R^2$, expressed as a percentage) when the BAG predictor was added to a null model containing covariates only. Both effect sizes and two-sided $p$-values with FDR correction were reported. To assess whether tract-specific BAGs account for more variance in follow-up cognitive performance than raw FA values, we conducted the same statistical analyses using tract-wise FA values, averaged from the harmonized tract-wise FA values.

**Sensitivity analysis on effects of pubertal maturation.** Pubertal maturation of the ABCD subjects was measured by the Pubertal Development Scale (PDS)[87]. We utilized the parent-reported PDS as it is likely more reliable than the self-reported version for the age range of our study participants and widely used in other ABCD studies[88]. Of 8,681 baseline participants, 8,263 had pubertal development data assessed by the Pubertal Development Scale (PDS), which included puberty category measures ranging from prepubertal to postpubertal. For male participants, the puberty category score was calculated by summing items related to voice deepening, body hair growth, and facial hair growth. These scores were then categorized as follows[89]: prepubertal (= 3), early pubertal (4 or 5), mid pubertal (≥ 6 and ≤ 8), late puberty (≥ 9 and ≤ 11); postpubertal (= 12). For female participants, the puberty category score was derived by summing items on body hair growth and breast development, with categorization as follows: prepubertal (= 2), early pubertal (= 3 and no menarche), mid pubertal (> 3 and no menarche), late puberty (≤ 7 and menarche); postpubertal (= 8 and menarche). To assess the impact of

pubertal maturation on our results, we included the puberty category as an additional covariate in the GLM analysis described above.

## Functional decoding based on spatial association between brain regions connected by WM tracts and task-related brain activation

To validate the functional significance of the tract-based BAGs of the sCCA modes, we assessed spatial correlation between brain regions connected by these tracts and specific task-related brain activation patterns. For task-related brain activation maps, we used data from the latest NeuroSynth database (version 7)[36], a meta-analytic tool that aggregates findings from 14,371 published fMRI studies. We selected 31 topics (See Supplementary Table 6) related to cognitive-control, language, emotion, or psychiatric conditions from a set of 100 topics generated using Latent Dirichlet Allocation (LDA). These topic maps were generated through reverse inference, denoting the probability of a topic being relevant given a particular brain activation. For the white matter tracts, we utilized the Population-Probability Atlas[38], which provides voxel probability maps for 25 association and projection WM tracts included in our sCCA study. To refine this atlas, we excluded voxels with a probability lower than 0.5 and those with less than 50% probability according to the GM probability map (MNI-maxprob-thr50-2mm.nii.gz in FSL). We calculated the average probability of topic activation levels within the generated tract regions of interest (ROIs). To evaluate their spatial correlation, we conducted Spearman correlation analyses between the loadings of two sCCA modes and the z-scores from the brain activation meta-analytic maps for each topic. Statistical significance was evaluated using a spin-based spatial permutation test (1000 iterations; https://github.com/murraylab/brainsmash/tree/master)[95], which preserves the spatial autocorrelation structure of GM maps. During each iteration, the voxel-wise task activation map was spatially rotated and the brain activation levels of the GM regions connected by each tract were recomputed. The empirical p-value was defined as the proportion of permuted Spearman correlation coefficients that exceeded the observed correlation and corrected by the FDR method.

## Mitochondrial mapping analyses

We based our analysis on whole-brain maps of mitochondrial properties developed by Mosharov et al. [27]. Six mitochondrial maps (CI, CII, CIV, MitoD, TRC and MRC) were included in our study. For each tract, we averaged the mitochondrial values within the voxels whose probability higher than 50% in Population-Probability Atlas[38] in MNI152 volumetric space. The tract-wise mitochondrial profiles were shown in Fig. 4A. Spatial correlation between sCCA loading pattern and mitochondrial profiles were performed by Spearman correlation. Statistical significance was evaluated using a spin-based spatial permutation test (10,000 iterations; https://github.com/frantisekvasa/rotate_parcellation)[96]. We then estimated the number of null correlation coefficients that were significantly higher than expected by chance as p-value and corrected by the FDR method.

## Reporting summary

Further information on research design is available in the Nature Portfolio Reporting Summary linked to this article.

## Data availability

The HCP-D data used in this study are available at (https://www.humanconnectome.org/study/hcp-lifespan-development/) with data access. Preprocessed SRC files for the HCP-D dataset can be downloaded from the Fiber Data Hub (https://brain.labsolver.org/hcp_d.html) with HCP-D data access. The ABCD data used in this study are available through the National Institute of Mental Health Data Archive (NDA; https://abcdstudy.org) upon formal application and approval by the ABCD consortium. preprocessed dMRI data from the HBN study

are available at: s3://fcp-indi/data/Projects/HBN/BIDS_curated/derivatives/qsiprep/[75]. The Population-Probability Atlas of 25 tracts used in this study is available at (https://brain.labsolver.org/hcp_trk_atlas.html). Task-related brain activation maps used in this study were obtained from the NeuroSynth database (version 7; https://github.com/neurosynth/neurosynth-data). Whole-brain mitochondrial property maps are available through NeuroVault (https://neurovault.org/collections/16418/)[27]. Source data are provided with this paper.

## Code availability

This study used openly available codes and software, specifically DSI studio (Chen-2022-12-22; (https://hub.docker.com/r/dsistudio/dsistudio/), R (4.5.1), MATLAB (R2024a), Python (3.10.10) and Connectome Workbench (v2.1.0). The pipeline for data harmonization is available at (https://github.com/Jfortin1/ComBatHarmonization/tree/master/Matlab/). The pipeline for building brain-age models is available on GitHub (https://github.com/garedaba/brainAges). The GM voxel-wise spin-based spatial permutation procedure is available at (https://github.com/murraylab/brainsmash/tree/master/). The WM tract-wise spin-based spatial permutation pipeline is available at (https://github.com/frantisekvasa/rotate_parcellation/).

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

## Acknowledgements

This work was supported by the Intramural Research Program of the National Institute on Drug Abuse, National Institutes of Health (NIH) and utilized the computational resources of the NIH HPC Biowulf cluster (https://hpc.nih.gov). The contributions of the NIH authors were made as part of their official duties as NIH federal employees, are in compliance with agency policy requirements, and are considered Works of the United States Government. However, the findings and conclusions presented in this paper are those of the author(s) and do not necessarily reflect the views of the NIH or the U.S. Department of Health and Human Services. The authors used data from the Adolescent Brain Cognitive Development Study (ABCD, abcdstudy.org), the Healthy Brain Networks (HBN, data.healthybrainnetwork.org) and the Lifespan Human Connectome Project Development Study (HCP-D, https://www.humanconnectome.org/study/hcp-lifespan-development). The ABCD Study, held in the National Institute of Mental Health (NIMH) Data Archive (NDA), is a multisite, longitudinal study designed to recruit more than

10,000 children aged 9-10 and follow them over 10 years into early adulthood. The ABCD Study is supported by the NIH and additional federal partners under award numbers U01DA041048, U01DA050989, U01DA051016, U01DA041022, U01DA051018, U01DA051037, U01DA050987, U01DA04 1174, U01DA041106, U01DA041117, U01DA041028, U01DA041134, U01DA050988, U01DA051039, U01DA041156, U01DA041025, U01DA 041120, U01DA051038, U01DA041148, U01DA041093, U01DA041089, U24DA041123, and U24DA041147. A full list of supporters is available at https://abcdstudy.org/federal-partners.html. A listing of participating sites and a complete listing of the study investigators can be found at https://abcdstudy.org/consortium_members/. HCP-D was supported by the NIMH of the NIH under Award Number U01MH109589 and by funds provided by the McDonnell Center for Systems Neuroscience at Washington University in St. Louis. HCP-D, HBN and ABCD consortium investigators designed and implemented the study and/or provided data but did not necessarily participate in analysis or the writing of this report. This manuscript reflects the views of the authors and may not reflect the opinions or views of any other agency, organization, employer or company.

## Author contributions

D.W., Y.Y., C.J.H., B.J.S., A.J. and T.J.R. concepted and designed the study. D.W. analyzed the data under the guidance of X.X., T.J.R. and Y.Y. D.W., C.J.H., B.J.S., X.X., A.J., L.M., T.J.R. and Y.Y. drafted the manuscript. D.W., C.J.H., B.J.S., X.X., L.M., H.G., T.Z., A.Q., J.H., H.N., H.L., A.J., T.J.R. and Y.Y. contributed to the interpretation of the results. All authors provided analytical support and contributed to the final manuscript. Y.Y. supervised the overall work.

## Funding

## Competing interests

The authors declare that they have no known competing financial interests or personal relationships that could have appeared to influence the work reported in this paper.
