## [Transparent Peer Review file · Nature Communications]

Deviation in Development of Dorsal Association Tracts during Preadolescence Links to Concurrent and Future Cognitive Performance and Transdiagnostic Psychopathology

Corresponding Author: Dr Yihong Yang

Version 0:

Reviewer comments:

Reviewer #1

(Remarks to the Author)

General Assessment

This manuscript presents a rigorous and ambitious analysis of white matter brain-age modelling in youth, using three large-scale neuroimaging cohorts (ABCD, HCP-D, and HBN). The authors build tract-specific diffusion-based brain age prediction models and link resulting brain age gaps (BAGs) to both cognitive performance and transdiagnostic psychopathology, with supporting analyses. The study is well-written and addresses important questions in developmental neuroscience and psychiatry using state-of-the-art computational approaches. However, several conceptual and methodological issues, particularly regarding the interpretation of brain age gap (BAG) in youth, the implications of age-bias correction, and the handling of scanner/site effects, require clarification. These issues affect how the findings can be interpreted and generalised, and warrant revision before the manuscript is suitable for publication.

Major Comments

1. Interpretation and Conceptual Clarity of Brain Age Gap (BAG)

The manuscript treats BAG as a proxy for "maturation" or "delay," yet this assumes a linear and biologically grounded trajectory of white matter development. In reality, white matter volume and FA follow curvilinear trajectories across development, typically increasing between ages 9–14 before plateauing or decreasing later. Interpreting positive BAGs as "more mature" and negative ones as "delayed" is conceptually problematic, particularly when such deviations may reflect normal individual variation or artefacts of modeling rather than meaningful maturational differences. The current framing risks conflating statistical deviation with biological interpretation.

2. Handling and Interpretation of Age-Bias Correction

While the authors aim for transparency by including both corrected and uncorrected BAGs, the use of both in parallel without clear analytic separation introduces interpretive ambiguity. The combination of age-bias correction and inclusion of age as a covariate may result in compounded adjustments that are difficult to interpret, and the implications of this approach are not clearly addressed.

3. Over-interpretation of Predictive Findings

The manuscript repeatedly uses predictive language (e.g., BAGs "predicted" later outcomes) based on associations in observational data. While some associations are longitudinal, the analyses do not constitute true prediction or causal inference. This wording overstates the findings and could mislead readers about the strength of evidence.

4. Cross-Dataset Generalisability and Domain Shift

The authors train models in one dataset and apply them to others without detailed evaluation of generalisability. Domain shifts between cohorts (e.g., due to scanner differences or population composition) may distort predictions. Without clearly presenting out-of-sample performance or calibration by cohort, it is difficult to assess the stability and transportability of the models.

5. Harmonisation

Diffusion metrics like FA are sensitive to scanner and acquisition parameters, yet the manuscript does not describe any harmonisation approach (e.g. longCombat). This raises concerns about site-related confounding and biases in BAG estimates across cohorts.

6. Interpretation of BAG Directionality

While the authors include sensitivity analyses controlling for pubertal status and SES, the biological interpretation of BAG directionality remains speculative. The assumption that smaller BAGs reflect developmental “delay” or risk is not validated through direct associations with developmental benchmarks such as age, pubertal stage, or sex-specific trajectories. It remains unclear how BAG varies across these axes or aligns with known nonlinear patterns of white matter maturation.

7. Functional and Mitochondrial Decoding Over-interpretation

The use of a single adult brain for mitochondrial mapping limits generalisability, particularly in a paediatric sample. Despite acknowledging this limitation in the discussion, the main results treat the interpretation too strongly given the mismatch in age and sample source.

8. Terminology Around “Maturation”

The manuscript frequently equates positive BAG with “advanced maturation” and negative BAG with “delay.” This language presumes a normative linear developmental trajectory and may not reflect true neurobiological differences. It also risks misinterpretation by clinicians or translational researchers.

9. Literature overview and “filling gaps”

There is a lack of engagement with earlier dMRI brain age literature. The study would benefit from highlighting these, especially studies in youth that may have investigated cognition, psychopathology, similar.

Minor Comments

1. RMSE and Pearson’s R are not reported, though they are standard metrics for model evaluation. Their absence makes it harder to contextualise the predictive performance.
2. The sentence “...are poorly understand” should be “poorly understood”.
3. Some references have gaps between the word before it (e.g. reference 1) while others do not.
4. The label “Association Development mode” used to describe sCCA components implies a specificity that may not be warranted by the data.
5. The manuscript briefly mentions differences between GM- and WM-derived BAGs but does not integrate this contrast meaningfully into the main interpretation.
6. Figures do not consistently indicate whether the BAG values shown are corrected or uncorrected.
7. The choice of Gaussian Process Regression (GPR) is not justified against alternative modeling approaches.

Reviewer #2

(Remarks to the Author)

The manuscript by Wang et al assesses whether adolescents’ white matter brain-age gap (BAG) relates to cognition or psychopathology. Here, BAG is the difference in true ages and ages predicted by a machine learning model trained on white-matter tracts’ fractional anisotropy. The model is trained in HCPD and applied to primarily ABCD (with further validation in HBN). The authors also characterize the covariance structure of tract-wise BAG and a variety of cognitive and behavioral traits as assessed in the ABCD sample using sparse canonical correlation analysis (sCCA), describing two primary modes of covariance - Association and Subcortical/limbic development modes. Finally, they test whether the spatial patterning of these modes correlated with post-mortem patterns of metabolic markers and if individual BAGs across these modes, across all tracts, and/or within specific brain tracts are associated with cognitive metrics or diagnostic status. Overall, the paper is expansive and intended to address the very relevant issue of how non-typical white matter development in various tracts may be associated with cognitive function and psychopathology. Strengths include the use of 3 independent datasets and the inclusion of sensitivity analyses for pubertal stage and the effects of age-correction. However there are several major areas of concern relating to clarity, possible circularity, and other methodological issues. These concerns significantly limit the potential impact of the paper especially relative to the many other BAG papers published in the last several years with overlapping samples and research questions. We hope the following specific comments are helpful.

1. It is not adequately justified why BAG model is fitted in HCPD (smaller sample, bigger age range) and then tested in ABCD. This is especially important given that the out of sample r^2 of the age predictions in ABCD is quite low (0.185) raising significant concerns about the interpretability of the downstream BAG analyses.
2. There is concern that several of the analyses presented are redundant or circular. For instance, the two sCCA modes were defined based on BAG’s covariance with cognitive and behavioral measures. This makes their subsequent tests of sCCA’s spatial covariance with neurosynth terms (Figure 3) and sCCA variants’ associations with KSADS and cognitive variables (which are almost certainly highly covariant with the cognitive and behavioral variables used to define the modes) problematic.
3. Similarly, repeating associations between cognitive/clinical markers and BAG at the level of sCCA mode, whole-brain, tract, and network are presented without clarity as to the interdependence of these tests. Greater justification, clearer interpretation, or clarity as to which analyses are considered supplementary would likely be helpful.

4. It is not clear why the BAG tends to be the outcome variable in GLMs when the authors are interested in how much BAG contributes to current/future psychiatric/cognitive symptoms. Why aren't the models set up so that behavior/cognition are the outcome variables and BAGs are the predictor variables (while controlling for demographic covariates)?

Reviewer #3

(Remarks to the Author)

Overview: This study developed a model of brain age using diffusion-weighted imaging data from the HCP development dataset and validated it in the ABCD and Healthy Brain Network (HBN) cohorts with modest effect sizes for their validation. To determine clinical relevance, they identify associations between derived brain age estimates and cognition and clinical measures in ABCD at baseline and at 2-year follow-up. Finally, they use postmortem data to link finding to mitochondrial WM profiles. This is a lot of work and the overall result is a relatively cohesive characterization of how individual differences in brain-age derived estimates of WM may link to clinical and cognitive features during preadolescence. However, there are some methodological and 'big picture' concerns, as described below:

Big picture: In recent years a number of studies have sought to build predictive models of chronological age using structural MRI data as input. Larger gaps between brain age and actual age, i.e., the 'brain age gap' (BAG) are often interpreted as indicative of poorer brain health or to reflect developmental 'disruptions'. However, many different factors may influence brain age predictions – for example, a single dose of ibuprofen can reduce brain age estimates generated from grey matter morphometry data by 1 year <https://pmc.ncbi.nlm.nih.gov/articles/PMC6510235/> - making interpretation of findings difficult. Nonetheless, here the authors expand on the brain age literature by estimating this using diffusion tensor imaging data, specifically fractional anisotropy (FA), in a large sample of youth from the ABCD study. While the concept of studying brain age within the context of white matter maturation in a developmental cohort is thus somewhat novel, it is highly likely that brain age estimates derived from white matter data are also influenced by a large number of factors, including, but not limited to, those that are well-recognized confounds in diffusion-weighted imaging and in the use of simplistic tensor derived measures such as FA <https://pubmed.ncbi.nlm.nih.gov/22846632/>. Thus, one primary concern with this study is the concept of brain age itself, including as indexed using FA. For this reason, use of terms such as 'delayed maturation' (as is done throughout the discussion) to summarize findings should be used with caution.

Additional comments:

As far as I can tell, brain age predictions in ABCD were applied to both baseline and year 2 follow-up data in a single model (only a single r^2 value is provided). However, given the sample size of each time point, there must have been overlap between the baseline and year two samples. Brain age estimates from the same interval calculated two years apart are presumably not independent. How was this addressed?

The model's performance in ABCD is modest, particularly when done without age-bias correction. This is glossed over and presented in a supplemental table, with the in-text summary simply stating that all tracts were significant, however procedures for significance testing (e.g., permutation testing, multiple comparisons correction?) are not included in the methods (presumably this should be included around lines 813-814).

To test clinical relevance, they identify associations between derived brain age estimates and cognition and clinical measures using sparse CCA in ABCD at baseline. They identify two components/modes – an Association mode and a Subcortical/Limbic mode. Both map onto various cognitive and clinical measures at baselines, as indicated by the CCA. The authors then test whether these modes predict academic performance, emotional Stroop performance and math ability at 2 and 3 year follow-up. However, it is unclear if this is driven by associations between the baseline behavioral variables, by the brain age estimates, or both. Finally, the authors use SVR to predict academic performance, emotional Stroop performance and math ability at 2 and 3 year follow-up specifically using the tract-based BAGs alone (i.e., not the CCA modes) and find some modest predictive power. Overall, this collection of analyses is somewhat unsatisfying and it is not clear if, for example, simple individual participant FA values for each tract, rather than BAGs, would do equally well (or better) in predicting these measures. The same concern applies to the prediction of KSADS diagnoses (i.e., are BAGs better predictors than 'raw' FA values)?

Minor: please define all acronyms in figure legends (not just in Table S1)

Reviewer #4

(Remarks to the Author)

Version 1:

Reviewer comments:

Reviewer #1

(Remarks to the Author)

Thank you to the authors for their effort in addressing my comments. I have no further concerns for the manuscript and wish the authors the best of luck going forward.

Reviewer #2

(Remarks to the Author)

This is a substantive revision with detailed responses to each of our points. The responses to points 1,3,4, and 5 are very good and we no longer have concerns about these points. While the paper is thus significantly improved, the response to point 2 does not fully address our concerns about the methodological approach, in that it remains unclear whether the experiments justify some of the inferences made by the authors. This concern limits the impact of the paper and is not something that we believe can be addressed on further review. We also agree with Reviewer 3 that the manuscript is only somewhat novel, and it is hard to differentiate its added value from other brain age gap papers.

Reviewer #3

(Remarks to the Author)

The authors have been responsive to the reviewer comments and the submission now has greater clarity.

Reviewer #4

(Remarks to the Author)

Response to Reviewers:

Content

Response to all Reviewers	Page 1
Response to Reviewer #1	Page 3
Response to Reviewer #2	Page 25
Response to Reviewer #3	Page 35

Response to all Reviewers:

We thank all the reviewers for their constructive and thoughtful feedback, which we have integrated into this revised manuscript. We believe that the revisions made based on your feedback have greatly strengthened the manuscript and improved its clarity and readability. Below we summarize the major revisions:

- In response to reviewer feedback, we have reframed how we describe brain age gap (BAG) to emphasize that tract-based BAG reflects statistical deviation from age-normative white-matter trajectories, rather than a direct biological measure of accelerated or delayed maturation. We now provide additional validation analyses demonstrating that higher whole-brain BAG is associated with more advanced pubertal stages and higher pubertal hormone levels, supporting its neurodevelopmental relevance.
- To address concerns regarding overlapping samples and age-range effects on prediction, we incorporated newly and partially released ABCD 4-year follow-up dMRI data (Annual Curated Release 5.1). Using non-overlapping samples and matched sample sizes, we observed consistent and improved predictive performance, particularly in testing datasets with wider age ranges.
- In addition to the full-behavior-derived model, we conducted supplementary analyses rerunning our brain-behavior sparse canonical correlation analyses (sCCA) using only cognitive or only psychopathological measures. Results from these supplemental analyses were similar to our original findings. Across all three cases (cognition-, psychopathology-, and full-behavior-derived), we observed highly consistent tract-/behavioral- loading patterns and downstream associations with cognitive/psychopathological measures, highlighting the robustness of the identified BAG-behavior relationships.
- We added supplementary analyses comparing the explained variance of tract-based BAG with tract-level fractional anisotropy (FA) values for cognitive/clinical measures. Although both contained meaningful variance, BAG explained additional variance in cognitive and clinical measures beyond FA in multiple tracts, suggesting that BAG captures complementary developmental information not fully represented by raw FA.

Below we provide detailed responses to each reviewer comment. We emphasize all reviewer comments with **gray-colored** text and provide individual responses below in black font. For

convenience, we incorporated resulting changes to the manuscript directly in the responses in quotation marks. Major changes are also highlighted with light green color in our revised manuscript.

Reviewer #1

This manuscript presents a rigorous and ambitious analysis of white matter brain-age modelling in youth, using three large-scale neuroimaging cohorts (ABCD, HCP-D, and HBN). The authors build tract-specific diffusion-based brain age prediction models and link resulting brain age gaps (BAGs) to both cognitive performance and transdiagnostic psychopathology, with supporting analyses. The study is well-written and addresses important questions in developmental neuroscience and psychiatry using state-of-the-art computational approaches. However, several conceptual and methodological issues, particularly regarding the interpretation of brain age gap (BAG) in youth, the implications of age-bias correction, and the handling of scanner/site effects, require clarification. These issues affect how the findings can be interpreted and generalised, and warrant revision before the manuscript is suitable for publication.

We appreciate the reviewer for the positive evaluation of our work and for the thoughtful and constructive feedback. The reviewer's comments (see below) are very helpful that lead to clarification of our manuscript, particularly in conceptual and methodological issues. We also performed additional analyses based on suggestions of the reviewer to strengthen the manuscript with more comprehensive information about the study. We believe that our manuscript has been significantly improved with these modifications.

- We have revised the interpretation of brain age gap (BAG) to clarify that BAG reflects the degree of deviation of an individual's tract-based microstructural profile from age-normative trajectory, rather than a biological measure of accelerated or delayed maturation.
- To validate the neurodevelopmental underpinnings of the BAG in youth, we now add the association analysis between whole-brain BAG and developmental benchmarks (pubertal stages and hormone levels). We observed that significantly positive correlations between whole-brain BAGs and more advanced pubertal stages, as well as higher pubertal hormone levels.
- We now make more clarifications for the scanner/site effects of our utilized diffusion MRI (dMRI) features and data harmonization. We also make more clear clarifications of the results with or without age-bias correction.

Reviewer #1, Comment 1:

1. Interpretation and Conceptual Clarity of Brain Age Gap (BAG)

The manuscript treats BAG as a proxy for "maturation" or "delay," yet this assumes a linear and biologically grounded trajectory of white matter development. In reality, white matter volume and FA follow curvilinear trajectories across development, typically increasing between ages 9–14 before plateauing or decreasing later. Interpreting positive BAGs as "more mature" and negative ones as "delayed" is conceptually problematic, particularly when such deviations may reflect normal individual variation or artefacts of modeling rather than meaningful maturational differences. The current framing risks conflating statistical deviation with biological interpretation.

Response:

We appreciate the reviewer’s insightful comment. We agree that the terminology like “delayed maturation” might risk conflating statistical deviation with biological interpretation, particularly given the potential influence of modeling artefacts. To address this concern, we have revised the manuscript to clarify that tract-based BAG reflects the degree of deviation of an individual’s tract-based microstructural profile from age-normative trajectory, rather than a definitive measure of accelerated or delayed maturation. We have also added supplementary association analyses linking BAGs to developmental benchmarks (pubertal stages and hormone levels) to validate its underpinning neurodevelopmental meanings (See our responses to Reviewer #1, Comment 6 for more details).

We updated the manuscript accordingly:

Title:

“Deviation in Development of Dorsal Association Tracts during Preadolescence Links to Concurrent and Future Cognitive Performance and Transdiagnostic Psychopathology”

Abstract:

Page 2, line 41-42:

“However, it remains unclear whether deviations from normal WM development during this period contribute to psychopathology.”

Page 2, line 43-44:

“We found that tract-specific deviations in WM development of association and limbic/subcortical systems ...”

Page 2, line 46-47:

“Importantly, delayed brain-age especially in dorsal association tracts predicted psychiatric disorders across diagnoses and disorder onset over a 2-year follow-up.”

Page 2, line 48-49:

“... this study provides a valuable framework for tracking individualized brain development and understanding the neurobiological underpinnings of cognition and transdiagnostic psychopathology”

Results:

Page 8, line 176-182:

“For each participant, the WM tract predicted age is referred to as brain age, and the difference between the brain age and chronological age is defined as tract-based BAG. The tract-based BAG quantifies the extent to which an individual’s tract-specific biological age relative to other same-aged peers based on their tract-specific FA profiles. A negative BAG indicates that an individual’s predicted brain age is less than their chronological age (i.e., individual deviation under model-derived developmental trajectory), while a positive

BAG indicates that an individual's predicted brain age is higher than their chronological age (i.e., individual deviation over model-derived developmental trajectory)."

Page 16, line 332:

"Baseline tract-based BAGs are prospectively linked to cognitive performance two to three years later"

Page 16, line 345-346:

"Among the three cognitive performances, more positive tract-based BAGs were associated with better cognitive performance assessed 2 to 3 years later."

Page 17, line 378-379:

"More negative tract-based BAGs at baseline concurrently associated with greater number of psychiatric (KSADS-5) diagnoses"

Page 20, line 424-425:

"Deviation in WM developmental trajectory assessed via BAGs at baseline prospectively predicted cumulative number of psychiatric (KSADS-5) diagnoses at 2-year follow-up"

Page 20, line 447-448:

"Deviation in WM developmental trajectory assessed via BAGs at baseline prospectively predicted psychiatric diagnosis status transitions from baseline to 2-year follow-up"

Discussion:

Page 24, line 539-547:

"Conceptually, BAG represents the statistical deviation from the age-normative trajectory. However, the calculation of BAG could be impacted by both physiological factors (e.g., neurodevelopmental deviation) and technical noise (e.g. MRI data quality). Therefore, validating that BAG captures neurodevelopmentally meaningful variance is essential. In our study, tract-based BAG showed significant positive associations with pubertal development, including both pubertal stage and hormone levels. This is in line with previous findings showing that puberty is associated with increased FA in major tracts of adolescents⁴⁵, and that higher FAs in WM regions such as the CC and cingulum are linked to elevated gonadal hormone levels in adolescent females⁴⁶. Together, these results provide initial evidence that tract-based BAGs track with known metrics of developmental benchmarks and are consistent with prior BAG findings derived from GM morphology⁴⁷."

And in the Limitations and Future Work section of the Discussion (Page 28-29, line 673-680):

"Second, BAGs may be influenced by multiple factors beyond true neurodevelopmental variation. Previous studies have shown that both technical aspects (e.g. dMRI acquisition and preprocessing schemes⁶⁵) and non-technical factors (e.g. pharmacological

manipulation⁶⁶) can affect BAG estimations. Although we validated the relationships between tract-based BAG and developmental benchmarks (pubertal stages and hormone levels), tract-based BAGs should nonetheless be interpreted as relative indices of deviation from normative WM developmental trajectories. Further work leveraging large-scale longitudinal datasets, integrating a broader range of lifestyle and environmental factors, and more biologically informed tract-based imaging modalities may help disentangle these sources of variability and improve interpretability.”

Reference:

45. Piekarski DJ, Colich NL, Ho TC. The effects of puberty and sex on adolescent white matter development: A systematic review. *Dev Cogn Neurosci* 60, 101214 (2023).
46. Ho TC, Colich NL, Sisk LM, Oskirko K, Jo B, Gotlib IH. Sex differences in the effects of gonadal hormones on white matter microstructure development in adolescence. *Dev Cogn Neurosci* 42, 100773 (2020).
47. Whitmore LB, Weston SJ, Mills KL. BrainAGE as a measure of maturation during early adolescence. *Imaging Neurosci* 1, (2023).
65. Jones DK, Knosche TR, Turner R. White matter integrity, fiber count, and other fallacies: the do's and don'ts of diffusion MRI. *Neuroimage* 73, 239-254 (2013).
66. Le TT, et al. Effect of Ibuprofen on BrainAGE: A Randomized, Placebo-Controlled, Dose-Response Exploratory Study. *Biol Psychiatry Cogn Neurosci Neuroimaging* 3, 836-843 (2018).

Reviewer #1, Comment 2:

2. Handling and Interpretation of Age-Bias Correction

While the authors aim for transparency by including both corrected and uncorrected BAGs, the use of both in parallel without clear analytic separation introduces interpretive ambiguity. The combination of age-bias correction and inclusion of age as a covariate may result in compounded adjustments that are difficult to interpret, and the implications of this approach are not clearly addressed.

Response:

We thank the Reviewer for pointing out this confusion and are happy to clarify. We have further clarified the results with and without age-bias correction in the revised manuscript accordingly. In this study, we separated the analytic purposes for using corrected and uncorrected BAGs. Specifically, the uncorrected BAGs were only used for validation of brain age models (e.g., evaluation of prediction performance) to avoid overestimating prediction accuracy, whereas the age-bias corrected BAGs were used for all primary analyses regarding clinical or behavioral associations, thereby minimizing residual correlations with chronological age.

To avoid compounded adjustments, we conducted additional sensitivity analyses using three adjustment strategies: i) uncorrected BAGs with chronological age included as a covariate (Supplementary Table 14); ii) corrected BAGs without age covariate (Supplementary Table 15); and iii) corrected BAGs with age covariate (primary results, Supplementary Table 12). As shown

in Supplementary Table 12 and Supplementary Table 14-15, the statistical results were highly consistent across these three strategies, as BAGs in ABCD dataset were age-bias-corrected through a linear model derived from the independent training dataset:

We have now further clarified the used of age-bias-corrected and uncorrected BAGs in the Results and Supplementary Material sections:

Results:

Page 8-9, line 185-190:

“We used age-bias corrected BAGs for all primary statistical analyses, whereas uncorrected BAGs were used for validating brain age model performance. While there is ongoing debate regarding the optimal method for bias correction in brain age modeling^{33,34}, our key findings remained robust regardless of whether age-bias correction was applied (see sensitivity analyses), as the BAGs were only adjusted by a linear model of chronological age derived from an independent dataset, and chronological age was additionally included as a covariate in all subsequent analyses.”

Page 9, line 205-206:

“In the sCCA, we included 30 tract-based BAGs (29 tract-specific and a whole-brain tract BAGs with age-bias correction) ...”

Page 11, line 249-250:

“Furthermore, we performed a post-hoc statistical analysis to explore the univariate relationships between the 30 tract-based BAGs (with age-bias correction) and 51 behavioral measures.”

Legend of Figure 5. Page 17:

“(B) Relationships between tract-based measures (with age-bias correction) and follow-up cognitive measures.”

Legend of Figure 6. Page 19:

“(B) Associations between baseline tract-based BAGs (with age-bias correction) and the cumulative number of KSADS-5 diagnoses at baseline, 2-y-follow-up and 2-y-follow-up transdiagnostic status transitions.”

Page 21, line 457-458:

“Furthermore, supplementary post-hoc tract-wise comparisons revealed that the age-bias-corrected BAG of the whole-brain tracts at baseline showed a significant association with diagnostic status transitions”

Page 22, line 496-499:

“... we repeated the analyses using two more alternative strategies: (i) uncorrected BAGs with chronological age included as a covariate, and (ii) age-bias-corrected BAGs without age as a covariate. The results (Supplementary Table 14-15) remained consistent, indicating that the observed associations were not dependent on the bias correction procedure.”

Supplementary materials:

Supplementary Table 14-15

Reference:

33. Gaser C, Kalc P, Cole JH. A perspective on brain-age estimation and its clinical promise. *Nat Comput Sci* 4, 744-751 (2024).

34. Butler ER, et al. Pitfalls in brain age analyses. *Hum Brain Mapp* 42, 4092-4101 (2021).

Reviewer #1, Comment 3:

3. Over-interpretation of Predictive Findings

The manuscript repeatedly uses predictive language (e.g., BAGs “predicted” later outcomes) based on associations in observational data. While some associations are longitudinal, the analyses do not constitute true prediction or causal inference. This wording overstates the findings and could mislead readers about the strength of evidence.

Response:

We agree that the use of predictive terms such as “predict” may unintentionally imply causal inference or out-of-sample predicting, which could not be implied by our study. Our analyses are prospectively observational and, although they include longitudinal associations (baseline BAGs linked to follow-up cognitive and clinical outcomes), they do not establish causality nor predictive modeling.

To address this concern, we have revised the manuscript by using more appropriate terminologies. Specifically:

Results:

Page 16, line 332:

“Baseline tract-based BAGs are prospectively linked to cognitive performance two to three years later”

Page 16, line 334-335:

“..., we next investigated whether developmental deviations in WM tracts could reflect meaningful variance in cognitive performance at follow-ups.”

Page 16, line 345-346:

“Among the three cognitive performances, more positive tract-based BAGs were associated with better cognitive performance assessed 2 to 3 years later.”

Discussion:

Page 26, line 620-622:

“Advanced BAG of specific tracts as a potential predictive biomarker of later cognitive performance

Our findings also highlight the potential of WM maturation in linking to cognitive performance two to three year later...”

Reviewer #1, Comment 4:

4. Cross-Dataset Generalisability and Domain Shift

The authors train models in one dataset and apply them to others without detailed evaluation of generalisability. Domain shifts between cohorts (e.g., due to scanner differences or population composition) may distort predictions. Without clearly presenting out-of-sample performance or calibration by cohort, it is difficult to assess the stability and transportability of the models.

Response:

We quite agree with the reviewer on this comment. Although diffusion tensor imaging (DTI) derived metrics show high reproducibility, differences related to scanner, acquisition protocols and population composition could introduce domain shifts of FA values across sites. Therefore, it is important to remove the site-specific variations prior to combining different sites. We made the following modifications in the revised manuscript: First, we added clarification of domain shifts between sites and cohorts. Second, we have added the Supplementary Figure 1 to demonstrate the domain shifts of FA values and the site-wise distributions of FA values after removing the site effects by data harmonization approach.

We further clarified on these aspects in the Results and added the analyses to the Supplementary Material:

Results:

Page 5, line 116-119:

“Because dMRI metrics are highly site-dependent (Supplementary Figure 1), which can hinder cross-site model training and application, we applied data harmonization (see Supplementary Figure 1 for harmonization effects) prior to all downstream analyses.”

Supplementary materials:

Supplementary Figure 1:

Supplementary Figure 1. Domain shifts of raw fractional anisotropy (FA) values across sites and cohorts and the effects of data harmonization. Ridge density plots depict the distributions of raw (left) and harmonized (right) whole-brain averaged FA values across multiple sites in three developmental cohorts (HCP-D, HBN, and ABCD). Each row corresponds to a site, with density shading reflecting the relative frequency of FA values (darker = higher density). Prior to harmonization, noticeable site-related shifts and variability in FA distributions were evident. After harmonization, distributions became more consistent across sites. Abbreviations: HCP-D, Human Connectome Project in Development); ABCD, Adolescent Brain Cognitive Development; HBN, Healthy Brain Network pediatric mental health.

Reviewer #1, Comment 5:

5. Harmonisation

Diffusion metrics like FA are sensitive to scanner and acquisition parameters, yet the manuscript does not describe any harmonisation approach (e.g. longCombat). This raises concerns about site-related confounding and biases in BAG estimates across cohorts.

Response:

We apologize for the unclear statement of harmonization procedure. We conducted harmonization procedure to correct site-wise differences for FA values. However, we only described this procedure in Methods section and did not demonstrate the harmonization performance. In the revised manuscript, we have added a description of harmonization procedure in the Results section and the site-wise distributions of whole-brain FA values before and after data harmonization. The relevant changes are:

Results:

Page 5, line 116-119:

“Because dMRI metrics are highly site-dependent (see Supplementary Figure 1 for domain shifts of FAs across sites), which can hinder cross-site model training and application, we applied data harmonization (see Supplementary Figure 1 for harmonization effects) prior to all downstream analyses.”

Supplementary materials:

Supplementary Figure 1 (see our response to “Reviewer #1, Comment 5”)

Reviewer #1, Comment 6:

6. Interpretation of BAG Directionality

While the authors include sensitivity analyses controlling for pubertal status and SES, the biological interpretation of BAG directionality remains speculative. The assumption that smaller BAGs reflect developmental “delay” or risk is not validated through direct associations with developmental benchmarks such as age, pubertal stage, or sex-specific trajectories. It remains unclear how BAG varies across these axes or aligns with known nonlinear patterns of white matter maturation.

Response:

We thank the reviewer for this excellent suggestion on validating the associations between BAGs and developmental benchmarks. Specifically, we added analyses showing the distribution of whole-brain BAG across five pubertal categories. In addition, we also tested the associations between whole-brain BAG and biologically developmental benchmarks of puberty categories and hormone levels of DHEA, Testosterone (ERT) and Estradiol (HSE). As shown in the Supplementary Figure 5, more positive whole-brain BAG is associated with more advanced pubertal stages, as well as higher DHEA, ERT and HSE levels.

The manuscript has been updated as follows:

Results:

Page 9, line 191-199:

“Puberty and its accompanying hormonal changes play a critical role in shaping brain development. To validate that BAGs capture biologically meaningful developmental processes, we examined their associations with both pubertal stage assessments and salivary hormone levels (DHEA, testosterone, and estradiol). Across the first two timepoints, higher whole-brain tract BAGs (with age-bias correction) were significantly associated with more advanced pubertal stage ($t_{(10,120)} = 4.685$, $p_{FDR} < 0.001$, Supplementary Figure 5), after controlling for age, sex, and visit. In addition, whole-brain tract BAGs showed significant positive associations with higher levels of DHEA ($t_{(10,630)} = 3.558$, $p_{FDR} < 0.001$), testosterone ($t_{(9,872)} = 4.705$, $p_{FDR} < 0.001$), and estradiol ($t_{(4,583)} = 2.238$, $p_{FDR} = 0.025$). These hormone analyses accounted for covariates including age,

sex, visit, caffeine intake and physical activity in the past 12 hours, collection time since midnight, saliva collection duration, and time from collection to freezer storage.”

Discussion:

Page 24, line 542-547:

“In our study, tract-based BAG showed significant positive associations with pubertal development, including both pubertal stage and hormone levels. This is in line with previous findings showing that puberty is associated with increased FA in major tracts of adolescents⁴⁵, and that higher FAs in WM regions such as the CC and cingulum are linked to elevated gonadal hormone levels in adolescent females⁴⁶. Together, these results provide initial evidence that tract-based BAGs track with known metrics of developmental benchmarks and are consistent with prior BAG findings derived from GM morphology⁴⁷”

Methods:

Page 35-36, line 874-907:

Biological interpretation of BAGs through developmental bench markers

We further included pubertal stage scales and salivary hormone levels as developmental bench markers. Pubertal maturation of the ABCD subjects was measured by the Pubertal Development Scale (PDS)⁸⁸. We utilized the parent-reported PDS as it is likely more reliable than the self-reported version for the age range of our study participants and widely used in other ABCD studies⁸⁹. Because the ABCD 4-year follow-up data are only partially released, analyses in this section focused on the baseline and 2-year follow-up data. Of 8,688 participants at baseline and 5,833 participants at 2-year follow-up, pubertal development data assessed by the Pubertal Development Scale (PDS) were available for 8,263 baseline participants and 4,532 follow-up participants. The PDS provides puberty category measures ranging from prepubertal to post-pubertal. For male participants, the puberty category score was calculated by summing items related to voice deepening, body hair growth, and facial hair growth. These scores were then categorized as follows⁹⁰: prepubertal (= 3), early pubertal (4 or 5), mid pubertal (≥ 6 and ≤ 8), late puberty (≥ 9 and ≤ 11); postpubertal (= 12). For female participants, the puberty category score was derived by summing items on body hair growth and breast development, with categorization as follows: prepubertal (= 2), early pubertal (=3 and no menarche), mid pubertal (> 3 and no menarche), late puberty (≤ 7 and menarche); postpubertal (= 8 and menarche). See Supplementary Figure 5 for distributions of sex-specific pubertal stage distributions across two timepoints. Linear Mixed Effects Models (LMMs) were conducted to explore the relationship between pubertal stage and whole-brain BAGs with age-bias correction:

$$BAG = \beta_0 + \beta_1 \text{Hormone} + \beta_2 \text{Age} + \beta_3 \text{Sex} + \beta_4 \text{Timepoint} + (1|\text{Subject})$$

Salivary hormone concentrations of DHEA, testosterone, and estradiol were assayed by Salimetrics (<https://salimetrics.com>), with samples collected between 7:00 am and 7:00 pm. Hormone data were obtained from the ABCD variables of `hormone_scr_dhea_mean` (DHEA), `hormone_scr_ert_mean` (testosterone), and `hormone_scr_hse_mean` (estradiol) in table `phy_y_sal_horm`. Data cleaning and quality control followed the previous published studies^{91,92}: i) participants with mismatched biological sex (reported at saliva

collection versus biological sex) were excluded; ii) hormone levels outside biologically plausible reference ranges (DHEA: 5~1000 pg/ml; Testosterone: 5~500 pg/ml; Estradiol: 0~1500 pg/ml) were removed; iii) factors of age, sex, visit timepoint, whether have caffeine intake and physical activity in the past 12 hours, saliva collection time since midnight (minutes), saliva collection duration (minutes), and time from collection to freezer storage (minutes) were included as the confounding effects in all analyses. After QC, hormone assessments were available for 10,302 participants for DHEA ($N_{\text{Baseline}} = 8,266$, $N_{2\text{-y-follow-up}} = 2,036$), 10,684 participants for testosterone ($N_{\text{Baseline}} = 8,323$, $N_{2\text{-y-follow-up}} = 2,361$), and 6,413 participants for estradiol ($N_{\text{Baseline}} = 3,944$, $N_{2\text{-y-follow-up}} = 2,469$). Statistical analyses were performed using LMMs:

$$\begin{aligned} \text{BAG} = & \beta_0 + \beta_1 \text{Hormone} + \beta_2 \text{Age} + \beta_3 \text{Sex} + \beta_4 \text{Timepoint} + \beta_5 \text{CaffineIntake} \\ & + \beta_6 \text{PhysicalActivity} + \beta_7 \text{CollectionTime} \\ & + \beta_8 \text{CollectionDuration} + \beta_9 \text{FreezeTime} + (1|\text{Subject}) \end{aligned}$$

Reference:

45. Piekarski DJ, Colich NL, Ho TC. The effects of puberty and sex on adolescent white matter development: A systematic review. *Dev Cogn Neurosci* 60, 101214 (2023).
46. Ho TC, Colich NL, Sisk LM, Oskirko K, Jo B, Gotlib IH. Sex differences in the effects of gonadal hormones on white matter microstructure development in adolescence. *Dev Cogn Neurosci* 42, 100773 (2020).
47. Whitmore LB, Weston SJ, Mills KL. BrainAGE as a measure of maturation during early adolescence. *Imaging Neurosci (Camb)* 1, (2023).
48. Petersen AC, Crockett L, Richards M, Boxer A. A self-report measure of pubertal status: Reliability, validity, and initial norms. *Journal of Youth and Adolescence* 17, 117-133 (1988).
49. Rasmussen AR, et al. Validity of self-assessment of pubertal maturation. *Pediatrics* 135, 86-93 (2015).
50. Kraft D, Alnaes D, Kaufmann T. Domain adapted brain network fusion captures variance related to pubertal brain development and mental health. *Nat Commun* 14, 6698 (2023).
51. Heller C, et al. Hormonal contraceptive intake during adolescence and cortical brain measures in the ABCD Study. *npj Women's Health* 3, 53 (2025).
52. Herting MM, et al. Correspondence Between Perceived Pubertal Development and Hormone Levels in 9-10 Year-Olds From the Adolescent Brain Cognitive Development Study. *Front Endocrinol (Lausanne)* 11, 549928 (2020).

Supplementary materials:

Supplementary Figure 5:

Supplementary Figure 5. Pubertal stage distributions and their associations with whole-brain BAGs.

(A) Distributions of pubertal categories (prepubertal, early pubertal, mid pubertal, late pubertal, and postpubertal) at baseline and 2-year follow-up visits in females and males. Subject counts within each category are shown separately by sex and timepoint.

(B) Distributions of whole-brain brain age gaps (BAGs) across pubertal categories at baseline (top) and 2-year follow-up (bottom), stratified by sex. Ridge plots illustrate how BAGs vary across pubertal stages, with earlier categories (prepubertal/early pubertal) generally showing younger predicted brain age relative to chronological age compared to later stages.

Reviewer #1, Comment 7:

7. Functional and Mitochondrial Decoding Over-interpretation

The use of a single adult brain for mitochondrial mapping limits generalisability, particularly in a paediatric sample. Despite acknowledging this limitation in the discussion, the main results treat the interpretation too strongly given the mismatch in age and sample source.

Response:

We thank the Reviewer for this thoughtful comment. The interpretation of the functional and mitochondrial decoding results should be made with caution, particularly given the mismatch in age between the functional activation and mitochondrial maps and the lack of inter-individual variability in these maps. While our study introduces a novel and explainable framework linking tract-wise patterns to existing functional and postmortem maps, we acknowledge that these findings require further validation using more matched maps.

Based on the Reviewer's suggestion, we have clarified the source of the maps and softened the tone in the Discussion to emphasize that these encoding results are exploratory and hypothesis-generating rather than conclusive. Specifically, we now describe the observed spatial correspondence between tract-based BAG patterns and mitochondrial maps as a preliminary indication of a potential and initial link between metabolically demanding association tracts and their heightened vulnerability to psychiatric diseases during development. These revisions help prevent overstatement and ensure that results are interpreted within the appropriate developmental and methodological constraints:

The manuscript has been updated accordingly:

Results:

Page 14, line 305-306:

“To explore potential biological mechanisms underlying the two cognition-psychopathology profiles, we incorporated the only available set of six high-resolution ex-vivo mitochondrial maps derived from a postmortem adult brain²⁷, ...”

Discussion:

Page 25, line 579-582:

“Our results showing cognitive and psychopathology associations with WM BAGs specific to association and subcortical/limbic tracts reinforces previous brain-behavior findings and may offer preliminary insights into how WM development contributes to distinct brain functional activation patterns derived from meta-analyses.”

Page 25, line 587-589:

“Through mitochondrial decoding analyses, our tract-based analyses provide exploratory evidence that WM tracts most strongly implicated in cognition and psychopathology also show higher mitochondrial enrichment (Fig. 4).”

Page 26, line 599-601:

“Our findings align with post-mortem studies that have shown reduced WM mitochondrial density in patients with psychiatric disorders compared to non-psychiatric controls¹⁴, providing the preliminary support for the “mitochondrial dysfunction” hypothesis in psychiatric disorders¹⁵.”

Reference:

14. Senko D, et al. White matter lipidome alterations in the schizophrenia brain. *Schizophrenia* 10, 123 (2024).

15. Manji H, et al. Impaired mitochondrial function in psychiatric disorders. *Nat Rev Neurosci* 13, 293-307 (2012).

27. Mosharov EV, et al. A human brain map of mitochondrial respiratory capacity and diversity. *Nature* 641, 749-758 (2025).

Reviewer #1, Comment 8:

8. Terminology Around “Maturation”

The manuscript frequently equates positive BAG with “advanced maturation” and negative BAG with “delay.” This language presumes a normative linear developmental trajectory and may not reflect true neurobiological differences. It also risks misinterpretation by clinicians or translational researchers.

Response:

We would like to refer to the above comment (please see Reviewer #1, Comment 1). We have revised the terminology throughout the Results to avoid implying biological “maturation.” Specifically, we replaced terms such as “advanced maturation” and “delayed maturation” with more objective descriptors, such as “more positive BAG” and “more negative BAG,” to emphasize that BAG reflects deviation from age-normative predictions rather than a direct measure of biological maturity. These changes aim to prevent overinterpretation and ensure clearer expression for both clinical and translational audiences.

Reviewer #1, Comment 9:

9. Literature overview and “filling gaps”

There is a lack of engagement with earlier dMRI brain age literature. The study would benefit from highlighting these, especially studies in youth that may have investigated cognition, psychopathology, similar.

Response:

We agree with the reviewer and now have substantially expanded our engagement with the existing dMRI brain age literature, particularly studies in youth populations and those investigating cognitive and psychopathological correlates. Our literature survey reveals that this study provides the first evidence on how dMRI-derived brain age gaps (BAGs), particularly tract-based BAGs, relate to both cognition and psychopathology in adolescents. While previous research has explored various aspects of dMRI brain age, no prior work has simultaneously examined tract-specific aging patterns and their associations with cognitive performance and psychiatric symptoms in youth. Prior dMRI BAG studies (Tønnesen et al., 2020; Zhu et al., 2023) in adult populations with

schizophrenia and bipolar disorder reported more positive whole-brain WM BAGs (accelerated aging), which contrasts with our findings in adolescents, highlighting the importance of age-specific investigations. Notably, our whole-brain BAG results align with the results by Roy et al. (2024), who found that more positive BAGs were associated with greater educational opportunity (an environmental measure serving as a proxy for cognitive enrichment) in the ABCD cohort, and Ullman et al. (2016) demonstrated that higher BAGs related to better working memory and numerical ability in 6-year-old children. Previous research also found more positive dMRI BAGs in schizophrenia patients aged >30 years, but not in patients <30 years old, and our study provides supportive evidence for the "maturational model" in psychiatric patients (Wang et al., 2021). Unlike previous whole-brain approaches, our tract-based analysis identifies the crucial roles of dorsal association tracts during adolescence, revealing tract-specific vulnerabilities that may be neglected in global metrics. As summarized in the table below, existing dMRI brain age studies in youth have been limited in scope, sample size, or have not examined the intersection of tract-specific aging with both cognitive and psychopathological outcomes. Our study uniquely bridges multiple gaps by focusing on the critical adolescent developmental period, employing tract-specific rather than solely whole-brain metrics, simultaneously examining cognitive and psychopathological correlates, and providing evidence for developmental specificity in the relationship between white matter maturation and clinical outcomes.

The dMRI brain age literature in youth or linking BAGs to cognition or psychopathology is summarized in the following table:

Reviewer Table 1. Literature review of WM dMRI brain age studies through youth or linking WM dMRI BAGs to cognition or psychopathology

dMRI features	Reference	Title	Subjects	Links to cognition or psychopathology
576 total and regional measures (e.g. FA, MD, AD)	Beck, et al., Biol. Psychiatry (2024)	Dimensions of early life adversity are differentially associated with patterns of delayed and accelerated brain maturation	8,834 ABCD subjects (age range: 8.9-13.8 years) from two timepoints	NA
Diffusion kurtosis FA profiles of 28 tracts	Roy, et al., Dev. Cogn. Neurosci. (2024)	Differences in educational opportunity predict white matter development	6,410 baseline data and 4,770 2-y follow-up data of ABCD dataset	NA
FA and MD profiles of 18 tracts	Richie-Halford, et al., PLoS Comput. Biol. (2021)	Multidimensional analysis and detection of informative features in human brain white matter.	1,651 subjects with ages 5-21 of HBN dataset; 76 subjects with ages 6-50 of Weston-Havens dataset	NA
FA voxel-wise maps	Ullman, H. et al., Cereb. Cortex (2016)	Timing of White Matter Development Determines Cognitive Abilities	Data set 1: 82 subjects (6-20 years old) Data set 2:	The predicted brain age positively correlated with working memory performance and

		at School Entry but Not in Late Adolescence	31 subjects (age: 6.8 ± 0.3 years)	numerical ability ($P < 0.01$, $P < 0.05$) in the 6-year-old children; This association between brain age and working memory was no longer significant in children older than 13 years of age
FA, AD and RD values of 41 bilateral WM regions	Shokri-Kojori, E. et al., Brain Res. (2021)	Estimates of brain age for gray matter and white matter in younger and older adults: Insights into human intelligence	100 participants (18-78 years old)	Higher WM age scores were associated with lower crystallized intelligence (measured by WASI vocabulary test) in older subjects (age > 40)
FA, MD, RD and AD skeleton values (96 measures)	Tønnesen, S. et al., Biol. Psychiatry: Cogn. Neurosci. Neuroimaging (2020)	Brain Age Prediction Reveals Aberrant Brain White Matter in Schizophrenia and Bipolar Disorder: A Multisample Diffusion Tensor Imaging Study	Training set: 928 healthy control (HC) subjects (18-94 years of age); Testing set: 648 patients with schizophrenia (SCZ), 185 patients with bipolar disorder (BD), 990 HC subjects	More positive BAGs in patients with SCZ and BD
FA voxel-wise maps for 48 white-matter regions	Zhu, et al., Transl. Psychiatry (2023).	Investigating brain aging trajectory deviations in different brain regions of individuals with schizophrenia using multimodal magnetic resonance imaging and brain-age prediction: a multicenter study	Training set: 230 HC subjects (age range: 20-84 years) Testing sets: 194 SCZs and 100 HCs; 50 HCs and 50 SCZs	10 WM regions showed more positive BAGs in SCZs compared to HCs
FA values for 43 WM regions	Wang, J. et al., Schizophr. Res. (2021).	White matter brain aging in relationship to schizophrenia and its cognitive deficit	Training set: 107 HC subjects (age range: 19.129-63.817 years) Testing sets:	More positive BAG in age > 30 patients with SCZs compared to HCs; No significant results in age < 30 patients;

			194 SCZs and 100 HCs; 50 HCs and 50 SCZs	In HCs, more positive dMRI BAG is negatively linked to worse processing speed; In SCZ patients, was significantly and negatively correlated with performance on the working memory and processing speed.
--	--	--	--	---

The Introduction and Discussion sections have been updated as follows:

Introduction:

Page 4, line 87-93:

“To date, this method has predominantly focused on studies using morphological measures of gray matter (GM), with aggregate results from these studies suggesting that psychopathology is associated with accelerated brain aging in adults and youth²²⁻²⁴, However, few BAG studies have used diffusion MRI (dMRI) measures of WM, and these studies are mostly in adults. Results from these WM-BAG studies also suggest that adult patients with single psychiatric diagnoses (e.g. schizophrenia and bipolar disorder) may have more advanced WM brain aging relative to non-psychiatric controls^{25, 26}. While these prior studies highlight the relevance of the BAG in psychopathology, the role of WM maturation, particularly the tract-specific development, in adolescents is still unclear.”

Discussion:

Page 24-25, line 560-574:

“Importantly, our findings extend previous dMRI brain age research by demonstrating that tract-specific BAGs capture shared dimensions of cognitive and psychopathological variation during adolescence. While prior dMRI studies have reported associations between whole-brain WM BAGs and psychiatric disorders in adults, the directionality of these findings was opposite to our results. For instance, studies in adult schizophrenia and bipolar disorder patients have shown more positive whole-brain WM BAGs^{25,26}, suggesting accelerated aging of WM microstructure. However, our results revealed that more positive tract-based BAGs were associated with better cognitive performance and reduced psychopathological risks. This contradiction may reflect fundamental differences between adolescent and adult neuropathology from the developmental view. Consistent with this neurodevelopmental interpretation, Roy et al. found that more positive whole-brain WM BAGs were associated with better educational opportunity in youth⁴⁴, and Ullman et al. demonstrated that higher whole-brain WM brain-age related to better working memory and numerical ability in 6-year-old children⁵¹. These age-dependent patterns in younger agers support that WM deficits were proposed to occur prior to illness onsets, suggesting that psychiatric disorders are neurodevelopmental in nature⁵². Furthermore, our tract-

specific approach further reveals that the crucial roles of dorsal association tracts during adolescence, revealing tract-specific vulnerabilities that may be neglected in global metrics.”

Reference:

22. Kaufmann T, et al. Common brain disorders are associated with heritable patterns of apparent aging of the brain. *Nat Neurosci* 22, 1617-1623 (2019).
23. Schnack HG, van Haren NE, Nieuwenhuis M, Hulshoff Pol HE, Cahn W, Kahn RS. Accelerated Brain Aging in Schizophrenia: A Longitudinal Pattern Recognition Study. *Am J Psychiatry* 173, 607-616 (2016).
24. Croyley VL, et al. Brain-Predicted Age Associates With Psychopathology Dimensions in Youths. *Biological Psychiatry: Cognitive Neuroscience and Neuroimaging* 6, 410-419 (2021).
25. Tonnesen S, et al. Brain Age Prediction Reveals Aberrant Brain White Matter in Schizophrenia and Bipolar Disorder: A Multisample Diffusion Tensor Imaging Study. *Biol Psychiatry Cogn Neurosci Neuroimaging* 5, 1095-1103 (2020).
26. Zhu JD, Wu YF, Tsai SJ, Lin CP, Yang AC. Investigating brain aging trajectory deviations in different brain regions of individuals with schizophrenia using multimodal magnetic resonance imaging and brain-age prediction: a multicenter study. *Transl Psychiatry* 13, 82 (2023).
44. Roy E, et al. Differences in educational opportunity predict white matter development. *Dev Cogn Neurosci* 67, 101386 (2024).
51. Ullman H, Klingberg T. Timing of White Matter Development Determines Cognitive Abilities at School Entry but Not in Late Adolescence. *Cereb Cortex* 27, 4516-4522 (2017).
52. Kochunov P, Hong LE. Neurodevelopmental and Neurodegenerative Models of Schizophrenia: White Matter at the Center Stage. *Schizophrenia Bulletin* 40, 721-728 (2014).

Reviewer #1, Comment 10:

Minor:

1. RMSE and Pearson’s R are not reported, though they are standard metrics for model evaluation. Their absence makes it harder to contextualise the predictive performance.

Response:

We thank the Reviewer for the helpful suggestion to report RMSE and Pearson’s *R* values for model evaluation. These metrics have now been added to Tables S2-S4 in the Supplementary Material and incorporated accordingly into the revised manuscript:

Results:

Page 5, line 126-128:

*“The 5-fold cross-validation performance of the GPR model showed a Pearson *R* for the relationship between chronological age and predicted age of 0.855 and a root mean squared error (RMSE) of 2.264 years.”*

Page 5, line 130-132:

“As shown in Supplementary Table 3 and Supplementary Figure 2, whole-brain brain-age model significantly estimated age in HBN dataset with an R of 0.622 and a RMSE of 3.337 years.”

Page 5, line 132-135:

“For the prediction of brain ages in another independent dataset, the ABCD study ($N_{Baseline} = 8,688$, 4,205 females, $N_{2-y-follow-up} = 5,883$, 2,741 females; $N_{4-y-follow-up} = 2,351$, 1,118 females, age range = 8.91-15.70 years), of which the age range was much narrower and the prediction performance was with an R of 0.543 and a RMSE of 2.041 (Supplementary Figure 2).”

Reviewer #1, Comment 11:

2. The sentence "...are poorly understand" should be "poorly understood".

Response:

We thank the Reviewer for pointing this out and apologize for the typo. We have now corrected the sentence accordingly.

Introduction:

Page 3, line 75-76:

“However, links between WM development, mitochondrial functioning, and psychopathology are poorly understood.”

Reviewer #1, Comment 12:

3. Some references have gaps between the word before it (e.g. reference 1) while others do not.

Response:

We apologize for the mistake. We have carefully reviewed the manuscript and corrected the formatting inconsistencies by removing unnecessary spacing before the reference numbers.

Reviewer #1, Comment 13:

4. The label “Association Development mode” used to describe sCCA components implies a specificity that may not be warranted by the data.

Response:

We have rephrased “Association Development mode” and “Subcortical/limbic Development mode” to “Association BAG mode” and “Subcortical/limbic BAG mode”, respectively.

Reviewer #1, Comment 14:

5. The manuscript briefly mentions differences between GM- and WM-derived BAGs but does not integrate this contrast meaningfully into the main interpretation.

Response:

Thank you for this insightful comment on integrating deeper interpretation for the contrast between GM- and WM-derived BAGs. We have now expanded the Discussion to integrate neurodevelopmental frameworks that highlight distinct developmental trajectories and cellular processes in GM versus WM, as well as how these may differentially relate to psychiatric symptoms. Specifically, we emphasize that while advanced GM maturation may reflect premature synaptic pruning, advanced WM maturation may indicate protracted myelination supporting improved communication between distinct brain regions. Furthermore, according to life-along charts of GM and WM neurodevelopmental trajectories, WM maturation is a protracted and ongoing process throughout adulthood. In contrast, GM maturation is characterized by changes in cortical volume, surface area and thickness that are followed by inverted U-shaped developmental trajectories. These distinctions may explain the divergent associations of GM- and WM-derived BAGs with psychopathology and align with emerging literature suggesting that neurodevelopmental “pace” may manifest differently across tissue types.

Discussion:

Page 26, line 605-618:

“The divergent associations of WM- and GM-derived BAGs with psychopathology suggest that brain age reflects distinct and multi-dimensional maturational processes. GM maturation, reflected by changes in cortical volume, surface area and thickness, typically followed by an inverted U-shaped developmental trajectory by early synaptogenesis peaks in early childhood, followed by synaptic pruning and cortical thinning extending through adolescence²¹. In contrast, WM maturation reflects a more protracted course characterized by continued increases in myelination and fiber organization into early adulthood, supporting increasingly efficient long-range communication⁹. Thus, a more “advanced” GM age during adolescence may indicate accelerated pruning and reduced plasticity²⁴, consistent with prior work linking higher GM BAG to vulnerability to psychiatric symptoms. Conversely, a more “advanced” WM age likely reflects greater myelination and fiber organization. This interpretation aligns with developmental frameworks suggesting that deviations in neurodevelopmental pace may differ across tissue compartments and that low-risk environments support prolonged plasticity in GM but continued strengthening of WM pathways³⁶. The opposing directions of GM and WM BAG associations therefore likely represent complementary neurodevelopmental clocks rather than conflicting findings. Future multimodal longitudinal studies are needed to trace how GM and WM maturation jointly shape risk versus resilience trajectories across adolescence.”

Reference:

9. de Faria O, Pivonkova H, Varga B, Timmler S, Evans KA, Káradóttir RT. Periods of synchronized myelin changes shape brain function and plasticity. *Nature Neuroscience* 24, 1508-1521 (2021).
21. Bethlehem RAI, et al. Brain charts for the human lifespan. *Nature* 604, 525-533 (2022).

24. Cropley VL, et al. Brain-Predicted Age Associates With Psychopathology Dimensions in Youths. *Biological Psychiatry: Cognitive Neuroscience and Neuroimaging* 6, 410-419 (2021).

56. Rakesh D, Whittle S, Sheridan MA, McLaughlin KA. Childhood socioeconomic status and the pace of structural neurodevelopment: accelerated, delayed, or simply different? *Trends Cogn Sci* 27, 833-851 (2023).

Reviewer #1, Comment 15:

6. Figures do not consistently indicate whether the BAG values shown are corrected or uncorrected.

Response:

We thank the Reviewer for the suggestion. We have now revised all figure captions to clearly indicate whether the BAG values presented are age-bias-corrected or uncorrected accordingly:

Results:

Legend of Figure 5. Page 17:

“(B) Relationships between tract-based measures (with age-bias correction) and follow-up cognitive measures.”

Legend of Figure 6. Page 19:

“(B) Associations between baseline tract-based BAGs (with age-bias correction) and the cumulative number of KSADS-5 diagnoses at baseline, 2-y-follow-up and 2-y-follow-up transdiagnostic status transitions.”

Supplementary materials:

Legend of Supplementary Figure 7

“(B) Associations between baseline tract-based BAGs (with age-bias correction) and the cumulative number of 2-y-follow-up cumulative Kiddie Schedule for Affective Disorders and Schizophrenia for DSM-5 (KSADS-5) diagnoses.”

Reviewer #1, Comment 16:

7. The choice of Gaussian Process Regression (GPR) is not justified against alternative modeling approaches.

Response:

We thank the Reviewer for the suggestion to further justify the choice of Gaussian Process Regression (GPR) model to predict chronological age by tract-profiles. GPR is a nonparametric Bayesian machine learning method that can capture nonlinear developmental trajectories of WM microstructure. In addition, GPR has been widely applied in the brain age studies. Previous comparative studies have shown that GPR performs competitively or better than alternative approaches. We have now updated the manuscript as follows:

Results:

Page 5, line 122-125:

“The models were trained by a Gaussian Process Regression (GPR) algorithm²⁸, a nonparametric Bayesian machine learning approach capable of capturing the nonlinear developmental trajectories of WM microstructure. In addition, GPR has been widely adopted in brain-age research^{29,30}, and previous comparative studies have demonstrated that GPR performs competitively or better than alternative algorithm^{30,31}”

Reference:

28. Rasmussen CE. Gaussian processes in machine learning. *Lect Notes Artif Int* 3176, 63-71 (2004).
29. Cole JH, et al. Brain age predicts mortality. *Molecular Psychiatry* 23, 1385-1392 (2017).
30. Ball G, Kelly CE, Beare R, Seal ML. Individual variation underlying brain age estimates in typical development. *NeuroImage* 235, (2021).
31. More S, et al. Brain-age prediction: A systematic comparison of machine learning workflows. *Neuroimage* 270, 119947 (2023).

Reviewer 2#

The manuscript by Wang et al assesses whether adolescents' white matter brain-age gap (BAG) relates to cognition or psychopathology. Here, BAG is the difference in true ages and ages predicted by a machine learning model trained on white-matter tracts' fractional anisotropy. The model is trained in HCPD and applied to primarily ABCD (with further validation in HBN). The authors also characterize the covariance structure of tract-wise BAG and a variety of cognitive and behavioral traits as assessed in the ABCD sample using sparse canonical correlation analysis (sCCA), describing two primary modes of covariance - Association and Subcortical/limbic development modes. Finally, they test whether the spatial patterning of these modes correlated with post-mortem patterns of metabolic markers and if individual BAGs across these modes, across all tracts, and/or within specific brain tracts are associated with cognitive metrics or diagnostic status. Overall, the paper is expansive and intended to address the very relevant issue of how non-typical white matter development in various tracts may be associated with cognitive function and psychopathology. Strengths include the use of 3 independent datasets and the inclusion of sensitivity analyses for pubertal stage and the effects of age-correction. However there are several major areas of concern relating to clarity, possible circularity, and other methodological issues. These concerns significantly limit the potential impact of the paper especially relative to the many other BAG papers published in the last several years with overlapping samples and research questions. We hope the following specific comments are helpful.

Response:

We appreciate the reviewer's constructive suggestions and insightful comments. We believe that our revisions made based on the reviewer's comments have strengthened the manuscript and improved clarity. Below we summarize the key changes:

- We have added sensitivity analyses and new dMRI data from the ABCD 4-year-followup waves to examine the influence of age range on brain-age prediction performance. After controlling the sample size across comparison sets, we observed higher predictive performance in test subsets with wider age ranges.
- To rule out the potential circularity between full-set behavioral inputs and sCCA BAG variates, we performed additional sCCA analyses using only cognition or psychopathology measures. The resulting BAG and behavioral loading patterns of the two sCCA variates showed high consistency across these three cases (cognition-, psychopathology-, and full-behavior-derived). Moreover, the associations with follow-up cognition were replicated using the psychopathology- and full-behavior-derived sCCA BAG variants. The associations with psychiatric diagnoses were also replicated using the cognition- and full-behavior-derived sCCA BAG variants.
- We revised the text to clarify the analytical framework in our study. We also clarified the hierarchical structure of analyses across tract- and sCCA-derived BAG variates.

Reviewer #2, Comment 1:

1. It is not adequately justified why BAG model is fitted in HCPD (smaller sample, bigger age range) and then tested in ABCD. This is especially important given that the out of sample r^2 of the age predictions in ABCD is quite low (0.185) raising significant concerns about the interpretability of the downstream BAG analyses.

Response:

We thank the reviewer for this important comment. In this study, we intended to train the brain age models using an independent dataset and then apply the models to ABCD and HBN datasets for brain-behavior association analyses, as the ABCD study includes rich imaging, cognitive, and clinical assessments. Ideally, the training dataset should have a large sample size of healthy individuals covering a wide age-range including adolescence, as well as high-quality diffusion MRI. Based on these criteria, we chose HCP-D dataset to train the models, as it fits for the majority of the requirements, although its sample size is moderate (N=611). Training models in an independent dataset, rather than in only one dataset, avoids from potential overfitting of the models. However, the predictive performance in ABCD dataset is lower than that in HBN dataset. One of the reasons might be that the ABCD dataset had narrower age-range compared to HBN dataset. Below we added two supplementary analyses to address the concern about model performance:

- 1) We additionally performed simulation analyses, by sampling the same number of HBN subjects (N = 100) for 100 times with different age ranges, to test how age ranges of testing sets influence the predictive performance. As shown in Supplementary Figure 3, the results demonstrate that R increases monotonically with broader age ranges, indicating that narrower age range (e.g., 9–13 years as in ABCD) inherently constrain the variance explained by brain age models.

Supplementary Figure 3. Effect of age range on the predictive performance of brain age model. Simulation analyses were conducted using the Healthy Brain Network (HBN) dataset to examine how the age range of testing samples influences model performance. For each age range (indicated on the y-axis), 100 subjects were randomly sampled for 100 times. The x-axis

shows the distributions of Pearson correlations between the predicted ages (whole-brain brain ages without age-bias correction) and chronological ages across iterations.

- 2) To further examine the effect of age range on predictive performance in the ABCD dataset, we incorporated the newly released 4-year follow-up dMRI data ($N = 2,351$). We kept the sample size constant ($N = 1,500$) across 100 resampling iterations and included only non-overlapped subjects across all three time points, as well as across just the baseline and 2-year follow-up waves. As shown in Reviewer Figure 1, predictive performance (R value) improved substantially in data with 3 time points (baseline, 2-year, and 4-year follow-up), relative to that with only the baseline and 2-year follow-up.

Reviewer Figure 1. Influence of age-range on predictive performance in the ABCD cohort.

Density distributions of prediction accuracy (Pearson’s R between chronological age and predicted age) across 100 resampling iterations ($N_{\text{sample}} = 1,500$ per iteration). Two sampling strategies were compared: (i) using only baseline and 2-year follow-up dMRI data, and (ii) using all three available time points (baseline, 2-year, and 4-year follow-up).

The manuscript has been updated as follows:

Results:

Page 6, line 136-141:

“To evaluate the effects of age-range and intra-subject variability on the predictive performance of the brain-age models, we performed two additional analyses. First, we repeatedly resampled the same number of HBN participants ($N = 100$) across 100 iterations while systematically varying the age range. ... As shown in Supplementary Figure 3, predictive performance (R values) increased as the age-range becomes wider.”

Materials and Methods:

Page 34, line 839-841:

“We used the HCP-D dataset for training and cross-validation of brain age prediction models because it included high-quality dMRI from a moderate cohort of healthy participants ($N = 611$) covering a wide age range (5.58 - 21.90 years).”

Discussion:

Page 28, line 669-673:

“Additionally, we trained the tract-based models using the HCP-D dataset, which provides high-quality dMRI of healthy adolescents and covers a wide age-range in adolescence, allowing us to avoid overfitting by training models independently from the ABCD and HBN datasets. However, the moderate sample size of HCP-D ($N = 611$) suggests that future work may benefit from incorporating larger and more diverse multisite datasets to further enhance model generalizability.”

Supplementary materials:

Supplementary Figure 3: (please see the above figure)

Reviewer #2, Comment 2:

2. There is concern that several of the analyses presented are redundant or circular. For instance, the two sCCA modes were defined based on BAG’s covariance with cognitive and behavioral measures. This makes their subsequent tests of sCCA’s spatial covariance with neurosynth terms (Figure 3) and sCCA variants’ associations with KSADS and cognitive variables (which are almost certainly highly covariant with the cognitive and behavioral variables used to define the modes) problematic.

Response:

We thank the reviewer for pointing out this potential issue. In the revised manuscript, we have included further clarifications and analyses for testing consistency of the two sCCA variate loadings by all 51 behavioral measures, or solely by 20 cognitive or 31 psychopathological measures. As shown in Supplementary Figure 10-12, the sCCA loading patterns of Association and Subcortical/limbic BAG modes showed significant consistency among the 3 cases for both behavioral (combined vs. cognition only, $\rho_{\text{Association}} = 0.95$, $p_{\text{Association}} < 0.01$, $\rho_{\text{Subcortical/limbic}} = 0.91$, $p_{\text{Subcortical/limbic}} < 0.01$; combined vs. psychopathology, $\rho = \rho_{\text{Association}} = 0.92$, $p_{\text{Association}} < 0.01$, $\rho_{\text{Subcortical/limbic}} = 0.87$, $p_{\text{Subcortical/limbic}} < 0.01$) and BAG (combined vs. cognition only, $\rho_{\text{Association}} = 0.97$, $p_{\text{Association}} < 0.01$, $\rho_{\text{Subcortical/limbic}} = 0.97$, $p_{\text{Subcortical/limbic}} < 0.01$; combined vs. psychopathology, $\rho_{\text{Association}} = 0.81$, $p_{\text{Association}} < 0.01$, $\rho_{\text{Subcortical/limbic}} = 0.40$, $p_{\text{Subcortical/limbic}} < 0.01$) sets after false-discovery-rate correction. The high cross-domain consistency across the 3 cases indicated the tight association of tract-based BAGs between the general cognition “g-factor” and the general psychopathology “p-factor”. Specifically, the cognitive and psychopathological measures are aligned in the same dimension (although in opposite directions) in the multi-dimensional behavioral space, rather than orthogonal each other.

Next, we examined the associations between psychopathology-derived sCCA BAG variates and follow-up cognitive performance. Consistent with the full-behavior-derived results, psychopathology-derived Association BAG variate still exhibited stronger associations with follow-up cognitive performance than the Subcortical/limbic BAG variate. Furthermore, we also investigated the relationships between cognition-derived sCCA BAG variates and the cumulative number of KSADS-5 diagnoses at baseline, at 2-y-follow-up, and the transition of the

transdiagnostic status between baseline and 2-y-follow-up. GLM analyses demonstrated the statistical patterns were consistent with the behavior-derived results: more negative Association and Subcortical/limbic BAG variate scores were associated with more comorbid psychiatric disorders both at baseline and at 2-y-follow-up. In addition, the NeuroSynth findings were derived from meta-analytic functional decoding rather than from dMRI data. Thus, the functional associations we report reflect an independent mapping between the structurally defined WM tracts and large-scale functional activation patterns identified from thousands of task-based fMRI studies. This provides preliminary evidence for tract-level structure-function coupling.

We have now added these analyses and clarifications in the revised manuscript:

Results:

Page 21-22, line 467-490:

“Consistency of the BAG variates and their associations with cognitive performance and psychiatric diagnoses

To further evaluate the consistency of the two sCCA BAG variates, we repeated the sCCA analyses using behavioral measures from either the cognition or psychopathology domain. In both domain-specific analyses, the sCCA variates of Association and Subcortical/limbic BAG modes still showed significance after permutation testing. The loading patterns of the re-derived sCCA variates showed significant consistency with the Association and Subcortical/limbic BAG modes using full behavioral set, respectively, on both behavioral and BAG sides. See Supplementary Figure 10-12 for more details.

Next, we examined the associations between psychopathology-derived sCCA BAG variates and follow-up cognitive performance. Consistent with the full-behavior-derived results, psychopathology-derived Association BAG variate (School Grade: $F_{(1, 7868)} = 58.86$, $\Delta R_{adjusted}^2 = 0.72$, $p < 0.001$; SMARTE: $F_{(1, 5623)} = 93.50$, $\Delta R_{adjusted}^2 = 1.55$, $p < 0.001$; STRP_ACC: $F_{(1, 5967)} = 32.13$, $\Delta R_{adjusted}^2 = 0.51$, $p < 0.001$) exhibited stronger associations with follow-up cognitive performance than the Subcortical/limbic BAG variate (School Grade: $F_{(1, 7868)} = 19.62$, $\Delta R_{adjusted}^2 = 0.23$, $p < 0.001$; SMARTE: $F_{(1, 5623)} = 65.47$, $\Delta R_{adjusted}^2 = 1.08$, $p < 0.001$; STRP_ACC: $F_{(1, 5967)} = 11.63$, $\Delta R_{adjusted}^2 = 0.18$, $p < 0.001$).

Furthermore, we also investigated the relationships between cognition-derived sCCA BAG variates and the cumulative number of KSADS-5 diagnoses at baseline, at 2-y-follow-up, and the transition of the transdiagnostic status between baseline and 2-y-follow-up. GLM analyses demonstrated the statistical patterns were consistent with the behavior-derived results: more negative Association and Subcortical/limbic BAG variate scores were associated with more comorbid psychiatric disorders both at baseline (Association mode: $F_{(2, 8591)} = 14.96$, $p < 0.001$; Subcortical/limbic mode: $F_{(2, 8591)} = 6.54$, $p = 0.0015$) and at 2-y-follow-up (Association mode: $F_{(2, 7904)} = 11.94$, $p < 0.001$; Subcortical/limbic mode: $F_{(2, 7904)} = 7.64$, $p < 0.001$), and the longitudinal transition of KSADS-5 diagnosis status (Association mode: $F_{(3, 7826)} = 10.44$, $p < 0.001$; Subcortical/limbic mode: $F_{(3, 7826)} = 5.07$, $p = 0.0017$).”

Supplementary materials:
Supplementary Figure 10

Supplementary Figure 10. sCCA loadings for tract-based BAG measures and behavioral variables across cognition- (blue), psychopathology- (red), and full-behavior-derived (gray) Association BAG modes.

(A) Tract-based BAG loadings for the Association BAG mode derived from sCCA variates separately by cognition measures (top, blue), psychopathology measures (middle, red), and the combined behavioral set (bottom, gray). Bars represent feature weights (loadings), indicating the contribution of each tract-based BAG measure to this sCCA variate. Tracts within the dorsal association systems showed the strongest positive contributions across all derived models.

(B) Behavioral loadings corresponding to the same cognition-, psychopathology-, and combined behavioral Association BAG modes. These loadings highlight the consistency

of tract-based BAG signatures with behavioral dimensions of cognitive function and psychopathology.

Supplementary Figure 11:

Supplementary Figure 11. sCCA loadings for tract-based BAG measures and behavioral variables across cognition- (blue), psychopathology- (red), and full-behavior-derived (gray) Subcortical/limbic BAG modes.

(A) Tract-based BAG loadings for the Subcortical/limbic BAG mode derived from sCCA variates separately by cognition measures (top, blue), psychopathology measures (middle, red), and the combined behavioral set (bottom, gray). Bars represent feature weights (loadings), indicating the contribution of each tract-based BAG measure to this sCCA variate. Tracts within the limbic or subcortical systems showed the strongest positive contributions across all derived models.

(B) Behavioral loadings corresponding to the same cognition-, psychopathology-, and combined behavioral Subcortical/limbic BAG modes. These loadings highlight the

consistency of tract-based BAG signatures with behavioral dimensions of cognitive function and psychopathology.

Supplementary Figure 12:

Supplementary Figure 12. Consistency of tract-BAG and behavioral loading patterns across cognition-, psychopathology-, and full-behavior-derived sCCA modes.

Heatmaps display pairwise Spearman correlation coefficients between loading patterns obtained from sCCA modes derived separately from cognition measures, psychopathology measures, and the full behavioral set. Left panel: Association BAG mode; right panel: Subcortical/limbic BAG mode. Upper triangular matrices depict correlations between tract-BAG loadings, and lower triangular matrices depict correlations between behavioral loadings. More positive correlations indicate more convergent loading patterns across analytic frameworks, highlighting the robustness and shared structure of the Association BAG mode, with less consistent correspondence for the Subcortical/limbic BAG mode. Color bar indicates Spearman correlation coefficient.

Reviewer #2, Comment 3:

3. Similarly, repeating associations between cognitive/clinical markers and BAG at the level of sCCA mode, whole-brain, tract, and network are presented without clarity as to the interdependence of these tests. Greater justification, clearer interpretation, or clarity as to which analyses are considered supplementary would likely be helpful.

Response:

We appreciate the suggestion and have added the clarifications in the Results section. In our association analyses between tract-based BAG-related measures and independent cognitive/psychopathological assessments, the sCCA tract-based variates represent the primary results, as they capture the primary tract-wise developmental dimensions concurrently associated with better cognitive function and less psychopathological risks at baseline. The tract-specific analyses were performed as post-hoc analyses to identify which specific tracts or WM systems contributed most to the cognitive/clinical markers. Additionally, the previous network-level

analyses using support vector regression (SVR) have been removed from the manuscript, as they are largely redundant with the tract-level results, as Reviewer 3 suggested.

The manuscript has been updated as follows:

Results:

Page 16, line 343-345:

“Post-hoc tract-wise analyses further revealed significant associations between tract-based BAGs association and the three cognitive measures, as shown in Figure 5B and Supplementary Table 8-10.”

Page 18, line 390-391:

“In addition, we also conducted the post-hoc association analyses on the relationships of BAGs (with age-bias correction) obtained from individual tracts and whole-brain tracts ...”

Page 20, line 433-434:

“Post-hoc tract-wise comparisons demonstrated that age-bias-corrected BAG of the whole-brain tracts at baseline were significantly associated with the cumulative number of diagnoses at 2-year follow-up ...”

Page 21, line 457-458:

“Furthermore, supplementary post-hoc tract-wise comparisons revealed that the age-bias-corrected BAG of the whole-brain tracts at baseline showed a significant association with diagnostic status transitions ...”

Reviewer #2, Comment 4:

4. It is not clear why the BAG tends to be the outcome variable in GLMs when the authors are interested in how much BAG contributes to current/future psychiatric/cognitive symptoms. Why aren't the models set up so that behavior/cognition are the outcome variables and BAGs are the predictor variables (while controlling for demographic covariates)?

Response:

We thank the Reviewer for this comment and have clarified our analyses for the association linking tract-wise BAG measures and cognitive/psychiatric measures. In our study, the choice of statistical model structure depends on whether the analysis focused on group-wise comparisons or continuous behavioral associations, rather than predicting the cognitive/psychiatric measures. Our primary interest was to determine whether inter-individual differences in tract-based BAG measures capture clinically meaningful variance. Specifically, for the clinical diagnosis analyses, our goal was to test group-wise differences in tract-based BAG measures across diagnostic subgroups. Accordingly, BAGs were modeled as the dependent variables to quantify how these neurodevelopmental indices vary as a function of the cumulative number of psychiatric diagnoses, while controlling for demographic covariates (e.g., age and sex). The statistical modeling was

analogous to our previous functional imaging study (Xiao et al., 2023) where imaging-derived metrics serve as dependent variable to characterize group differences.

The manuscript has been updated as follows:

Results:

Page 16, line 333-335:

“Having established the associations between maturation of WM tracts and both cognitive function and psychopathology, we next investigated whether developmental deviations in WM tracts could reflect meaningful variance in cognitive performance at follow-ups.”

Methods:

Page 37, line 943-944:

“To assess whether inter-individual differences in the tract-BAG measures capture clinically relevant differences in psychiatric diagnoses, ...”

Reference:

Xiao X, et al. Brain Functional Connectome Defines a Transdiagnostic Dimension Shared by Cognitive Function and Psychopathology in Preadolescents. Biological Psychiatry, (2023).

Reviewer #2, Comment 5:

Minor: please define all acronyms in figure legends (not just in Supplementary Table 1)

Response:

Thank you for the suggestion. We have adjusted all figure legends to ensure that the full names of all tracts are defined. This change has been implemented throughout the main and supplementary figures for clarify and consistency.

Reviewer 3#

Overview: This study developed a model of brain age using diffusion-weighted imaging data from the HCP development dataset and validated it in the ABCD and Healthy Brain Network (HBN) cohorts with modest effect sizes for their validation. To determine clinical relevance, they identify associations between derived brain age estimates and cognition and clinical measures in ABCD at baseline and at 2-year follow-up. Finally, they use postmortem data to link finding to mitochondrial WM profiles. This is a lot of work and the overall result is a relatively cohesive characterization of how individual differences in brain-age derived estimates of WM may link to clinical and cognitive features during preadolescence.

Response:

We appreciate the reviewer's thoughtful and comprehensive comments on our work, including the conceptual interpretation of brain age metrics and the challenge of disentangling biological meaning from potential technical and physiological confounds. In response, we have clarified the developmental interpretation of the white-matter brain-age gap (BAG), conducted additional comparison analyses, and strengthened our discussion of limitations and biological plausibility. Key revisions include:

- We revised the text throughout the manuscript to emphasize that tract-based BAG reflects deviation from age-normative white-matter microstructural trajectories, rather than a direct measure of “advanced” or “delayed” maturation. We replaced interpretive terms (e.g., “delayed maturation”) with neutral phrasing (e.g., “more negative BAG”), and clearly state that BAG should be interpreted as an index of deviation, not a deterministic biological marker.
- To support the neurodevelopmental relevance of BAG in youth, we now include analyses linking whole-brain BAG with established neurodevelopmental benchmarks of puberty (pubertal stage and hormone levels). As expected, higher BAG values were associated with more advanced pubertal stage and higher pubertal hormone levels, reinforcing that BAG captures variance aligned with known developmental processes.
- We added direct comparisons between tract-based BAGs and tract-specific FA values in associating follow-up cognitive performance and KSADS diagnoses. While both metrics contained meaningful information, tract-based BAGs explained additional variance beyond FA in multiple tracts, suggesting that BAG captures complementary and differential developmental variance not reflected by raw FA alone.
- We strengthened the limitations section to acknowledge factors that may influence BAG, including technical properties of diffusion imaging and non-biological influences, and emphasized the importance when interpreting BAG in developmental cohorts.

Reviewer #3, Comment 1:

However, there are some methodological and ‘big picture’ concerns, as described below: Big picture: In recent years a number of studies have sought to build predictive models of chronological age using structural MRI data as input. Larger gaps between brain age and actual age, i.e., the

‘brain age gap’ (BAG) are often interpreted as indicative of poorer brain health or to reflect developmental ‘disruptions’. However, many different factors may influence brain age predictions – for example, a single dose of ibuprofen can reduce brain age estimates generated from grey matter morphometry data by 1 year <https://pmc.ncbi.nlm.nih.gov/articles/PMC6510235/> - making interpretation of findings difficult. Nonetheless, here the authors expand on the brain age literature by estimating this using diffusion tensor imaging data, specifically fractional anisotropy (FA), in a large sample of youth from the ABCD study. While the concept of studying brain age within the context of white matter maturation in a developmental cohort is thus somewhat novel, it is highly likely that brain age estimates derived from white matter data are also influenced by a large number of factors, including, but not limited to, those that are well-recognized confounds in diffusion-weighted imaging and in the use of simplistic tensor derived measures such as FA <https://pubmed.ncbi.nlm.nih.gov/22846632/>. Thus, one primary concern with this study is the concept of brain age itself, including as indexed using FA. For this reason, use of terms such as ‘delayed maturation’ (as is done throughout the discussion) to summarize findings should be used with caution.

Response:

We thank the reviewer for highlighting this conceptual issue and agree that interpretation of brain-age metrics requires caution, particularly in developmental cohorts and when derived from imaging modalities such as diffusion MRI. We have revised the Results and Discussion sections to clarify that our tract-based BAG estimates should be interpreted as deviation from age-normative trajectories rather than direct biological indicators of accelerated or delayed maturation. Accordingly, we have replaced interpretive terminology such as “delayed maturation” with more objective phrasing such as “lower BAG”.

To further support the developmental validity of the BAG measure, we added supplementary analyses demonstrating that higher BAG values are associated with more advanced pubertal stage and higher pubertal hormone levels, aligning with known developmental benchmarks. This concern was also raised by Reviewer 1, and we refer the Reviewer to our detailed responses to Reviewer 1 (Comments 1 & 6) for additional clarification and context.

The updated text is provided below:

Discussion:

“Limitation and future work” section. Page 28-29, line 673-680:

“Second, BAGs may be influenced by multiple factors beyond true neurodevelopmental variation. Previous studies have shown that both technical aspects (e.g. dMRI acquisition and preprocessing schemes⁶⁵) and non-technical factors (e.g. pharmacological manipulation⁶⁶) can affect BAG estimations. Although we validated the relationships between tract-based BAG and developmental benchmarks (pubertal stages and hormone levels), tract-based BAGs should nonetheless be interpreted as relative indices of deviation from normative WM developmental trajectories. Further work leveraging large-scale longitudinal datasets, integrating a broader range of lifestyle and environmental factors, and more biologically informed tract-based imaging modalities may help disentangle these sources of variability and improve interpretability.”

Reference:

65. Jones DK, Knosche TR, Turner R. White matter integrity, fiber count, and other fallacies: the do's and don'ts of diffusion MRI. *Neuroimage* 73, 239-254 (2013).

66. Le TT, et al. Effect of Ibuprofen on BrainAGE: A Randomized, Placebo-Controlled, Dose-Response Exploratory Study. *Biol Psychiatry Cogn Neurosci Neuroimaging* 3, 836-843 (2018).

Reviewer #3, Comment 2:

As far as I can tell, brain age predictions in ABCD were applied to both baseline and year 2 follow-up data in a single model (only a single r2 value is provided). However, given the sample size of each time point, there must have been overlap between the baseline and year two samples. Brain age estimates from the same interval calculated two years apart are presumably not independent. How was this addressed?

Response:

We thank the Reviewer for raising this point. To clarify, all brain-age models were trained exclusively on an independent HCP-D dataset, and no ABCD data (baseline or follow-up) were used for model building. The trained models were then applied separately to ABCD baseline and 2-year-follow-up datasets to generate predicted ages, ensuring strict out-of-sample evaluation and preventing data leakage. Although overlapping participants contributed data at both time points, these data were used only for out-of-sample testing, not for model training. For all downstream association analyses linking cognitive/clinical measures, we used tract-based BAGs derived from the baseline data as the primary results. When repeated measures were involved (associations with pubertal stages and hormone levels), we used linear mixed-effects models with participant as a random effect to appropriately account for within-subject dependence.

To further address the concern regarding dependence across timepoints, we further included the new dMRI data of ABCD 4-year-follow-up wave (Annual Curated Release 5.1). After controlling for test-set sample size across resampling samples (100 resampling times), we still observed significant predictive performance in the testing dataset with no participant overlap across timepoints. The two sampling strategies, by non-overlapping samples and overlapping samples, yielded highly similar predictive performance (Supplementary Figure 4). Of note, predictive performance (in term of R) improved as the age range of the testing sample increased. Together, these results suggested that (i) the predictive performance of the brain-age models is not driven by participant overlap across time points, (ii) age-range of the testing sample would influence the predictive performance of the brain age models, and (iii) our models exhibit strong generalizability and robustness across independent ABCD waves.

The full details are below:

Results:

Page 6, line 136-145:

“To evaluate the effects of age-range and intra-subject variability on the predictive performance of the brain-age models, we performed two additional analyses. ... Second, we resampled the same number of ABCD participants (N = 500 per timepoint) with 100

repetitions, and compared samples composed of non-overlapping subjects versus samples including overlapping subjects across the three time points. ... In the ABCD resampling experiment, the distributions of R values were comparable between the results from overlapped (0.592 ± 0.010 ; Supplementary Figure 4) and non-overlapped (0.580 ± 0.016 ; Supplementary Figure 4) samples. The overlapped samples exhibited only 1.3% greater variance explained, indicating that intra-subject overlap had limited influence on overall model performance.”

Supplementary materials:

Supplementary Figure 4:

Supplementary Figure 4. Influence of intra-subject overlap on tract-based brain-age prediction performance.

Density distributions of prediction accuracy (Pearson's R between chronological and predicted age) across 100 resampling iterations of ABCD participants. Each iteration sampled 500 participants per timepoint using either (i) non-overlapping samples (each subject included once across all time points) or (ii) overlapping samples (subjects allowed to appear in all time points). The two sampling strategies yielded highly similar predictive performance, with mean R values of 0.580 ± 0.016 for non-overlapping samples and 0.592 ± 0.010 for overlapping samples.

Reviewer #3, Comment 3:

The model's performance in ABCD is modest, particularly when done without age-bias correction. This is glossed over and presented in a supplemental table, with the in-text summary simply stating that all tracts were significant, however procedures for significance testing (e.g., permutation testing, multiple comparisons correction?) are not included in the methods (presumably this should be included around lines 813-814).

Response:

We apologize for the lack of clarity on the significance testing for model's performance. In the revised manuscript, we have now described in the Methods that statistical significance of the tract-wise age-prediction models were assessed via non-parametric permutation testing (1,000 iterations) by comparing the observed model R against the null distribution obtained by shuffling chronological ages across each iteration.

We quite agree with the Reviewer that the predictive performance of the tract-based models may be further improved through more advanced modeling algorithms or neuroimaging techniques. Nevertheless, all our tract-based brain-age models showed statistically significant chronological-age predictive performance in independent datasets, including in testing sets with narrower age ranges than those typically studied in adult aging cohorts, which limits the achievable R and RMSE values.

In addition, this comment raises a conceptual issue regarding the “ideal” predictive accuracy for brain age models. Perfect chronological age prediction (e.g. $R \approx 1$) would drive all BAG values toward zero, eliminating inter-individual variability and thus diminishing the biological and clinical interpretability of this metric. Consistent with recent methodological discussions in the brain-age and organ-age fields, our aim is not to maximize age-prediction accuracy, but rather to demonstrate biologically meaningful variation reflected in tract-specific BAGs. Supporting this, tract-specific models, with lower chronological-age correlation than the whole-brain model, showed more sensitivity in associating clinical/cognitive measures. For example, 9 tract-specific BAGs explained more variance for school grade at 2-y-follow-up, and 4 tract-specific BAGs explained more variance for transdiagnostic status at 2-y-follow-up, compared with the whole-brain BAG.

The manuscript has been updated as follows:

Methods:

Page 34, line 852-859:

“To assess whether tract-specific brain-age prediction exceeded chance, we generated an empirical null distribution by randomly permuting chronological ages across participants and re-establishing the prediction model on each permuted dataset. This procedure was repeated 1,000 times, yielding a null distribution of RMSE values under the assumption that there is no predictive utility of WM features on chronological age prediction. The p value is determined as the proportion of permutation-based RMSE values derived greater than or equal to the observed RMSE. Values of $p < 0.05$ were considered as the predictive performance significantly exceeded chance. Multiple comparisons across tracts ($n = 30$) were implemented using the Benjamini–Hochberg false-discovery-rate procedure (FDR) procedure.”

Reviewer #3, Comment 4:

To test clinical relevance, they identify associations between derived brain age estimates and cognition and clinical measures using sparse CCA in ABCD at baseline. They identify two components/modes – an Association mode and a Subcortical/Limbic mode. Both map onto various cognitive and clinical measures at baselines, as indicated by the CCA. The authors then test whether these modes predict academic performance, emotional Stroop performance and math ability at 2 and 3 year follow-up. However, it is unclear if this is driven by associations between the baseline behavioral variables, by the brain age estimates, or both. Finally, the authors use SVR to predict academic performance, emotional Stroop performance and math ability at 2 and 3 year follow-up specifically using the tract-based BAGs alone (i.e., not the CCA modes) and find some modest predictive power. Overall, this collection of analyses is somewhat unsatisfying and it is not clear if, for example, simple individual participant FA values for each tract, rather than BAGs, would do equally well (or better) in predicting these measures. The same concern applies to the prediction of KSADS diagnoses (i.e., are BAGs better predictors than ‘raw’ FA values)?

Response:

We thank the reviewer for these very important points. We have now added the comparison analyses by harmonized FA values to link to follow-up cognitive performance and KSADS

diagnoses. As shown in Supplementary Figure 13A, we found there are more tracts where BAGs explained more variance than whole-tract FAs for follow-up STRP_ACC and school grades. In addition, we also explored the relationships between harmonized FA values and KSADS diagnoses. As shown in Supplementary Table 13, we observed that more tracts where BAGs were significantly associated with KSADS diagnoses than whole-tract FAs. In conclusion, tract-based BAGs explained additional variance beyond FA in multiple tracts, suggesting that BAG captures complementary and differential developmental variance not reflected by raw FA alone.

The manuscript has been updated as follows:

Results:

Page 16, line 347-358:

“To evaluate whether tract-specific BAGs capture more and distinct variance for follow-up cognitive performance compared with tract-wise FA values, we conducted the same statistical analyses using baseline tract-wise FA values. As shown in Supplementary Figure 12, baseline FAs of the Cingulum Parahippocampal explained the greatest variance in 2-year follow-up school grades ($F_{(1, 7868)} = 49.15$, $\Delta R_{adjusted}^2 = 0.602$, $p_{FDR} < 0.001$) and 3-year follow-up SMARTE performance ($F_{(1, 5623)} = 88.31$, $\Delta R_{adjusted}^2 = 1.46$, $p_{FDR} < 0.001$), whereas baseline FAs of the FAT accounted for the highest variance in 2-year follow-up STRP_Acc ($F_{(1, 5967)} = 21.47$, $\Delta R_{adjusted}^2 = 0.338$, $p_{FDR} < 0.001$) among the 30 tracts. Across all three follow-up cognitive measures, a greater number of tracts showed higher variance explained by tract-specific BAGs than by FA values (Supplementary Figure 12A). In addition, we examined the similarity of effect patterns between tract-specific BAGs and FA values using Spearman correlation. As illustrated in Supplementary Figure 12B, the FA-based effect patterns did not significantly resemble those derived from the tract-specific BAGs, suggesting that tract-specific BAGs provide complementary and differential variance in predicting follow-up cognitive performance.”

Page 18, line 397-399:

“By contrast, none of the tract-wise FA values at baseline were significantly associated with the cumulative number of concurrent diagnoses when analyzed using the same statistical models (Supplementary Table 13).”

Page 20, line 439-445:

“In addition, supplementary comparative analyses using tract-wise FA values revealed that lower harmonized FA in the Corticospinal Tract (CST: $F_{(2, 7904)} = 12.98$, $p_{FDR} < 0.001$) and Cingulum Parahippocampal Parietal (C_PHP: $F_{(2, 7904)} = 5.14$, $p_{FDR} = 0.044$) were significantly associated with a greater number of diagnoses. Results for other tracts are provided in Supplementary Table 13. The tract-wise group effects based on FA exhibited a different directional pattern of variance in relation to the number of concurrent diagnoses compared with those observed for tract-wise BAGs (FA vs. BAG: $\rho = 0.366$, $p_{FDR} = 0.1380$, Supplementary Figure 14).”

Page 21, line 462-465:

“Comparative analyses showed that the tract-wise FAs in tracts such as the CST and the C_PHP were significantly associated with diagnostic status transitions. However, no significant relationship was observed between FA- and BAG-based group effects related to the diagnostic status transitions ($\rho = 0.205$, $p_{\text{FDR}} = 0.278$, Supplementary Figure 14).”

Discussion:

Page 27, line 635-638:

“Importantly, tract-based BAGs explained more variance than “raw” whole-tract FA values in many tracts, indicating that BAG captures developmentally specific variance beyond microstructural integrity alone. Incorporating age-normative information appear to provide complementary and differential sensitivity to individual differences relevant to cognitive development.”

Methods:

Page 39, line 997-1000:

“To assess whether tract-specific BAGs account for more variance in follow-up cognitive performance than raw FA values, we conducted the same statistical analyses using tract-wise FA values, averaged from the harmonized tract-wise FA values.”

Supplementary materials:

Supplementary Figure 13

Supplementary Figure 13. Comparison of tract-specific brain age gaps (BAGs) and whole-tract FAs in linking to follow-up cognitive performance.

(A) Effect sizes (ΔR^2) of tract-specific brain age gaps (BAGs; circles) and whole-tract fractional anisotropy (FAs; squares) accounting significant variance () for three follow-up cognitive measures: school grade, overall performance of Emotional Stroop Task (STRP Acc), and math ability evaluated by Stanford Mental Arithmetic Response Time Evaluation (SMARTE). Pie charts summarize the number of tracts where BAGs (green) explained more variance than FAs, versus tracts where FAs (gray) explained more variance than BAGs.

(B) Scatterplots showing correlations between the effect sizes of BAGs and FAs across tracts for the same three cognitive outcomes. Each dot represents a tract (colored by tract identity), plotted with FA effect on the x-axis and BAG effect on the y-axis. Correlation

coefficients (Spearman rho values) and p-values indicate the degree of similarity between BAG- and FA-derived effect size patterns across tracts.

Supplementary Table 13

Supplementary Table 13. GLM analysis results on the relationships between baseline tract-wise FA measures (with data harmonization) and the cumulative number of KSADS-5 at baseline, at 2-y-follow-up, and the transition of the transdiagnostic status between baseline and 2-y-follow-up.

	$BAG_{Baseline} \sim KSADS_{baseline}$		$BAG_{Baseline} \sim KSADS_{followup}$		$BAG_{Baseline} \sim KSADS_{conversion}$	
	F	p_{FDR}	F	p_{FDR}	F	p_{FDR}
Whole-brain	3.251	0.150	3.152	0.128	3.967	0.047
AF	1.508	0.369	0.464	0.786	2.145	0.137
FAT	2.001	0.254	1.945	0.268	2.900	0.090
PAT	1.308	0.386	2.797	0.166	1.162	0.387
SLF2	4.079	0.114	1.018	0.542	2.725	0.091
SLF3	0.128	0.943	0.345	0.817	0.455	0.738
SLF1	2.345	0.206	4.449	0.070	4.313	0.036
IFOF	3.914	0.114	1.139	0.506	2.114	0.137
ILF	4.765	0.085	3.941	0.097	2.852	0.090
UF	1.091	0.420	0.560	0.769	0.777	0.563
MdLF	2.604	0.173	1.467	0.407	3.276	0.075
VOF	0.048	0.970	0.528	0.769	0.406	0.749
OR	1.375	0.383	2.612	0.169	0.960	0.474
CST	5.273	0.085	12.983	0.000	6.804	0.004
C_PHP	2.900	0.150	5.137	0.044	4.617	0.036
C_PH	3.027	0.150	3.487	0.122	2.935	0.090
C_FP	2.141	0.235	1.385	0.417	2.351	0.111
C_FPH	1.214	0.387	0.143	0.866	1.815	0.185
C_PO	1.652	0.338	0.168	0.866	2.425	0.111
F	1.234	0.387	1.982	0.268	2.779	0.091
CS_A	0.132	0.943	0.236	0.866	2.385	0.111
CS_P	5.000	0.085	3.310	0.122	3.194	0.075
CS_S	2.989	0.150	2.696	0.169	2.352	0.111
TR_A	1.364	0.383	0.408	0.798	1.216	0.378
TR_P	0.663	0.618	2.067	0.268	2.619	0.098
TR_S	0.556	0.662	3.366	0.122	3.343	0.075
CC_Body	3.001	0.150	5.716	0.033	4.432	0.036
CC_Major	3.782	0.114	8.580	0.003	3.732	0.054
CC_Minor	0.030	0.970	0.174	0.866	1.980	0.156
CC_Tap	2.589	0.173	0.908	0.576	0.667	0.613

Supplementary Figure 14.

Supplementary Figure 14. Correlations between tract-specific BAG and FA group effects (F values of KSADS diagnoses) with measures related to transdiagnostic psychopathology. Scatterplots show the relationships between BAG- and FA-derived effects (y- and x-axes, respectively) for the cumulative number of KSADS-5 diagnoses at baseline (left), at 2-year follow-up (middle), and for conversion status between baseline and 2-year-follow-up (right). Each point represents a tract (colored by tract system).